# VARIATIONAL BAYESIAN PSEUDO-CORESET

**Hyungi Lee**
KAIST
lhk2708@kaist.ac.kr

**Seungyoo Lee**
KAIST
punctuate@kaist.ac.kr

**Juho Lee**
KAIST
juholee@kaist.ac.kr

## ABSTRACT

The success of deep learning requires large datasets and extensive training, which can create significant computational challenges. To address these challenges, pseudo-coresets, small learnable datasets that mimic the entire data, have been proposed. Bayesian Neural Networks, which offer predictive uncertainty and probabilistic interpretation for deep neural networks, also face issues with large-scale datasets due to their high-dimensional parameter space. Prior works on Bayesian Pseudo-Coresets (BPC) attempt to reduce the computational load for computing weight posterior distribution by a small number of pseudo-coresets but suffer from memory inefficiency during BPC training and sub-optimal results. To overcome these limitations, we propose Variational Bayesian Pseudo-Coreset (VBPC), a novel approach that utilizes variational inference to efficiently approximate the posterior distribution, reducing memory usage and computational costs while improving performance across benchmark datasets.

## 1 INTRODUCTION

While deep learning has shown remarkable performance across various fields, its success requires large amounts of data storage and extensive training. However, handling such large datasets can impose a significant computational burden, especially when training new models or updating existing ones with new data. In settings like continual learning, where the model must be trained continuously on new data, this challenge becomes more pronounced due to the risk of catastrophic forgetting. To mitigate this, a small subset of representative data, called a *coreset*, is needed to preserve knowledge from previously learned data. Instead of creating a small dataset as a subset of the entire data to represent it, the approach of treating the small dataset itself as learnable parameters and training it to mimic the entire dataset is known as *dataset distillation* or *pseudo-coreset* (Nguyen et al., 2020; 2021; Zhou et al., 2022; Loo et al., 2023).

On the other hand, Bayesian Neural Networks (BNNs) have gained attention in fields like healthcare (Abdullah et al., 2022; Lopez et al., 2023) and climate analysis (Vandal et al., 2018) because they provide a posterior distribution over the weights of a deep neural network, enabling the measurement of predictive uncertainty and allowing for a probabilistic interpretation of parameters (Papamarkou et al., 2024). While this method is promising for enabling various types of statistical analysis, BNNs face significant challenges when applied to real-world scenarios that involve large-scale datasets. The high-dimensional parameter space and structure of BNNs often lead to posterior landscapes with multiple modes, which complicates efficient and straightforward computation of predictive uncertainty. To overcome this, BNNs typically rely on indirect methods such as Stochastic Gradient Markov Chain Monte Carlo (SGMCMC; Welling & Teh, 2011; Chen et al., 2014; Ma et al., 2015) or variational inference (VI; Blei et al., 2017; Fiedler & Lucia, 2023; Harrison et al., 2024b) instead of directly calculating the posterior distribution in closed form. However, these approaches still depend on gradient-based updates of model weights for large-scale datasets. In particular, SGMCMC-based methods face the challenge of increased computational load, as the amount of training grows linearly with the number of weight samples needed.

To overcome these issues, prior works on *Bayesian Pseudo-Coreset* (BPC; Manousakas et al., 2020; Kim et al., 2022; 2023; Tiwary et al., 2024) aim to learn a small synthetic dataset that helps efficiently compute the posterior distribution of BNNs' weights. These studies train the pseudo-coreset by minimizing the divergence between the posterior obtained using the full dataset and the posterior obtained using the pseudo-coreset. However, these studies face three major problems: 1) require

expert trajectories for training, 2) use stop-gradient during training, and 3) still rely on SGMCMC sampling for weight space posterior computation. First, expert trajectories refer to the trajectories of model weights trained using the full dataset. In previous studies, these trajectories are saved for every epoch with multiple different seeds, and they are used to approximate and match the posterior distribution. This creates the problem of needing to store the model weights for the number of epochs multiplied by the number of seeds in order to train the pseudo-coreset. Secondly, when training BPC, the posterior distribution is computed using the BPC for loss computation via gradient-based methods. As the updates progress, the computational graph required to update the pseudo-coreset based on the loss becomes significantly larger, resulting in increased memory demands. To address this memory issue, prior works have used the stop-gradient method to reduce memory consumption. However, this approach leads to sub-optimal results because it prevents accurate updates. Finally, even after training the pseudo-coreset, the weight posterior distribution remains multi-modal, meaning that while the training cost is reduced, sequential training through SGMCMC sampling is still required for each sample. Additionally, after obtaining the samples, forward computation is needed for each sample to calculate the predictive distribution during Bayesian inference.

To address these issues, we propose a novel BPC approach called Variational Bayesian Pseudo-Coreset (VBPC). In learning VBPC, unlike previous works, we employ VI, specifically last-layer VI (Fiedler & Lucia, 2023; Harrison et al., 2024b), to approximate the posterior distribution. During the VBPC training and inference process, we demonstrate that this variational formulation allows us to obtain the closed-form posterior distribution of the last layer weights, which frees our method from relying on stop-gradient. This resolves the issue of suboptimal performance seen in previous approaches. And, we propose a memory-efficient method to approximate the predictive distribution with only a single forward pass instead of multiple forwards, making the approach computationally and memory-efficient. Furthermore, we empirically show that VBPC achieves better performance compared to other baselines on various benchmark datasets.

## 2 Preliminaries

### 2.1 Bayesian Neural Networks and Bayesian Model Averaging

In Bayesian Neural Network frameworks (Papamarkou et al., 2024; Lee et al., 2024), the main objective is to compute the predictive distribution for a given input $x$, while accounting for model uncertainty (i.e., epistemic uncertainty), as shown below:

$$p(y|x, \mathcal{D}) = \int p(y|x, \theta)p(\theta|\mathcal{D})\mathrm{d}\theta, \tag{1}$$

where $\mathcal{D}$ represents the observed data, and $\theta$ denotes the model parameters. This process is known as Bayesian Model Averaging (BMA). To perform BMA, we need to compute the posterior distribution $p(\theta|\mathcal{D})$ and evaluate the integral. However, due to the complexity of the model and the high-dimensional parameter space, directly computing a closed-form solution for $p(\theta|\mathcal{D})$ is impractical. Therefore, in practice, we typically rely on posterior sampling methods such as SGMCMC (Welling & Teh, 2011; Chen et al., 2014; Ma et al., 2015) or VI (Blei et al., 2017; Fiedler & Lucia, 2023) to approximate the posterior distribution.

### 2.2 Bayesian Pseudo-Coreset

As mentioned in Section 1, the large size of the training dataset makes it computationally intensive to perform SGMCMC or VI for approximating the posterior distribution of BNNs. To address these challenges and efficiently compute the posterior distribution in terms of both computation and memory, previous works (Kim et al., 2022; 2023; Tiwary et al., 2024) introduced BPC within the SGMCMC framework. Specifically, BPC $\mathcal{S}$ is optimized using the following objective:

$$\mathcal{S}^* = \arg\min_{\mathcal{S}} \ \mathrm{D}(p(\theta|\mathcal{D}), p(\theta|\mathcal{S})), \tag{2}$$

where D can be various divergences between the two distributions (Kim et al., 2022). The optimization poses a challenge, as the posteriors $p(\theta|\mathcal{D})$ and $p(\theta|\mathcal{S})$ are intractable for most of the cases. Previous works (Kim et al., 2022; 2023; Tiwary et al., 2024) attempt to approximate them using weight checkpoints obtained from training trajectories based on the dataset $\mathcal{D}$ (i.e., expert trajectories) which requires expensive computation and memory cost.

## 2.3 NATURAL GRADIENT VARIATIONAL INFERENCE WITH EXPONENTIAL FAMILIES

Although several methods exist for approximating the posterior $p(\theta|\mathcal{D})$, in this paper, we focus on VI (Bishop, 2006; Blundell et al., 2015; Blei et al., 2017). In VI, we approximate the target posterior with a variational distribution that is easier to handle and optimize the parameters of the variational distribution to minimize the Kullback-Leibler (KL) divergence between the approximate and target posterior distributions. Among the many possible choices for variational distributions, we focus on the *exponential family*. We assume that both the prior $p_{\lambda_0}(\theta)$ and the variational distribution $q_\lambda(\theta)$ belong to the same class of exponential family distributions:

$$q_\lambda(\theta) \propto \exp(\langle\psi(\theta),\lambda\rangle - A(\lambda)), \quad p_{\lambda_0}(\theta) \propto \exp(\langle\psi(\theta),\lambda_0\rangle - A(\lambda_0)), \tag{3}$$

where $\psi(\cdot)$ represents the sufficient statistics, $A(\cdot)$ is the log partition function, and $\lambda$ and $\lambda_0$ are the natural parameters for $q_\lambda$ and $p_{\lambda_0}$, respectively. We further assume that the exponential family is *minimal*, meaning that there is no non-zero vector $x$ such that $\langle x, \psi(\theta)\rangle$ evaluates to a constant. Under this setting, we can optimize the variational parameter $\lambda$ by minimizing the following loss:

$$\mathcal{L}_\mathcal{D}(\lambda) := \mathbb{E}_{q_\lambda}[-\log p(\mathcal{D}|\theta)] + \beta D_{\mathrm{KL}}[q_\lambda(\theta)\|p_{\lambda_0}(\theta)], \tag{4}$$

where $\beta > 0$ is a temperature controlling the strength of the KL regularization (Blundell et al., 2015; Wenzel et al., 2020). When $\beta = 1$, minimizing Eq. 4 is equivalent to minimizing $D_{\mathrm{KL}}[q_\lambda(\theta)\|p(\theta|\mathcal{D})]$. Optimizing equation 4 with natural gradient descent (Amari, 1998) has been shown to be effective, especially for large-scale deep neural networks (Khan et al., 2018; Osawa et al., 2019; Shen et al., 2024). The optimal solution of Eq. 4 must satisfy the following equation,

$$\lambda^* = \lambda_0 + \beta^{-1}\nabla_\mu \mathbb{E}_{q_{\lambda^*}}[\log p(\mathcal{D}|\theta)], \tag{5}$$

where $\mu = \mathbb{E}_{q_\lambda}[\psi(\theta)] = \nabla_\lambda A(\lambda)$ is the *mean parameter* corresponding to the natural parameter $\lambda$. Except for some cases, Eq. 5 does not admit a closed-form expression for $\lambda^*$. Therefore, one must rely on iterative algorithms to obtain it. This approach, which solves the variational inference using iterative natural gradient descent steps, covers a broad spectrum of machine learning algorithms and is commonly referred to as the Bayesian Learning Rule (BLR) (Khan & Rue, 2023).

## 3 VARIATIONAL BAYESIAN PSEUDOCORESET

In this section, we propose a novel method called Variational Bayesian Pseudo-Coreset (VBPC) which effectively learns $\mathcal{S}$ and thereby well approximates the variational posterior distribution with full dataset distribution. Several recent studies (Fiedler & Lucia, 2023; Harrison et al., 2024b) have shown that using only a last layer for variational inference is simple and computationally cheap, yet it performs comparably to more complex methods. Motivated by these findings, we seek to learn a pseudo-coreset $\mathcal{S}$ that effectively approximates the last layer variational posterior for the classification task, all while ensuring computational and memory efficiency.

### 3.1 PROBLEM SETUP

Consider a supervised learning problem with a dataset $\mathcal{D} = (x_i, y_i)_{i=1}^n$. While our discussion can be easily extended to more general problems, in this paper, we focus on $k$-way classification tasks, where $y_i \in \{0,1\}^k$ is a one-hot vector representing a category. Given $\mathcal{D}$ and a model $f_\theta$ parameterized by $\theta$, we aim to learn a synthetic dataset (pseudocoreset) $\mathcal{S} := (\hat{x}_i, \hat{y}_i)_{i=1}^{\hat{n}}$ solving Eq. 2 under a constraint $\hat{n} \ll n$. We approximate the pseudocoreset posterior $p(\theta|\mathcal{S})$ by solving the following variational inference problem,

$$\mathcal{L}_\mathcal{S}(\lambda) := \ell_\mathcal{S}(\lambda) + \beta_\mathcal{S} D_{\mathrm{KL}}[q_\lambda(\theta)\|p_{\lambda_0}(\theta)], \quad \lambda_\mathcal{S}^* = \arg\min_\lambda \mathcal{L}_\mathcal{S}(\lambda), \tag{6}$$

where $\ell_\mathcal{S}(\lambda) := -\mathbb{E}_{q_\lambda}[\sum_{i=1}^{\hat{n}} \log p_\mathcal{S}(y_i|x_i,\theta)]$ is the expected sum of negative log-likelihoods over $\mathcal{S}$ given a choice of likelihood $p_\mathcal{S}(y|x,\theta)$. Throughout the paper, we call Eq. 6 as *coreset VI* problem. Ideally, we would like to match the optimal solution of the coreset VI problem to the optimal variational distribution computed with the original dataset $\mathcal{D}$,

$$\mathcal{L}_\mathcal{D}(\lambda) := \ell_\mathcal{D}(\lambda) + \beta_\mathcal{D} D_{\mathrm{KL}}[q_\lambda(\theta)\|p_{\lambda_0}(\theta)], \quad \lambda_\mathcal{D}^* = \arg\min_\lambda \mathcal{L}_\mathcal{D}(\lambda), \tag{7}$$

where $\ell_\mathcal{D}(\lambda) := -\mathbb{E}_{q_\lambda}[\sum_{i=1}^n \log p_\mathcal{D}(y_i|x_i,\theta)]$ for a likelihood $p_\mathcal{D}(y|x,\theta)$. We call Eq. 7 as *dataset VI* problem. After obtaining $\lambda_\mathcal{S}^*$ and $\lambda_\mathcal{D}^*$, to learn $\mathcal{S}$, we can minimize $\mathrm{D}(q_{\lambda_\mathcal{S}^*}, q_{\lambda_\mathcal{D}^*})$ for some predefined divergence D.

## 3.2 BILEVEL OPTIMIZATION

It is often challenging to first compute the approximate solutions of Eqs. 6 and 7 and then backpropagate through the divergence $D(q_{\lambda_S^*}, q_{\lambda_D^*})$. Instead, considering the optimization nature of the VI, we cast the problem of coreset learning as a *bilevel optimization* as follows:

$$\mathcal{S}^* = \arg\min_{\mathcal{S}} \ \mathcal{L}_{\mathcal{D}}(\lambda_S^*) \text{ where } \lambda_S^* = \arg\min_{\lambda} \ \mathcal{L}_S(\lambda). \tag{8}$$

Note that similar approaches have also been considered in the dataset distillation literature (Loo et al., 2023). Under the bilevel optimization formulation, learning $\mathcal{S}$ requires the derivative

$$\nabla_{\mathcal{S}} \mathcal{L}_{\mathcal{D}}(\lambda_S^*) = (\nabla_{\mathcal{S}} \mu_S^*) \nabla_\mu \mathcal{L}_{\mathcal{D}}(\lambda_S^*), \tag{9}$$

where $\mu_S^* = \nabla_\lambda A(\lambda_S^*)$ is the mean parameter corresponding to $\lambda_S^*$. To obtain $\nabla_{\mathcal{S}} \mu_S^*$, we may apply the implicit function theorem (Bengio, 2000; Krantz & Parks, 2002) to Eq. 5. Specifically, if we let:

$$F(\mathcal{S}, \mu) := \lambda - \lambda_0 + \beta_S^{-1} \nabla_\mu \ell_S(\lambda) \tag{10}$$

With $F(\mathcal{S}, \mu_S^*) = 0$, applying the implicit function theorem,

$$\nabla_{\mathcal{S}} F(\mathcal{S}, \mu_S^*) + (\nabla_{\mathcal{S}} \mu_S^*) \nabla_\mu F(\mathcal{S}, \mu_S^*) = 0 \Rightarrow \nabla_{\mathcal{S}} \mu_S^* = -\nabla_{\mathcal{S}} F(\mathcal{S}, \mu_S^*) \nabla_\mu F(\mathcal{S}, \mu_S^*)^{-1},$$
$$\nabla_{\mathcal{S}} \mu_S^* = -\beta_S^{-1} (\nabla_{\mathcal{S}} \nabla_\mu \ell_S(\lambda_S^*)) \left( \nabla_\mu \lambda_S^* + \beta_S^{-1} \nabla_\mu^2 \ell_S(\lambda_S^*) \right)^{-1}. \tag{11}$$

Plugging this back into the above equation, we get the expression for the gradient

$$\nabla_{\mathcal{S}} \mathcal{L}_{\mathcal{D}}(\lambda_S^*) = -\beta_S^{-1} (\nabla_{\mathcal{S}} \nabla_\mu \ell_S(\lambda_S^*)) \left( \nabla_\mu \lambda_S^* + \beta_S^{-1} \nabla_\mu^2 \ell_S(\lambda_S^*) \right)^{-1} \nabla_\mu \mathcal{L}_{\mathcal{D}}(\lambda_S^*). \tag{12}$$

Unfortunately, the term involving the inverse is usually intractable, so one needs an approximation (e.g., Lorraine et al. (2020)). In the next section, we describe a case where the derivatives can be computed in closed form, and develop Bayesian pseudo-coreset algorithm based on it.

## 3.3 LAST LAYER VARIATIONAL BAYESIAN PSEUDOCORESET

Recently, there has been growing interest in subspace Bayesian neural networks (BNNs), where only a subset of the network's parameters are treated as random, while the remaining parameters are kept deterministic (Sharm et al., 2023; Shen et al., 2024). An extreme form of a subspace BNN would be the last layer randomization, where a neural network $f_\theta(x) \in \mathbb{R}^k$ is decomposed as a feature extractor $\phi(x) \in \mathbb{R}^h$ followed by a linear layer $W \in \mathbb{R}^{h \times k}$. Denoting the $j^{\text{th}}$ column of $W$ as $w_j$ and the $j^{\text{th}}$ output from $f_\theta(x)$ as $[f_\theta(x)]_j$, we have $[f_\theta(x)]_j = \phi(x)^\top w_j$ for $j \in [k]$. Adapting the last layer randomization scheme, we treat only the parameter $W$ of the linear layer as random while keeping the feature extractor $\phi(x)$ deterministic. From below, we describe our model more in detail.

**Variational distributions.** We assume the Gaussian priors and variational posteriors for $W$,

$$p_{\lambda_0}(W) = \prod_{j=1}^K \mathcal{N}(w_j | 0, \rho^{-1} I_h), \quad q_\lambda(W) = \prod_{j=1}^k \mathcal{N}(w_j | m_j, V_j), \tag{13}$$

with the natural parameters and the corresponding mean parameters are given as,

$$\lambda_0 = \text{concat}([0 \ -(\rho^{-1}/2) I_h]_{j=1}^k), \quad \mu_0 = \text{concat}([0, \rho^{-1} I_h]_{j=1}^k),$$
$$\lambda = \text{concat}((\lambda_j)_{j=1}^k), \quad \mu = \text{concat}((\mu_j)_{j=1}^k), \tag{14}$$

where $\lambda_j = \left[ V_j^{-1} m_j \ -(1/2) V_j \right]$, and $\mu_j = \left[ m_j, V_j + m_j m_j^\top \right]$. Here, we denote $I_d$ as the $d \times d$ identity matrix and $\rho$ is a pre-defined precision hyperparameter of the prior. Note that the block-wise approximation $q_\lambda(W)$ reduces the space complexity of the variance parameter $V := (V_j)_{j=1}^k$ from $O(k^2 h^2)$ to $O(k h^2)$ while keeping flexibility compare to mean field approximation.

**Likelihoods.** For a classification problem, it is common to use a softmax categorical likelihood, and we follow that convention for the dataset VI problem with $p_{\mathcal{D}}$. However, for the coreset VI problem, the softmax categorical likelihoods would not allow a closed-form solution, which would necessitate approximations involving iterative computations to solve the bilevel optimization Eq. 8. This would, for instance, require storing the unrolled computation graph (Vicol et al., 2021) of the iterative updates and performing backpropagation through it, leading to significant computational and memory overhead (Werbos, 1990). As a detour, we use the Gaussian likelihood for the $p_{\mathcal{S}}$, as it allows us to obtain a closed-form solution. While using Gaussian likelihoods may seem counterintuitive for a classification problem, it is widely used in the literature on infinitely-wide neural networks (Lee et al., 2017; 2019; 2022), and one can also interpret it as solving the classification problem as a regression, using one-hot labels as the target vector. More specifically, we set $p_{\mathcal{S}}(y|x,\theta) = \mathcal{N}(y|W^{\top}\phi(x),\gamma^{-1}I_k)$ where $\gamma^{-1}$ is the precision hyperparameter for the likelihood. With our choices for $p_{\mathcal{D}}$ and $p_{\mathcal{S}}$ we can expand the bilevel optimization problem as follows.

$$\lambda_{\mathcal{S}}^* = \arg\min_{\lambda} -\mathbb{E}_{q_{\lambda}}\left[\sum_{i=1}^{\hat{n}}\log\mathcal{N}(\hat{y}_i|W^{\top}\phi(\hat{x}_i),\gamma^{-1}I_k)\right] + \beta_{\mathcal{S}}D_{\mathrm{KL}}[q_{\lambda}\|p_{\lambda_0}], \quad (15)$$

$$\mathcal{S}^* = \arg\min_{\mathcal{S}}\ \mathbb{E}_{q_{\lambda_{\mathcal{S}}^*}}\left[-\sum_{i=1}^{n}\sum_{j=1}^{k}y_{i,j}\log\frac{\exp(\phi(x_i)^{\top}w_j)}{\sum_{l=1}^{k}\exp(\phi(x)^{\top}w_l)}\right] + \beta_{\mathcal{D}}D_{\mathrm{KL}}[q_{\lambda_{\mathcal{S}}^*}\|p_{\lambda_0}]. \quad (16)$$

## 3.4 SOLVING CORESET VI PROBLEM

Based on our choices described in the previous section, we show how we can obtain closed-form expressions for the coreset VI problem. The likelihood term for the coreset VI problem is

$$\mathbb{E}_{q_{\lambda}}\left[-\sum_{i=1}^{\hat{n}}\log\mathcal{N}(\hat{y}_i|W^{\top}\phi(\hat{x}_i),\gamma^{-1}I_k)\right] \overset{c}{=} \frac{\gamma}{2}\sum_{i=1}^{\hat{n}}\sum_{j=1}^{k}\mathbb{E}_{q_{\lambda}}\left[(\hat{y}_{i,j}-\phi(\hat{x}_i)^{\top}w_j)^2\right], \quad (17)$$

where $\hat{y}_{i,j}$ indicates $j$th element of $\hat{y}_i$ for all $i \in [\hat{n}]$ and $\overset{c}{=}$ denotes equality up to a constant. Then we can further elaborate Eq. 17 as follows:

$$\frac{\gamma}{2}\sum_{i=1}^{\hat{n}}\sum_{j=1}^{k}\mathbb{E}_{q_{\lambda}}\left[(\hat{y}_{i,j}-\phi(\hat{x}_i)^{\top}w_j)^2\right] \overset{c}{=} \frac{\gamma}{2}\sum_{j=1}^{k}\left(-2\hat{y}_{:,j}^{\top}\Phi\mu_j^{(1)} + \mathrm{Tr}\left(\Phi^{\top}\Phi\mu_j^{(2)}\right)\right), \quad (18)$$

where $\hat{y}_{:,j} := [\hat{y}_{1,j},\ldots,\hat{y}_{\hat{n},j}]^{\top}$, $\Phi := [\phi(\hat{x}_1),\ldots,\phi(\hat{x}_{\hat{n}})]^{\top}$, $\mu_j^{(1)} = m_j$, and $\mu_j^{(2)} = V_j + m_j m_j^{\top}$ for all $j \in [k]$. Then by Eq. 18, the gradient of the likelihood with respect to $\mu_j$ can be computed as:

$$\nabla_{\mu_j^{(1)}}\ell_{\mathcal{S}}(\lambda) = -\gamma\Phi^{\top}\hat{y}_{:,j}, \quad \nabla_{\mu_j^{(2)}}\ell_{\mathcal{S}}(\lambda) = \frac{\gamma}{2}\Phi^{\top}\Phi, \quad (19)$$

Then from Eq. 5, we obtain the closed-form solution for the coreset VI problem as follows:

$$\lambda_{\mathcal{S},j}^* = \left[\frac{\gamma}{\beta_{\mathcal{S}}}\Phi^{\top}\hat{y}_{:,j}\ -\frac{\rho}{2}I_h - \frac{\gamma}{2\beta_{\mathcal{S}}}\Phi^{\top}\Phi\right], \quad \forall j \in [k], \quad (20)$$

with Woodbury formula (Woodbury, 1950) which leads to

$$m_j^* = \Phi^{\top}\left(\frac{\rho\beta_{\mathcal{S}}}{\gamma}I_{\hat{n}} + \Phi\Phi^{\top}\right)^{-1}\hat{y}_{:,j}, \quad V_j^* = \frac{1}{\rho}I_h - \frac{\gamma}{\rho^2\beta_{\mathcal{S}}}\Phi^{\top}\left(I_{\hat{n}} + \frac{\gamma}{\rho\beta_{\mathcal{S}}}\Phi\Phi^{\top}\right)^{-1}\Phi. \quad (21)$$

For all $j \in [k]$, the values $V_j^*$ are identical, meaning the full covariance calculation, though $O(kh^2)$, only requires computing and storing the variance once, $O(h^2)$. We will refer to this shared variance as $V^*$. See Appendix A.1 and Appendix A.2 for detailed derivations in this section.

**Bilevel optimization as an influence maximization.** Before proceeding to the dataset VI problem, let us describe how the last-layer variational model simplifies the coreset gradient Eq. 12. From Eq. 19, we have $\nabla_{\mu}^2\ell_{\mathcal{S}}(\lambda_{\mathcal{S}}^*) = 0$, leading to $\nabla_{\mu}^2\mathcal{L}_{\mathcal{S}}(\lambda_{\mathcal{S}}^*) = \nabla_{\mu}\lambda_{\mathcal{S}}^*$. Using this, we can show that

$$\nabla_{\mathcal{S}}\mathcal{L}_{\mathcal{D}}(\lambda_{\mathcal{S}}^*) = \nabla_{\mathcal{S}}\left(-\nabla_{\mu}\mathcal{L}_{\mathcal{S}}(\lambda_{\mathcal{S}}^*)^{\top}\left(\nabla_{\mu}^2\mathcal{L}_{\mathcal{S}}(\lambda_{\mathcal{S}}^*)\right)^{-1}\nabla_{\mu}\mathcal{L}_{\mathcal{D}}(\lambda_{\mathcal{S}}^*)\right). \quad (22)$$

Here, $-\nabla_{\mu}\mathcal{L}_{\mathcal{S}}(\lambda_{\mathcal{S}}^*)^{\top}\left(\nabla_{\mu}^2\mathcal{L}_{\mathcal{S}}(\lambda_{\mathcal{S}}^*)\right)^{-1}\nabla_{\mu}\mathcal{L}_{\mathcal{D}}(\lambda_{\mathcal{S}}^*)$ is the variant (in a sense that it is defined w.r.t. the gradient of the variational objective by the mean parameters) of the *influence function* (Koh & Liang, 2017), measuring the influence of the coreset $\mathcal{S}$ on the dataset VI loss computed with $\mathcal{D}$.

### 3.5 COMPUTATION FOR DATASET VI PROBLEM

Now with these coreset VI problem solutions, we have to find the optimal $S^*$ by solving Eq. 16. However, unlike the coreset VI problem, since we use a categorical likelihood with a softmax output, a closed-form solution cannot be obtained from Eq. 16. Thus we have to use iterative updates, such as Stochastic Gradient Descent (SGD), for the outer optimization problem. Then because $\phi(x)^\top w_j \sim \mathcal{N}(\phi(x)^\top m_j^*, \phi(x)^\top V^* \phi(x))$ for all $j \in [k]$, the dataset VI problem changed into

$$\mathcal{L}_\mathcal{D}(\lambda_\mathcal{S}^*) = -\sum_{i=1}^{n}\sum_{j=1}^{k} y_{i,j} \mathbb{E}_{\mathbf{z}\sim\mathcal{N}(\bar{\mathbf{m}}^*(x_i),\bar{\Sigma}^*(x_i))}\left[\log\frac{\exp z_j}{\sum_{i=1}^{k}\exp z_i}\right] + \beta_\mathcal{D} D_{\mathrm{KL}}\left[q_{\lambda_\mathcal{S}^*}||p_{\lambda_0}\right], \quad (23)$$

where $\mathbf{z} = [z_1,\ldots,z_k]$, $\bar{\mathbf{m}}^*(x) = [\bar{m}_1^*(x),\ldots,\bar{m}_k^*(x)]$, $\bar{\Sigma}^*(x) = \mathrm{diag}([\Sigma_1^*(x),\ldots,\Sigma_k^*(x)])$ and $(\bar{m}_i^*(x), \Sigma_i^*(x)) = (\phi(x)^\top m_i^*, \phi(x)^\top V^*\phi(x))$ for all $j \in [k]$ and $x$. For a simpler notation, we will denote $(\bar{m}_i^*(x), \bar{\Sigma}_i^*(x))$ as $(\bar{m}_i^*, \bar{\Sigma}_i^*)$. Then we have to approximate $\mathbb{E}_{\mathbf{z}\sim\mathcal{N}(\bar{\mathbf{m}}^*,\bar{\Sigma}^*)}\left[\log\frac{\exp z_j}{\sum_{i=1}^{k}\exp z_i}\right]$ to compute the loss $\mathcal{L}_\mathcal{D}(\lambda_\mathcal{S}^*)$ analytically. To compute approximate expectation for the likelihood, we first change the form as follows:

$$\mathbb{E}_z\left[\log\frac{\exp(z_j)}{\sum_{i=1}^{k}\exp z_i}\right] = \int \log\left(2 - K + \sum_{i\neq j}\frac{1}{\sigma(z_j - z_i)}\right)^{-1}\mathcal{N}(\mathbf{z}|\bar{\mathbf{m}}^*,\bar{\Sigma}^*)\mathrm{d}\mathbf{z}, \quad (24)$$

where $\sigma(\cdot)$ is the sigmoid function. Then we utilize mean-field approximation (Lu et al., 2020) to the $z_i$s to approximately compute the Eq. 24:

$$\mathbb{E}_{\mathbf{z}\sim\mathcal{N}(\bar{\mathbf{m}}^*,\bar{\Sigma}^*)}\left[\log\frac{\exp(z_j)}{\sum_{i=1}^{t}\exp(z_i)}\right] \approx \left[\log\mathrm{softmax}\left(\frac{\bar{\mathbf{m}}^*}{\sqrt{1+\alpha\Sigma^*}}\right)\right]_j, \quad (25)$$

where $\alpha = \frac{\pi}{8}$ and $\Sigma^* = \phi(x)^\top V^*\phi(x)$. Refer to Appendix A.3 for the complete derivation of Eq. 23, Eq. 24, and Eq. 25. By Eq. 25, our outer optimization loss has changed form as follows:

$$\mathcal{L}_\mathcal{D}(\lambda_\mathcal{S}^*) = -\sum_{i=1}^{n}\sum_{j=1}^{k} y_{i,j}\log\mathrm{softmax}\left[\left(\frac{\bar{\mathbf{m}}^*(x_i)}{\sqrt{1+\alpha\Sigma^*(x_i)}}\right)\right]_j + \beta_\mathcal{D} D_{\mathrm{KL}}[q_{\lambda_\mathcal{S}^*}||p_{\lambda_0}]. \quad (26)$$

Here, since $n$ is large, we need to employ the SGD method to optimize $\mathcal{S}$. Thus, using the training batch $\mathcal{B} \subset \{(x_1,y_1),\ldots,(x_n,y_n)\}$, we compute approximate loss $\tilde{\mathcal{L}}_\mathcal{D}$ for the batch and update $\mathcal{S}$ using stochastic loss as follows:

$$\tilde{\mathcal{L}}_\mathcal{D}(\lambda_\mathcal{S}^*) = -\frac{n}{|\mathcal{B}|}\sum_{i\in\mathcal{B}}\sum_{j=1}^{k} y_{i,j}\log\mathrm{softmax}\left[\left(\frac{\bar{\mathbf{m}}^*(x_i)}{\sqrt{1+\alpha\Sigma^*(x_i)}}\right)\right]_j + \beta_\mathcal{D} D_{\mathrm{KL}}[q_{\lambda_\mathcal{S}^*}||p_{\lambda_0}]. \quad (27)$$

### 3.6 TRAINING AND INFERENCE

**Memory Efficient Loss computing**  If we naïvely compute the gradient of $\mathcal{S}$ by directly evaluating Eq. 27, calculating $\Sigma^*$ and $D_{\mathrm{KL}}[q_{\lambda_\mathcal{S}^*}||p_{\lambda_0}]$ will require computations involving $V^*$, which demands $h^2$ memory. However, the quadratic memory requirements with respect to the feature dimension pose a challenge when training $\mathcal{S}$ for large-scale models. To address this issue, we propose a memory-efficient approach for computing loss during training in this paragraph. We will address the efficient computation of $\Sigma^*$ in the below paragraph **Variational Inference and Memory Efficient Bayesian Model Averaging**. Here, we will focus on efficiently computing the KL divergence. Since both $q_{\lambda_\mathcal{S}^*}$ and $p_{\lambda_0}$ are Gaussian distributions, the KL divergence can be expressed as follows:

$$D_{\mathrm{KL}}[q_{\lambda_\mathcal{S}^*}||p_{\lambda_0}] \overset{c}{=} \frac{1}{2}\left[-k\log|\det(V^*)| + k\rho\mathrm{Tr}(V^*) + \rho\|m^*\|^2\right]. \quad (28)$$

Thus we have to efficiently compute $\det V^*$ and $\mathrm{Tr}(V^*)$. For the $\det V^*$, we use Weinstein-Aronszajn identity (Pozrikidis, 2014) which results as follows:

$$\det V^* = \frac{1}{\rho^h \det(I_{\hat{n}} + \frac{\gamma}{\rho\beta_\mathcal{S}}\Phi\Phi^\top)}. \quad (29)$$

And for the $\mathrm{Tr}(V^*)$, we can easily change the form with a property of matrix trace computation:

$$\mathrm{Tr}(V^*) = \frac{\beta_{\mathcal{S}}}{\gamma} \left( \frac{\gamma h}{\rho \beta_{\mathcal{S}}} - \left( \frac{\gamma}{\rho \beta_{\mathcal{S}}} \right)^2 \mathrm{Tr} \left( \left( I_{\hat{n}} + \frac{\gamma}{\rho \beta_{\mathcal{S}}} \Phi \Phi^\top \right)^{-1} \Phi \Phi^\top \right) \right). \tag{30}$$

By these formula, we can calculate the KL divergence without directly computing $V^*$, reducing the memory from $\mathcal{O}(h^2)$ to $\mathcal{O}(\hat{n}^2)$. Refer to Appendix A.4 for the derivation of Eq. 28 and Eq. 29.

**Model Pool**  If we train $\mathcal{S}$ based on only one $\phi$, it may overfit to that single $\phi$, resulting in an inability to properly generate the variational posterior for other $\phi$'s. This overfitting issue is common not only in Bayesian pseudo-coresets but also in the field of dataset distillation (Zhou et al., 2022). While several prior studies (Wang et al., 2018; 2022) tackle this overfitting problem, we address it by employing a model pool during training, following the approach of Zhou et al. (2022); Loo et al. (2023). This model pool method involves generating $P$ different $\theta_i$'s through random initialization during the training of $\mathcal{S}$ and storing them in a set $\mathcal{M} = \{\theta_i\}_{i=1}^{P}$. At each step, one $\theta$ is sampled from $\mathcal{M}$, and $\phi$ is constructed using this $\theta$. Then, $\mathcal{S}$ is trained for one step using SGD with this $\phi$. Afterward, $\theta$ is updated by training it for one step using $\mathcal{S}$ and the Gaussian likelihood, and the original $\theta$ in $\mathcal{M}$ is replaced with this updated version. Once each $\theta_i$ has been trained for a pre-defined number of $T$ steps, it is replaced with a new $\theta$ generated through random initialization. Through this process, $\mathcal{S}$ is trained with a new $\phi$ at every step, allowing it to generalize better across different $\phi$'s and become more robust to various initialization. See Algorithm 1 for a summary of the whole VBPC training procedure.

**Variational Inference and Memory Efficient Bayesian Model Averaging**  After training $\mathcal{S}$, we use it for variational inference. During variational inference, to improve the quality of the model's feature map $\phi$, we first train the randomly initialized $\theta$ using data sampled from $\mathcal{S}$ for a small number of steps $T'$ with a Gaussian likelihood. Then, using the trained feature map $\phi$, we compute the variational posterior by finding the optimal mean $m_j^*$ and variance $V^*$ for each $\theta_j^L$ as determined in the inner optimization. However, the variance $V^*$ we computed corresponds to a full covariance matrix, leading to a memory cost of $h^2$. To address this, rather than calculating $V^*$ explicitly, we need a memory-efficient approach for conducting BMA on test points. This can be done easily by :

$$\Sigma^* = \frac{\beta_{\mathcal{S}}}{\gamma} \left( \frac{\gamma}{\rho \beta_{\mathcal{S}}} \Phi_{\mathrm{te}} \Phi_{\mathrm{te}}^\top - \left( \frac{\gamma}{\rho \beta_{\mathcal{S}}} \right)^2 \Phi_{\mathrm{te}} \Phi^\top \left( I_{\hat{n}} + \frac{\gamma}{\rho \beta_{\mathcal{S}}} \Phi \Phi^\top \right)^{-1} \Phi \Phi_{\mathrm{te}}^\top \right), \tag{31}$$

where $\Phi_{\mathrm{te}} \in \mathbb{R}^{n_{\mathrm{te}} \times h}$ denotes the feature matrix of $n_{\mathrm{te}}$ number of test points. Then by storing $\Phi \in \mathbb{R}^{\hat{n} \times h}$ and $(I_{\hat{n}} + \frac{\gamma}{\rho \beta_{\mathcal{S}}} \Phi \Phi^\top)^{-1} \in \mathbb{R}^{\hat{n} \times \hat{n}}$ instead of $V^*$, we can reduce the memory requirements to $\hat{n}h + \hat{n}^2$, which is much smaller than $h^2$. Refer to Algorithm 2 for an overview of variational inference and BMA. This procedure does not require multiple forwards for BMA.

## 4 RELATED WORKS

**Bayesian Pseudo-Coreset**  As discussed in Section 1 and Section 2, the large scale of modern real-world datasets leads to significant computational costs when performing SGMCMC or variational inference to approximate posterior distributions. To address this issue, previous works, such as Bayesian Coreset (BC; Campbell & Broderick, 2018; 2019; Campbell & Beronov, 2019), have proposed selecting a small subset from the full training dataset so that the posterior distribution built from this subset closely approximates the posterior from the full dataset. However, Manousakas et al. (2020) highlighted that simply selecting a subset of the training data is insufficient to accurately approximate high-dimensional posterior distributions, and introduced BPC for simple logistic regression tasks. Later, Kim et al. (2022) extended BPC to BNNs, using reverse KL divergence, forward KL divergence, and Wasserstein distance as measures for D in Eq. 2 to assess the difference between the full posterior and the BPC posterior. Subsequent works have used contrastive divergence (Tiwary et al., 2024) or calculated divergence in function space (Kim et al., 2023). However, as discussed in Section 1, computational and memory overhead remains an issue when training BPC and during inference using BMA. For the additional related works, refer to Appendix C.

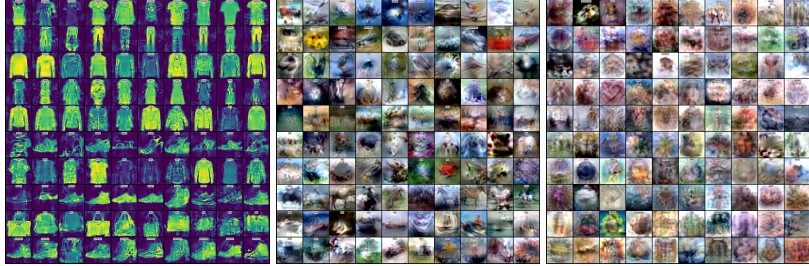

**Figure 1:** Learned VBPC images for the Fashion-MNIST (ipc=10; left), CIFAR10 (ipc=10; middle) and CIFAR100 (ipc=1; right) cases. These images construct trained mean for the distribution $\mathcal{S}^*$.

**Table 1:** Comparison of the VBPC with BPC baselines for the benchmark datasets. We report ACC and NLL for the VBPC and BPC baselines.

| Dataset | ipc | BPC-rKL ACC(↑) | NLL(↓) | BPC-fKL ACC(↑) | NLL(↓) | FBPC ACC(↑) | NLL(↓) | BPC-CD ACC(↑) | NLL(↓) | VBPC (Ours) ACC(↑) | NLL(↓) |
|---|---|---|---|---|---|---|---|---|---|---|---|
| MNIST | 1 | 74.8±1.2 | 1.90±0.01 | 83.0±2.2 | 1.87±0.03 | 92.5±0.1 | 1.68±0.01 | 93.4±0.1 | 1.53±0.01 | **96.7**±0.4 | **0.11**±0.02 |
| | 10 | 95.3±0.2 | 1.53±0.01 | 92.1±0.4 | 1.51±0.02 | 97.1±0.2 | 1.31±0.01 | 97.7±0.2 | 1.57±0.02 | **99.1**±0.1 | **0.03**±0.01 |
| | 50 | 94.2±0.3 | 1.36±0.02 | 93.6±1.8 | 1.36±0.02 | 98.6±0.1 | 1.39±0.02 | 98.9±0.2 | 1.36±0.01 | **99.4**±0.1 | **0.02**±0.01 |
| FMNIST | 1 | 70.5±1.1 | 2.47±0.02 | 72.5±2.5 | 2.30±0.02 | 74.7±1.4 | 1.81±0.03 | 77.3±0.5 | 1.90±0.03 | **82.9**±0.6 | **0.47**±0.03 |
| | 10 | 78.8±0.2 | 1.64±0.01 | 83.3±0.6 | 1.54±0.03 | 85.2±0.1 | 1.61±0.02 | 88.4±0.2 | 1.56±0.01 | **89.4**±0.2 | **0.30**±0.01 |
| | 50 | 77.0±0.6 | 1.48±0.02 | 74.8±0.5 | 1.47±0.02 | 76.7±0.4 | 1.46±0.02 | 89.5±0.1 | 1.30±0.02 | **91.0**±0.2 | **0.25**±0.01 |
| CIFAR10 | 1 | 21.6±0.8 | 2.57±0.01 | 29.3±1.1 | 2.10±0.03 | 35.5±0.3 | 3.79±0.04 | 46.9±0.2 | 1.87±0.02 | **55.1**±0.3 | **1.34**±0.08 |
| | 10 | 37.9±1.5 | 2.13±0.02 | 49.9±1.4 | 1.73±0.01 | 62.3±0.3 | 1.31±0.02 | 56.4±0.7 | 1.72±0.03 | **69.8**±0.7 | **0.89**±0.02 |
| | 50 | 37.5±1.3 | 1.93±0.03 | 42.3±2.9 | 1.54±0.01 | 71.2±0.2 | 1.03±0.05 | 71.9±0.2 | 1.57±0.03 | **76.7**±0.5 | **0.71**±0.03 |
| CIFAR100 | 1 | 3.6±0.1 | 4.69±0.02 | 14.7±0.2 | 4.20±0.10 | 21.0±0.8 | 3.76±0.11 | 24.0±0.1 | 4.01±0.02 | **38.4**±0.2 | **2.47**±0.04 |
| | 10 | 23.6±0.7 | 3.99±0.03 | 28.1±0.6 | 3.53±0.05 | 39.7±0.3 | 2.67±0.02 | 28.4±0.2 | 3.14±0.02 | **49.4**±0.1 | **2.07**±0.02 |
| | 50 | 30.8±0.5 | 3.57±0.17 | 37.1±0.3 | 3.28±0.24 | 44.5±0.4 | 2.63±0.05 | 39.6±0.2 | 3.02±0.01 | **52.4**±0.4 | **2.02**±0.02 |
| Tiny-ImageNet | 1 | 3.2±0.1 | 5.91±0.07 | 4.0±0.1 | 5.63±0.03 | 10.1±0.7 | 4.69±0.05 | 8.4±0.1 | 4.72±0.01 | **23.1**±0.2 | **3.65**±0.01 |
| | 10 | 9.8±0.6 | 5.26±0.05 | 11.4±0.5 | 5.08±0.05 | 19.4±0.5 | 4.14±0.02 | 17.8±0.4 | 3.64±0.05 | **25.8**±0.3 | **3.45**±0.02 |

## 5 EXPERIMENT

In this section, we present empirical results that demonstrate the effectiveness of posterior approximation using VBPC across various datasets and scenarios. We compare VBPC with four BPC algorithms that use SGMCMC to perform Bayesian Model Averaging (BMA) with posterior samples: BPC-rKL (Kim et al., 2022), BPC-fKL (Kim et al., 2022), FBPC (Kim et al., 2023), and BPC-CD (Tiwary et al., 2024). BPC-rKL and BPC-fKL employ reverse KL divergence and forward KL divergence, respectively, for the divergence term in Eq. 2. BPC-CD uses a more complex divergence called contrastive divergence, while FBPC also applies forward KL divergence but matches the posterior distribution in function space rather than weight space. Following all other prior works, we adopted a three-layer convolutional network with Batch Normalization (BN; Ioffe, 2015) as the base model architecture. For the target dataset, we used the MNIST (LeCun et al., 1998), Fashion-MNIST (Xiao et al., 2017), CIFAR10/100 (Krizhevsky, 2009), and Tiny-ImageNet (Le & Yang, 2015). Additionally, we used image-per-class (ipc) as the unit to count the number of pseudo-coresets. For a $k$-way classification task, $m$ ipc signifies that a total of $mk$ pseudo-coresets are trained. Along with evaluating classification accuracy (ACC) for each methods, we assess the performance of the resulting predictive distributions using negative log-likelihood (NLL).

In all tables, the best performance is indicated with **boldfaced underline**, while the second-best value is represented with underline in each row. See Appendix E for the additional experimental details.

### 5.1 BAYESIAN MODEL AVERAGING COMPARISON

We begin by evaluating the effectiveness of VBPC on five benchmark datasets by comparing the BMA performance across different methods. Table 1 clearly demonstrates that VBPC surpasses other BPC baselines across all benchmark datasets and ipcs in terms of ACC and NLL. Notably, VBPC achieves significantly better NLL, with large margins, while requiring only a single forward

**Table 2:** Comparison with dataset distillation baselines in terms of ACC. Here, $\downarrow$ indicates the performance drop compare to original method.

| Dataset | ipc | FRePo | FRePo VI | RCIG | RCIG VI | VBPC | AVBPC |
|---------|-----|-------|----------|------|---------|------|-------|
| **CIFAR10** | 1 | $46.8_{\pm0.7}$ | $28.2(18.6\downarrow)_{\pm0.9}$ | $53.9_{\pm1.0}$ | $27.8(24.1\downarrow)_{\pm0.7}$ | $\mathbf{55.1}_{\pm0.3}$ | $39.7(15.4\downarrow)_{\pm1.5}$ |
| | 10 | $65.5_{\pm0.4}$ | $55.7(9.8\downarrow)_{\pm0.5}$ | $69.1_{\pm0.4}$ | $55.6(13.8\downarrow)_{\pm1.5}$ | $\mathbf{69.8}_{\pm0.7}$ | $67.8(2.0\downarrow)_{\pm0.8}$ |
| **CIFAR100** | 1 | $28.7_{\pm0.1}$ | $19.9(8.8\downarrow)_{\pm0.4}$ | $\mathbf{39.3}_{\pm0.4}$ | $2.1(37.2\downarrow)_{\pm0.1}$ | $38.4_{\pm0.2}$ | $31.3(7.1\downarrow)_{\pm1.0}$ |
| | 10 | $42.5_{\pm0.2}$ | $34.8(7.7\downarrow)_{\pm0.4}$ | $44.1_{\pm0.4}$ | $2.5(41.6\downarrow)_{\pm0.4}$ | $\mathbf{49.4}_{\pm0.1}$ | $44.0(5.4\downarrow)_{\pm0.8}$ |

**Table 3:** Comparison of the VBPC with BPC baselines on the OOD setting with CIFAR10-C dataset. The $+A$ in the first column indicates that $A$ type corruption is applied to the CIFAR10 test dataset.

| Corruption | BPC-rKL ACC($\uparrow$) | BPC-rKL NLL($\downarrow$) | BPC-fKL ACC($\uparrow$) | BPC-fKL NLL($\downarrow$) | FBPC ACC($\uparrow$) | FBPC NLL($\downarrow$) | BPC-CD ACC($\uparrow$) | BPC-CD NLL($\downarrow$) | VBPC (Ours) ACC($\uparrow$) | VBPC (Ours) NLL($\downarrow$) |
|---|---|---|---|---|---|---|---|---|---|---|
| CIFAR10 | $37.9_{\pm1.5}$ | $2.13_{\pm0.02}$ | $49.9_{\pm1.4}$ | $1.73_{\pm0.01}$ | $62.3_{\pm0.3}$ | $\underline{1.31}_{\pm0.02}$ | $56.4_{\pm0.7}$ | $1.72_{\pm0.03}$ | $\mathbf{69.8}_{\pm0.7}$ | $\mathbf{0.89}_{\pm0.02}$ |
| +Gaussian Blur | $31.0_{\pm2.7}$ | $2.13_{\pm0.77}$ | $39.7_{\pm2.7}$ | $1.94_{\pm0.05}$ | $35.8_{\pm0.2}$ | $1.85_{\pm0.08}$ | $\underline{41.4}_{\pm0.7}$ | $\underline{1.73}_{\pm0.83}$ | $\mathbf{59.3}_{\pm0.9}$ | $\mathbf{1.20}_{\pm0.03}$ |
| +JPEG Compression | $30.4_{\pm0.9}$ | $2.13_{\pm0.02}$ | $37.3_{\pm2.9}$ | $1.95_{\pm0.06}$ | $\underline{40.1}_{\pm0.1}$ | $1.73_{\pm0.02}$ | $37.3_{\pm0.2}$ | $\underline{1.71}_{\pm0.03}$ | $\mathbf{61.9}_{\pm0.8}$ | $\mathbf{1.12}_{\pm0.02}$ |
| +Snow | $26.9_{\pm1.7}$ | $2.20_{\pm0.07}$ | $35.7_{\pm2.7}$ | $2.00_{\pm0.07}$ | $\underline{38.6}_{\pm0.4}$ | $\underline{1.78}_{\pm0.16}$ | $37.8_{\pm0.6}$ | $1.91_{\pm0.05}$ | $\mathbf{59.0}_{\pm0.1}$ | $\mathbf{1.20}_{\pm0.02}$ |
| +Zoom Blur | $31.7_{\pm1.2}$ | $2.09_{\pm0.04}$ | $35.1_{\pm2.9}$ | $2.04_{\pm0.07}$ | $28.9_{\pm0.2}$ | $2.19_{\pm0.11}$ | $\underline{38.3}_{\pm0.8}$ | $\underline{1.93}_{\pm0.13}$ | $\mathbf{58.1}_{\pm0.8}$ | $\mathbf{1.23}_{\pm0.04}$ |
| +Pixelate | $29.0_{\pm2.3}$ | $2.19_{\pm0.07}$ | $\underline{39.1}_{\pm3.2}$ | $1.93_{\pm0.06}$ | $38.0_{\pm0.1}$ | $\underline{1.77}_{\pm0.04}$ | $39.0_{\pm1.5}$ | $1.92_{\pm0.07}$ | $\mathbf{58.8}_{\pm0.9}$ | $\mathbf{1.26}_{\pm0.04}$ |
| +Defocus Blur | $27.6_{\pm1.3}$ | $2.20_{\pm0.05}$ | $36.7_{\pm3.7}$ | $1.99_{\pm0.08}$ | $31.7_{\pm0.4}$ | $2.07_{\pm0.19}$ | $\underline{37.2}_{\pm1.0}$ | $\underline{1.87}_{\pm0.04}$ | $\mathbf{63.0}_{\pm0.7}$ | $\mathbf{1.08}_{\pm0.02}$ |
| +Motion Blur | $17.4_{\pm2.5}$ | $2.73_{\pm0.14}$ | $35.2_{\pm3.3}$ | $2.01_{\pm0.05}$ | $27.9_{\pm0.2}$ | $2.29_{\pm0.15}$ | $\underline{37.1}_{\pm0.5}$ | $\underline{1.92}_{\pm0.04}$ | $\mathbf{55.9}_{\pm0.5}$ | $\mathbf{1.32}_{\pm0.03}$ |

pass for BMA. These results empirically validate that the variational distribution trained by VBPC effectively captures epistemic uncertainty with a small amount of synthetic data, while keeping performance. Refer to Fig. 1 for examples of VBPC-trained images from the Fashion-MNIST, CIFAR10, and CIFAR100 datasets. For more trained VBPC images for other settings, see Appendix G.

**Comparison with dataset distillation baselines** In addition to the BPC baselines, we compared VBPC with two notable dataset distillation baselines, FRePo (Zhou et al., 2022) and RCIG (Loo et al., 2023), which are recognized for their strong accuracy performance. Since FRePo and RCIG do not employ cross-entropy loss for training, we only report ACC, as comparing NLL would be unfair. As shown in Table 2, although VBPC is designed to learn pseudo-coresets to approximate the variational distribution from the training data, it outperforms these dataset distillation baselines, focused mainly on ACC, in nearly all tasks except for CIFAR100 with 1 ipc. The results for each methods (i.e., **FRePo**, **RCIG**, and **VBPC**) in Table 2 were evaluated based on each baseline's evaluation methods. However, one might question whether the significant performance of VBPC is due to the trained pseudo-coreset itself or the VI method. To verify that VBPC's performance isn't solely due to the VI method, we applied our VI method to the baselines' pseudo-coresets (i.e., **FRePo VI** and **RCIG VI**) and used FRePo's method to evaluate VBPC's pseudo-coresets (i.e., **AVBPC**). Although all methods saw some performance decline, VBPC exhibited a smaller drop, indicating that its performance is not solely due to the VI method, but to its ability to effectively learn the variational distribution. Full comparisons across all benchmark datasets, available in Appendix F.1, show that VBPC maintains a consistent trend over dataset distillation baselines across all the datasets.

## 5.2 RESULTS ON OUT OF DISTRIBUTION SCENARIOS

To further demonstrate that the predictive distribution derived from the VBPC dataset enhances robustness to distributional shifts and out-of-distribution (OOD) data, we assess the performance of VBPC and BPC baselines on a corrupted version of the CIFAR10 dataset, known as CIFAR10-C (Hendrycks & Dietterich, 2019). In this case, we use the CIFAR10 10ipc BPC data trained in Section 5.1 for all methods and evaluate their performance on the corrupted dataset across 7 different types of corruption. We assess performance using all 5 levels of severity provided in the dataset. Table 3 clearly illustrates that VBPC shows strong robustness against various types of corruption and consistently outperforms other baselines across all corruption types in terms of both ACC and NLL. These findings highlight that the predictive distribution obtained from the VBPC dataset improves robustness to distributional shifts and OOD scenarios.

**Table 4:** Comparison of the VBPC with BPC baselines on the architecture generalization. The $A - B$ in the first column indicates that $B$ type normalization layer is used for the $A$ model.

| Model | BPC-rKL ACC(↑) | NLL(↓) | BPC-fKL ACC(↑) | NLL(↓) | FBPC ACC(↑) | NLL(↓) | BPC-CD ACC(↑) | NLL(↓) | VBPC (Ours) ACC(↑) | NLL(↓) |
|---|---|---|---|---|---|---|---|---|---|---|
| Conv-BN | $37.9_{\pm1.5}$ | $2.13_{\pm0.02}$ | $49.9_{\pm1.4}$ | $1.73_{\pm0.01}$ | $\underline{62.3}_{\pm0.3}$ | $\underline{1.31}_{\pm0.02}$ | $56.4_{\pm0.7}$ | $1.72_{\pm0.03}$ | $\mathbf{69.8}_{\pm0.7}$ | $\mathbf{0.89}_{\pm0.02}$ |
| Conv-NN | $23.1_{\pm3.8}$ | $2.22_{\pm0.02}$ | $22.9_{\pm4.4}$ | $2.12_{\pm0.04}$ | $28.6_{\pm4.8}$ | $2.17_{\pm0.02}$ | $\underline{30.1}_{\pm4.4}$ | $\underline{2.05}_{\pm0.19}$ | $\mathbf{58.4}_{\pm0.8}$ | $\mathbf{1.46}_{\pm0.05}$ |
| Conv-GN | $28.5_{\pm4.5}$ | $2.85_{\pm0.23}$ | $29.1_{\pm4.4}$ | $2.81_{\pm0.24}$ | $\underline{31.5}_{\pm5.2}$ | $\underline{1.93}_{\pm0.01}$ | $23.8_{\pm4.2}$ | $3.07_{\pm0.42}$ | $\mathbf{66.8}_{\pm0.6}$ | $\mathbf{0.95}_{\pm0.04}$ |
| Conv-IN | $26.7_{\pm4.3}$ | $2.81_{\pm0.22}$ | $27.7_{\pm4.7}$ | $2.82_{\pm0.25}$ | $\underline{31.7}_{\pm5.3}$ | $\underline{1.96}_{\pm0.09}$ | $26.9_{\pm4.4}$ | $3.29_{\pm0.27}$ | $\mathbf{58.1}_{\pm0.8}$ | $\mathbf{1.22}_{\pm0.12}$ |
| AlexNet-NN | $24.2_{\pm3.8}$ | $2.23_{\pm0.01}$ | $21.4_{\pm4.3}$ | $2.82_{\pm0.24}$ | $\underline{32.1}_{\pm0.9}$ | $2.91_{\pm0.05}$ | $30.8_{\pm1.4}$ | $\underline{2.24}_{\pm0.11}$ | $\mathbf{48.0}_{\pm0.4}$ | $\mathbf{1.94}_{\pm0.05}$ |
| ResNet18-BN | $9.6_{\pm2.6}$ | $3.27_{\pm0.15}$ | $10.5_{\pm4.5}$ | $3.16_{\pm0.14}$ | $\underline{46.7}_{\pm1.2}$ | $\underline{1.81}_{\pm0.08}$ | $41.7_{\pm1.1}$ | $2.05_{\pm0.27}$ | $\mathbf{54.9}_{\pm0.5}$ | $\mathbf{1.36}_{\pm0.05}$ |
| VGG11-GN | $10.0_{\pm2.9}$ | $2.94_{\pm0.11}$ | $10.1_{\pm3.0}$ | $2.85_{\pm0.11}$ | $37.2_{\pm0.9}$ | $\underline{1.40}_{\pm0.05}$ | $\underline{44.5}_{\pm1.2}$ | $1.78_{\pm0.12}$ | $\mathbf{52.4}_{\pm1.1}$ | $\mathbf{1.44}_{\pm0.15}$ |

**Table 5:** Ablation results on memory allocation and time requirements on CIFAR10 10ipc.

| Naïve Training Memory (MB) | sec/100 steps | Training (Ours) Memory (MB) | sec/100 steps | Naïve BMA Memory (MB) | BMA (Ours) Memory (MB) |
|---|---|---|---|---|---|
| 542.9 | 54.0 | 272.9 | 9.9 | 542.9 | 268.9 |

## 5.3 Architecture generalization

To demonstrate that VBPC can be applied when performing BMA on unseen architectures, we conduct BMA using different model structures with various normalization layers. Specifically, we include the identity layer (NN), Group Normalization (GN; Wu & He, 2018), and Instance Normalization (IN; Ulyanov, 2016) as additional normalization methods. We also incorporate AlexNet (Krizhevsky et al., 2012), ResNet18 (He et al., 2016), and VGG11 (Simonyan & Zisserman, 2014) as new model architectures. Similar to Section 5.2, we use the CIFAR10 10ipc BPC data. As shown in Table 4, VBPC successfully performs VI across various architectures and effectively constructs predictive distributions through BMA. Notably, while other baselines are sensitive to changes in normalization layers, VBPC demonstrates robust learning over diverse feature maps through the model pool, resulting in strong ACC and NLL performance.

## 5.4 Memory allocation and time requirements

In this section, we perform an ablation study to compare memory usage and time requirements between the naive computation and the efficient computation for the variance $V^*$, $\Sigma^*$, and the loss terms during both training and inference. As we discussed in Section 3.6, naive loss computation requires $O(h^2)$ space complexity and $O(h^3)$ computational complexity. However, our computationally efficient loss computation method only requires $O(\hat{n}^2)$ space complexity and $O(\hat{n}^3)$ computational complexity. Therefore, in the BPC setting where $\hat{n} \ll h$ typically holds, we can significantly reduce the space and computational complexity required for training. This difference can be observed during the actual training process. As shown in Table 5, our computationally efficient training reduces the memory requirements for loss computation by nearly half and decreases the training time to under $20\%$. Also, we can see the similar results during the BMA procedure. Refer to Appendix F to see the various additional ablation studies including ablation on hyperparameters, pseudo-coreset initialization, and augmentations.

## 6 Conclusion

In this paper, we present a novel BPC method for VI, referred to as VBPC. By utilizing the Gaussian likelihood, we enable the computation of a closed-form solution for coreset VI, thereby removing the need to unroll the computation graph or use stop gradients. Leveraging this closed-form solution, we propose a method to approximate dataset VI without weight sampling during the training of VBPC. Additionally, we introduce a computationally efficient training and BMA inference method that significantly reduces both computational and space complexity. Finally, we empirically show that the variational distribution obtained from VBPC substantially outperforms the predictive distributions derived from other BPC baselines in BMA performance across various scenarios.

**Reproducibility Statement.** We present comprehensive derivations of all equations in the paper in Appendix A. The overall algorithms can be found in Appendix B. Details regarding the datasets, model architecture, data preprocessing, and hyperparameters are provided in Appendix E.

**Ethics Statement.** We propose a method that improves the computational and memory efficiency of the variational inference method for posterior approximation in Bayesian Neural Networks. Thus although our approach does not have a direct positive or negative impact on ethical or societal aspects, it can enhance privacy preservation. Specifically, our method facilitates Bayesian inference using private training data in neural network models by generating synthetic datasets, allowing for the computation of the posterior distribution while maintaining privacy.

**Acknowledgements** This work was partly supported by Institute of Information & communications Technology Planning & Evaluation(IITP) grant funded by the Korea government(MSIT) (No.RS-2019-II190075, Artificial Intelligence Graduate School Program(KAIST)), the National Research Foundation of Korea(NRF) grant funded by the Korea government(MSIT) (RS-2021-NR056917), Institute of Information & communications Technology Planning & Evaluation(IITP) grant funded by the Korea government(MSIT) (No.RS-2024-00509279, Global AI Frontier Lab), and Institute of Information & communications Technology Planning & Evaluation(IITP) grant funded by the Korea government(MSIT) (No.RS-2022-II220713, Meta-learning Applicable to Real-world Problems).

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

## A  FULL DERIVATIONS

### A.1  FULL DERIVATION FOR THE INNER OPTIMIZATION

In this section, we present the full derivation calculation for the inner optimization in Section 3.4. Let us first examine the term $\mathbb{E}_{q_\lambda}\left[-\sum_{i=1}^{\hat{n}}\log\mathcal{N}(\hat{y}_i|W^\top\phi(\hat{x}_i),\gamma^{-1}I_k)\right]$, which can be computed as follows:

$$\mathbb{E}_{q_\lambda}\left[-\sum_{i=1}^{\hat{n}}\log\mathcal{N}(\hat{y}_i|W^\top\phi(\hat{x}_i),\gamma^{-1}I_k)\right] = -\sum_{i=1}^{\hat{n}}\mathbb{E}_{q_\lambda}\left[\log\mathcal{N}(\hat{y}_i|W^\top\phi(\hat{x}_i),\gamma^{-1}I_k)\right] \tag{32}$$

$$= \sum_{i=1}^{\hat{n}}\mathbb{E}_{q_\lambda}\left[\frac{\gamma}{2}||\hat{y}_i-\phi(\hat{x}_i)^\top W||^2\right] + \text{constant} \tag{33}$$

$$= \frac{\gamma}{2}\sum_{i=1}^{\hat{n}}\sum_{j=1}^{k}\mathbb{E}_{q_\lambda}\left[\left(\hat{y}_{i,j}-\phi(\hat{x}_i)^\top w_j\right)^2\right] + \text{constant} \tag{34}$$

$$\stackrel{c}{=} \frac{\gamma}{2}\sum_{i=1}^{\hat{n}}\sum_{j=1}^{k}\mathbb{E}_{q_\lambda}\left[\left(\hat{y}_{i,j}-\phi(\hat{x}_i)^\top w_j\right)^2\right], \tag{35}$$

where $\hat{y}_{i,j}$ indicates $j$th element of $\hat{y}_i$ for all $i \in [\hat{n}]$. With this approximation, now we can compute $\mathbb{E}_{q_\lambda}\left[-\sum_{i=1}^{\hat{n}}\log\mathcal{N}(\hat{y}_i|W^\top\phi(\hat{x}_i),\gamma^{-1}I_k)\right]$ as follows:

$$\mathbb{E}_{q_\lambda}\left[-\sum_{i=1}^{\hat{n}}\log\mathcal{N}(\hat{y}_i|W^\top\phi(\hat{x}_i),\gamma^{-1}I_k)\right] \stackrel{c}{=} \frac{\gamma}{2}\sum_{i=1}^{\hat{n}}\sum_{j=1}^{k}\mathbb{E}_{q_\lambda}\left[\left(\hat{y}_{i,j}-\phi(\tilde{x}_i)^\top w_j\right)^2\right] \tag{36}$$

$$= \frac{\gamma}{2}\sum_{j=1}^{k}\mathbb{E}_{q_\lambda}\left[\hat{y}_j^\top\hat{y}_j - 2\hat{y}_j^\top\Phi w_j + w_j^\top\Phi^\top\Phi w_j\right] \tag{37}$$

$$\stackrel{c}{=} \frac{\gamma}{2}\sum_{j=1}^{k}\left(-2\hat{y}_j^\top\Phi m_j + \mathbb{E}_{q_\lambda}\left[w_j^\top\Phi^\top\Phi w_j\right]\right) \tag{38}$$

$$= \frac{\gamma}{2}\sum_{j=1}^{k}\left(-2\hat{y}_j^\top\Phi m_j + \text{Tr}\left(\Phi^\top\Phi\mathbb{E}_{q_\lambda}\left[w_j w_j^\top\right]\right)\right) \tag{39}$$

$$= \frac{\gamma}{2}\sum_{j=1}^{k}\left(-2\hat{y}_j^\top\Phi m_j + \text{Tr}\left(\Phi^\top\Phi\left[V_j + m_j m_j^\top\right]\right)\right) \tag{40}$$

$$= \frac{\gamma}{2}\sum_{j=1}^{k}\left(-2\hat{y}_j^\top\Phi\mu_j^{(1)} + \text{Tr}\left(\Phi^\top\Phi\mu_j^{(2)}\right)\right), \tag{41}$$

where $\hat{y}_j := [\hat{y}_{1,j},\dots,\hat{y}_{\hat{n},j}]^\top$, $\Phi := [\phi(\hat{x}_1),\dots,\phi(\hat{x}_{\hat{n}})]$, $\mu_j^{(1)} = m_j$, and $\mu_j^{(2)} = V_j + m_j m_j^\top$ for all $j \in [k]$. Here, $\stackrel{c}{=}$ denotes equality up to a constant. Eq. 39 derived from the fact that $\mathbb{E}_{q_\lambda}\left[w_j^\top\Phi^\top\Phi w_j\right]$ is scalar value and the property of the Tr function.

### A.2  NUMERICALLY STABLE MEAN AND VARIANCE

In this section, we present the full derivation calculation for the numerically stable mean and variance in Section 3.4. Due to the dimension of $\Phi$ is $\hat{n} \times h$ and usually $\hat{n} \ll h$, naïve computation of $m_j^*$ and $V_j^*$ lead numerically unstable results. To address this issue, we transformed the formulas for $m_j^*$ and $V_j^*$ into equivalent but more numerically stable forms. Specifically, when calculating $V_j^*$, we applied the Woodbury formula (Woodbury, 1950). First, we utilize the kernel trick to make

mean $m_j^*$ more numerically stable. The derivation is as follows:

$$m_j^* = -\frac{1}{2}\lambda_j^{(2)*-1}\lambda_j^{(1)*} \tag{42}$$

$$= -\frac{1}{2}(-\frac{\rho}{2}I_h - \frac{\gamma}{2\beta_S}\Phi^\top\Phi)^{-1}\frac{\gamma}{\beta_S}\Phi^\top\hat{y}_j \tag{43}$$

$$= \frac{1}{2}(\frac{\rho}{2}I_h + \frac{\gamma}{2\beta_S}\Phi^\top\Phi)^{-1}\frac{\gamma}{\beta_S}\Phi^\top\hat{y}_j \tag{44}$$

$$= (\rho I_h + \frac{\gamma}{\beta_S}\Phi^\top\Phi)^{-1}\frac{\gamma}{\beta_S}\Phi^\top(\rho I_{\hat{n}} + \frac{\gamma}{\beta_S}\Phi\Phi^\top)(\rho I_{\hat{n}} + \frac{\gamma}{\beta_S}\Phi\Phi^\top)^{-1}\hat{y}_j \tag{45}$$

$$= \frac{\gamma}{\beta_S}(\rho I_h + \frac{\gamma}{\beta_S}\Phi^\top\Phi)^{-1}(\rho\Phi^\top + \frac{\gamma}{\beta_S}\Phi^\top\Phi\Phi^\top)(\rho I_{\hat{n}} + \frac{\gamma}{\beta_S}\Phi\Phi^\top)^{-1}\hat{y}_j \tag{46}$$

$$= \frac{\gamma}{\beta_S}\Phi^\top(\rho I_{\hat{n}} + \frac{\gamma}{\beta_S}\Phi\Phi^\top)^{-1}\hat{y}_j \tag{47}$$

$$= \Phi^\top(\frac{\rho\beta_S}{\gamma}I_{\hat{n}} + \Phi\Phi^\top)^{-1}\hat{y}_j. \tag{48}$$

Next, we utilize the Woodbury formula (Woodbury, 1950) to make variance $V_j^*$ more numerically stable. The derivation is as follows:

$$V_j^* = \frac{\beta_S}{\gamma}\left(\frac{\rho\beta_S}{\gamma}I_h + \Phi^\top\Phi\right)^{-1} \tag{49}$$

$$= \frac{\beta_S}{\gamma}\left(\left(\frac{\rho\beta_S}{\gamma}I_h\right)^{-1} - \left(\frac{\rho\beta_S}{\gamma}I_h\right)^{-1}\Phi^\top\left(I_{\hat{n}}^{-1} + \Phi\left(\frac{\rho\beta_S}{\gamma}I_h\right)^{-1}\Phi^\top\right)^{-1}\Phi\left(\frac{\rho\beta_S}{\gamma}I_h\right)^{-1}\right) \tag{50}$$

$$= \frac{\beta_S}{\gamma}\left(\frac{\gamma}{\rho\beta_S}I_h - \left(\frac{\gamma}{\rho\beta_S}\right)^2\Phi^\top\left(I_{\hat{n}} + \frac{\gamma}{\rho\beta_S}\Phi\Phi^\top\right)^{-1}\Phi\right) \tag{51}$$

$$= \frac{1}{\rho}I_h - \frac{\gamma}{\rho^2\beta_S}\Phi^\top\left(I_{\hat{n}} + \frac{\gamma}{\rho\beta_S}\Phi\Phi^\top\right)^{-1}\Phi. \tag{52}$$

It is important to note that for all $j \in [k]$, the $V_j^*$ values are identical. This implies that while calculating the full covariance for all $j \in [k]$ can be computationally intensive (i.e. $O(kh^2)$), we only need to compute and store the variance once (i.e. $O(h^2)$).

## A.3 FULL DERIVATION FOR OUTER OPTIMIZATION PROBLEM

In this section, we present the full derivation for the outer optimization problem. Here, we first change $\mathcal{L}_\mathcal{D}(\lambda_S^*)$ as follows:

$$\mathcal{L}_\mathcal{D}(\lambda_S^*) = \mathbb{E}_{\theta^L \sim q_{\lambda_S^*}}[-\sum_{i=1}^{n}\log p_\mathcal{D}(y_i|x_i, \theta^L)] + \beta_\mathcal{D}D_{\mathrm{KL}}[q_{\lambda_S^*}||p_{\lambda_0}] \tag{53}$$

$$= \mathbb{E}_{q_{\lambda_S^*}}\left[-\sum_{i=1}^{n}\sum_{j=1}^{k}y_{i,j}\log\frac{\exp(\phi(x_i)^\top w_j)}{\sum_{l=1}^{k}\exp(\phi(x)^\top w_l)}\right] + \beta_\mathcal{D}D_{\mathrm{KL}}[q_{\lambda_S^*}(W)||p_{\lambda_0}(W)] \tag{54}$$

$$= -\sum_{i=1}^{n}\sum_{j=1}^{k}y_{i,j}\mathbb{E}_{q_{\lambda_S^*}}\left[\log\frac{\exp(\phi(x_i)^\top w_j)}{\sum_{l=1}^{k}\exp(\phi(x)^\top w_l)}\right] + \beta_\mathcal{D}D_{\mathrm{KL}}[q_{\lambda_S^*}(W)||p_{\lambda_0}(W)]. \tag{55}$$

Next, in order to compute approximate expectation $\mathbb{E}_{\mathbf{z}\sim\mathcal{N}(\bar{\mathbf{m}}^*,\bar{\Sigma}^*)}\left[\log\frac{\exp z_j}{\sum_{i=1}^k\exp z_i}\right]$, we first change the form as follows:

$$\mathbb{E}_{\mathbf{z}\sim\mathcal{N}(\bar{\mathbf{m}}^*,\bar{\Sigma}^*)}\left[\log\frac{\exp(z_j)}{\sum_{i=1}^k\exp z_i}\right] = \int\log\frac{\exp z_j}{\sum_{i=1}^k\exp z_i}\mathcal{N}(\mathbf{z}|\bar{\mathbf{m}}^*,\bar{\Sigma}^*)\mathrm{d}\mathbf{z} \tag{56}$$

$$= \int\log\frac{1}{1+\sum_{i\neq j}\exp(-(z_j-z_i))}\mathcal{N}(\mathbf{z}|\bar{\mathbf{m}}^*,\bar{\Sigma}^*)\mathrm{d}\mathbf{z} \tag{57}$$

$$= \int\log(1+\sum_{i\neq j}\exp(-(z_j-z_i)))^{-1}\mathcal{N}(\mathbf{z}|\bar{\mathbf{m}}^*,\bar{\Sigma}^*)\mathrm{d}\mathbf{z} \tag{58}$$

$$= \int\log(2-K+\sum_{i\neq j}(1+\exp(-(z_j-z_i))))^{-1}\mathcal{N}(\mathbf{z}|\bar{\mathbf{m}}^*,\bar{\Sigma}^*)\mathrm{d}\mathbf{z} \tag{59}$$

$$= \int\log(2-K+\sum_{i\neq j}\frac{1}{\sigma(z_j-z_i)})^{-1}\mathcal{N}(\mathbf{z}|\bar{\mathbf{m}}^*,\bar{\Sigma}^*)\mathrm{d}\mathbf{z}, \tag{60}$$

where $\sigma(\cdot)$ is the sigmoid function. Then we utilize mean-field approximation (Lu et al., 2020) to the $z_i$s to approximately compute the Eq. 24:

$$\mathbb{E}_{\mathbf{z}\sim\mathcal{N}(\mathbf{m}^*,\bar{\Sigma}^*)}\left[\log\frac{\exp(z_j)}{\sum_{i=1}^t\exp(z_i)}\right] \approx \log\left(2-k+\sum_{i\neq j}\frac{1}{\mathbb{E}_{(z_j,z_i)\sim\mathcal{N}(\bar{m}_{j,i}^*,\bar{\Sigma}_{j,i}^*)}[\sigma(z_j-z_i)]}\right)^{-1} \tag{61}$$

$$\approx \log\left(2-K+\sum_{i\neq j}\frac{1}{\sigma\left(\frac{\bar{m}_j^*-\bar{m}_i^*}{\sqrt{1+\alpha\bar{\Sigma}_j^*}}\right)}\right)^{-1} \tag{62}$$

$$= \log\frac{1}{1+\sum_{i\neq j}\exp(-\frac{\bar{m}_j^*-\bar{m}_i^*}{\sqrt{1+\alpha\bar{\Sigma}_j^*}})} \tag{63}$$

$$= \left[\log\text{softmax}\left(\frac{\bar{\mathbf{m}}^*}{\sqrt{1+\alpha\bar{\Sigma}_j^*}}\right)\right]_j \tag{64}$$

$$= \left[\log\text{softmax}\left(\frac{\bar{\mathbf{m}}^*}{\sqrt{1+\alpha\Sigma^*}}\right)\right]_j, \tag{65}$$

where $\alpha = \frac{\pi}{8}$ and $\Sigma^* = \phi(x)^\top V^*\phi(x)$.

## A.4 FULL DERIVATION FOR TRAINING AND INFERENCE

In this section, we present the full derivation for the training and inference. Since both $q_{\lambda_{\mathcal{S}}^*}$ and $p_{\lambda_0}$ are Gaussian distributions, the KL divergence can be expressed as follows:

$$D_{\text{KL}}[q_{\lambda_{\mathcal{S}}^*}||p_{\lambda_0}] = \frac{1}{2}[k\log\frac{|\det(\rho^{-1}I_h)|}{|\det(V^*)|} - kh + k\text{Tr}(\rho I_h^{-1}V^*) + \sum_{j=1}^k(m_j^*)^\top(\rho I_h^{-1})m_j^*] \tag{66}$$

$$= \frac{1}{2}[k\log\frac{|\det(\rho^{-1}I_h)|}{|\det(V^*)|} - kh + k\text{Tr}(\rho V^*) + \rho\|m^*\|^2] \tag{67}$$

$$\overset{c}{=} \frac{1}{2}[-k\log|\det(V^*)| + k\rho\text{Tr}(V^*) + \rho\|m^*\|^2]. \tag{68}$$

Here, we have to reduce the memory requirements for the $\det(V^*)$ and the $\text{Tr}(V^*)$ as they require $\mathcal{O}(h^2)$ memory to compute directly from $V^*$. For the $\det V^*$, we used Weinstein-Aronszajn iden-

tity (Pozrikidis, 2014) which results as follows:

$$\det V^* = \det(\rho I_h + \frac{\gamma}{\beta_{\mathcal{S}}} \Phi^\top \Phi)^{-1} \tag{69}$$

$$= \frac{1}{\det(\rho I_h + \frac{\gamma}{\beta_{\mathcal{S}}} \Phi^\top \Phi)} \tag{70}$$

$$= \frac{1}{\rho^h \det(I_h + \frac{\gamma}{\rho\beta_{\mathcal{S}}} \Phi^\top \Phi)} \tag{71}$$

$$= \frac{1}{\rho^h \det(I_{\hat{n}} + \frac{\gamma}{\rho\beta_{\mathcal{S}}} \Phi\Phi^\top)}. \tag{72}$$

Thus we have:

$$\log \det V^* = -h \log \rho - \log \det \left( I_{\hat{n}} + \frac{\gamma}{\rho\beta_{\mathcal{S}}} \Phi\Phi^\top \right) \tag{73}$$

$$\stackrel{c}{=} -\log \det \left( I_{\hat{n}} + \frac{\gamma}{\rho\beta_{\mathcal{S}}} \Phi\Phi^\top \right). \tag{74}$$

Also we can compute trace as follows:

$$\mathrm{Tr}(V^*) = \mathrm{Tr} \left( \frac{\beta_{\mathcal{S}}}{\gamma} \left( \frac{\gamma}{\rho\beta_{\mathcal{S}}} I_h - \left( \frac{\gamma}{\rho\beta_{\mathcal{S}}} \right)^2 \Phi^\top \left( I_{\hat{n}} + \frac{\gamma}{\rho\beta_{\mathcal{S}}} \Phi\Phi^\top \right)^{-1} \Phi \right) \right) \tag{75}$$

$$= \frac{\beta_{\mathcal{S}}}{\gamma} \left( \frac{\gamma h}{\rho\beta_{\mathcal{S}}} - \left( \frac{\gamma}{\rho\beta_{\mathcal{S}}} \right)^2 \mathrm{Tr} \left( \Phi^\top \left( I_{\hat{n}} + \frac{\gamma}{\rho\beta_{\mathcal{S}}} \Phi\Phi^\top \right)^{-1} \Phi \right) \right) \tag{76}$$

$$= \frac{\beta_{\mathcal{S}}}{\gamma} \left( \frac{\gamma h}{\rho\beta_{\mathcal{S}}} - \left( \frac{\gamma}{\rho\beta_{\mathcal{S}}} \right)^2 \mathrm{Tr} \left( \left( I_{\hat{n}} + \frac{\gamma}{\rho\beta_{\mathcal{S}}} \Phi\Phi^\top \right)^{-1} \Phi\Phi^\top \right) \right) \tag{77}$$

$$\stackrel{c}{=} -\frac{\gamma}{\beta_{\mathcal{S}}\rho^2} \mathrm{Tr} \left( \left( I_{\hat{n}} + \frac{\gamma}{\rho\beta_{\mathcal{S}}} \Phi\Phi^\top \right)^{-1} \Phi\Phi^\top \right). \tag{78}$$

These computations allow us to reduce memory requirements during training from $\mathcal{O}(h^2)$ to $\mathcal{O}(\hat{n}^2)$, which represents a significant reduction when dealing with a high-dimensional feature space $h$.

**Memory Efficient Bayesian Model Averaging** For the variance $V^*$ we computed corresponds to a full covariance matrix, leading to a memory cost of $h^2$. To address this, rather than calculating $V^*$ explicitly, we need a memory-efficient approach for conducting BMA on test points. This can be done easily by calculating $\Sigma^*$ as follows:

$$\Sigma^* = \Phi_{\text{te}} V^* \Phi_{\text{te}}^\top \tag{79}$$

$$= \frac{\beta_{\mathcal{S}}}{\gamma} \left( \frac{\gamma}{\rho\beta_{\mathcal{S}}} \Phi_{\text{te}} \Phi_{\text{te}}^\top - \left( \frac{\gamma}{\rho\beta_{\mathcal{S}}} \right)^2 \Phi_{\text{te}} \Phi^\top \left( I_{\hat{n}} + \frac{\gamma}{\rho\beta_{\mathcal{S}}} \Phi\Phi^\top \right)^{-1} \Phi\Phi_{\text{te}}^\top \right), \tag{80}$$

where $\Phi_{\text{te}} \in \mathbb{R}^{n_{\text{te}} \times h}$ denotes the feature matrix of $n_{\text{te}}$ number of test points. Then by storing $\Phi \in \mathbb{R}^{\hat{n} \times h}$ and $(I_{\hat{n}} + \frac{\gamma}{\rho\beta_{\mathcal{S}}} \Phi\Phi^\top)^{-1} \in \mathbb{R}^{\hat{n} \times \hat{n}}$ instead of $V^*$, we can reduce the memory requirements to $\hat{n}h + \hat{n}^2$, which is much smaller than $h^2$.

## B ALGORITHM FOR TRAINING AND INFERENCE

In this section, we present algorithms for training and inference. In Algorithm 1, the overall training procedures are presented, and note that we utilize the model pool $\mathcal{M}$ to prevent overfitting. We also use the Gaussian likelihood to update the weights contained in the model pool. Additionally, in Algorithm 2, we present computationally and memory-efficient variational inference and BMA methods. Here, we store $\Phi$ and $(I_{\hat{n}} + \frac{\gamma}{\rho\beta_{\mathcal{S}}} \Phi\Phi^\top)^{-1}$ instead of directly computing $V^*$.

---

**Algorithm 1** Training Variational Bayesian Pseudo-Coreset (VBPC).

---

**Require:** Training dataset $\mathcal{D}$, learning rate $\delta$.
**Ensure:** Learned synthetic dataset $\mathcal{S}^*$.
  1: **Initialize:**    Initialize synthetic dataset distribution $\mathcal{S}$ with $\hat{n}$ pairs of $(\hat{x}_i, \hat{y}_i)$.
  2: **Initialize:**    Randomly initialize $P$ different $\theta_i$s and construct a model pool $\mathcal{M}$.
  3: **while** not converged **do**
  4:      Sample training batch $\mathcal{B}$ from training distribution $\mathcal{D}$.
  5:      Uniformly sample $\theta_i$ from model pool $\mathcal{M}$ and construct feature map $\phi$.
  6:      Efficiently compute loss Eq. 27 with Eq. 29, Eq. 30 and Eq. 31.
  7:      Update $\hat{x}_i$s and $\hat{y}_i$s using gradient descent: $\hat{x}_i \leftarrow \hat{x}_i - \delta \nabla_{\hat{x}_i} \tilde{\mathcal{L}}_{\mathcal{D}}$, $\hat{y}_i \leftarrow \hat{y}_i - \delta \nabla_{\hat{y}_i} \tilde{\mathcal{L}}_{\mathcal{D}}$
  8:      Update $\theta_i$ with $\mathcal{S}$ and the Gaussian likelihood.
  9:      Replace $\theta_i$ in $\mathcal{M}$ with updated $\theta_i$.
10:      If $\theta_i \in \mathcal{M}$ has been updated $T$ times, reinitialize $\theta_i$ and replace $\theta_i$ in $\mathcal{M}$.
11: **end while**

---

**Algorithm 2** Variational inference and Bayesian Model Averaging using VBPC.

---

**Require:** Learned synthetic dataset $\mathcal{S}^*$, MODE which is VI or BMA, and test dataset $\mathcal{T}$.
**Ensure:** Variational posterior or Bayesian Model Averaged output prediction.
  1: **Initialize:**    Randomly initialize $\theta$.
  2: **while** not converged **do**
  3:      Update $\theta$ with Gaussian likelihood and $\mathcal{S}^*$.
  4: **end while**
  5: Compute $m_j^*$, $\Phi$, and $(I_{\hat{n}} + \frac{\gamma}{\rho\beta_{\mathcal{S}}}\Phi\Phi^{\top})^{-1}$ with $(\hat{x}_i, \hat{y}_i)$s.
  6: **if** MODE == VI **then**
  7:      Compute $V^*$ with Eq. 21.
  8: **else if** MODE==BMA **then**
  9:      Compute $\Sigma^*$ with Eq. 31.
10:      Compute approximate expected predictive distribution for $\mathcal{T}$ similar to Eq. 25.
11: **end if**

---

## C   Additional Related Works

**Bayesian Pseudo-Coreset**   As discussed in Section 1 and Section 2, the large scale of modern real-world datasets leads to significant computational costs when performing SGMCMC (Welling & Teh, 2011; Ahn et al., 2012; Chen et al., 2014; Ma et al., 2015) or variational inference (Blei et al., 2017; Fiedler & Lucia, 2023) to approximate posterior distributions. To address this issue, previous works, such as Bayesian Coreset (BC; Campbell & Broderick, 2018; 2019; Campbell & Beronov, 2019), have proposed selecting a small subset from the full training dataset so that the posterior distribution built from this subset closely approximates the posterior from the full dataset. However, Manousakas et al. (2020) highlighted that simply selecting a subset of the training data is insufficient to accurately approximate high-dimensional posterior distributions, and introduced BPC for simple logistic regression tasks. Later, Kim et al. (2022) extended BPC to BNNs, using reverse KL divergence, forward KL divergence, and Wasserstein distance as measures for D in Eq. 2 to assess the difference between the full posterior and the BPC posterior. Subsequent works have used contrastive divergence (Tiwary et al., 2024) or calculated divergence in function space (Kim et al., 2023) using Function-space Bayesian Neural Network (FBNN; Rudner et al., 2021; 2022). However, as discussed in Section 1, computational and memory overhead remains an issue when training BPC and during inference using BMA.

**Dataset Distillation**   Similar to but distinct from BPC, dataset distillation (Wang et al., 2018) methods aim to train a pseudo-coreset that preserves the essential information contained in the full training dataset. These methods ensure that the model trained on the pseudo-coreset learns information that allows it to perform similarly to a model trained on the full dataset. This approach enables computationally efficient training of new models using the pseudo-coreset and helps prevent catastrophic forgetting in continual learning scenarios, leading to more stable learning.

To train these dataset distillation methods, a bilevel optimization problem must be solved, requiring the computation of meta-gradients through unrolled inner optimization to find the solution to the outer optimization problem. To address this challenge, various learning methods have been proposed in the dataset distillation field, which can be broadly categorized into three approaches: 1) using surrogate objectives, 2) closed-form approximations, and 3) employing the implicit function theorem.

Examples of works in the first category include Zhao & Bilen (2021), Zhao & Bilen (2023), and Cazenavette et al. (2022), where Zhao & Bilen (2021) uses gradient matching, Zhao & Bilen (2023) focuses on feature distribution alignment, and Cazenavette et al. (2022) employs a trajectory matching objective. Papers in the second category, Nguyen et al. (2020) and Zhou et al. (2022), calculate closed-form solutions by using the Neural Tangent Kernel (Jacot et al., 2018) and Neural Network Gaussian Process Kernel (Lee et al., 2017), respectively. Lastly, Loo et al. (2023), representing the third category, uses the implicit function theorem to compute gradients for unrolled inner optimization, allowing for the updating of the pseudo-coreset.

**Variational Inference** Variational inference (Bishop, 2006; Blundell et al., 2015; Blei et al., 2017), one of the most general methods for approximating most posterior distributions, is a technique that approximates the target posterior distribution using a variational distribution, which has a well-known and manageable form. The parameters of the variational distribution are learned by minimizing the KL divergence between the target posterior distribution and the variational distribution. Although using all the parameters of the variational distribution can enhance its expressiveness, allowing for more accurate approximations, two common approaches are typically employed to address the computational and memory challenges that arise when handling the large scale of BNN weights: 1) mean-field approximation (Blundell et al., 2015; Shen et al., 2024), and 2) computing the posterior distribution for only a subset of the network parameters (Dusenberry et al., 2020; Fiedler & Lucia, 2023; Harrison et al., 2024a). In both of these cases, the parameters of the variational distribution are optimized either directly using gradient descent methods to minimize the KL divergence (Blundell et al., 2015; Dusenberry et al., 2020; Shen et al., 2024), or a closed-form solution is found (Wang & Blei, 2013).

# D    ADDITIONAL DISCUSSION ON VBPC

**Future work direction** Here, we would like to discuss some concerns and challenges we foresee in adopting the Laplace approximation on the softmax likelihood instead of using variational inference with Gaussian likelihood.

Specifically, if we switch from using a Gaussian likelihood to employing a softmax likelihood with Laplace approximation for variational inference, there are two cases to consider: (1) using Laplace approximation on the last-layer weights without any updates, and (2) updating the last-layer weights with some gradient descent steps before applying Laplace approximation.

In the first case—applying Laplace approximation to weights without updating the last layer—two main issues may arise. First, the Laplace approximation assumes that the weights are near a minimum, allowing for the approximation of the first-order term in Taylor expansion as zero. However, this assumption may not hold for untrained weights, leading to significant approximation error. Additionally, the computational burden of calculating the Hessian for Laplace approximation is substantial, and the need to compute gradients through this Hessian during pseudo-coreset updates increases the computational load further.

In the second case—updating the last layer weights with gradient steps before applying Laplace approximation—there's the advantage of reducing Taylor expansion error. However, this approach involves a large computational graph, which can be problematic due to the computational expense typical in bilevel optimization settings. Additionally, the need to compute gradients through the Hessian remains a challenge.

Overall, we believe that solving these issues could lead to new meaningful future work for VBPC.

**Limitations of the Last-Layer Approximation** There might be concerns that considering the posterior distribution of only the last layer weights, rather than the entire parameter set, could limit

the model's ability to capture uncertainty effectively, especially as the model size increases and tasks become more complex. We fully agree that this is a valid concern and would like to provide a discussion based on related findings.

Specifically, Harrison et al. (2024b) provides extensive empirical evidence on the effectiveness of last-layer variational inference. Their experiments span diverse tasks, including regression with UCI datasets, image classification using a Wide ResNet model, and sentiment classification leveraging LLM features from the OPT-175B model. They compared their method with other Bayesian inference approaches such as Dropout, Ensemble methods, and Laplace approximation for the full model. Their results demonstrate that even though last-layer variational inference focuses solely on the final layer weights, it achieves performance comparable to other comprehensive Bayesian inference techniques across various tasks.

These findings indicate that while conducting Bayesian inference on the full set of weights in a neural network could potentially lead to more precise uncertainty estimation, employing last-layer variational inference is still effective in capturing meaningful uncertainty.

We believe that extending VBPC to incorporate full-weight variational inference could be a promising direction for future work, offering the potential to further enhance the method's uncertainty estimation capabilities. We will include this discussion in the final manuscript to provide a balanced perspective and acknowledge possible avenues for improvement.

## E EXPERIMENTAL DETAILS

Our VBPC code implementation is built on the official FRePo (Zhou et al., 2022)[1] codebase. The implementation utilizes the following libraries, all available under the Apache-2.0 license[2]: JAX (Bradbury et al., 2018), Flax (Babuschkin et al., 2020), Optax (Babuschkin et al., 2020), TensorFlow Datasets (Abadi et al., 2015), and Augmax[3]. For the baseline methods, we used the official code implementations provided for each. All experiments, except those on the Tiny-ImageNet (Le & Yang, 2015) dataset, were performed on NVIDIA RTX 3090 GPU machines, while Tiny-ImageNet experiments were conducted on NVIDIA RTX A6000 GPUs.

### E.1 DATASETS

**Datasets for the Bayesian Model Averaging comparison**   For the BMA comparison experiments, we utilize 5 different datasets: 1) MNIST (LeCun et al., 1998), 2) Fashion-MNIST (Xiao et al., 2017), 3) CIFAR10 (Krizhevsky, 2009), 4) CIFAR100 (Krizhevsky, 2009), and 5) Tiny-ImageNet (Le & Yang, 2015).

- **MNIST**: The MNIST dataset[4] contains 10 classes of handwritten digits with 60,000 training images and 10,000 test images, each with dimensions of $28 \times 28 \times 1$. All images were normalized using a mean of $[0.1307]$ and a standard deviation of $[0.3081]$.

- **Fashion-MNIST**: The Fashion-MNIST dataset[5] consists of 10 classes of fashion article images, with 60,000 training images and 10,000 test images, each with dimensions of $28 \times 28 \times 1$. Images were normalized using a mean of $[0.2861]$ and a standard deviation of $[0.3530]$.

- **CIFAR-10/100**: The CIFAR-10/100 dataset[6] contains 10/100 classes, with 50,000 training images and 10,000 test images sourced from the 80 Million Tiny Images dataset (Torralba et al., 2008). Each image has dimensions of $32 \times 32 \times 3$. For CIFAR-10, images were normalized with a mean of $[0.4914, 0.4822, 0.4465]$ and a standard deviation of $[0.2470, 0.2435, 0.2616]$, while CIFAR-100 images used a mean of $[0.5071, 0.4866, 0.4409]$ and a standard deviation of $[0.2673, 0.2564, 0.2762]$.

---

[1] https://github.com/yongchaoz/FRePo
[2] https://www.apache.org/licenses/LICENSE-2.0
[3] https://github.com/khdlr/augmax
[4] https://yann.lecun.com/exdb/mnist/
[5] https://github.com/zalandoresearch/fashion-mnist
[6] https://www.cs.toronto.edu/~kriz/cifar.html

- **Tiny-ImageNet**: The Tiny-ImageNet dataset[7] contains 200 classes, with 100,000 training images and 10,000 test images. Each image has dimensions of $64 \times 64 \times 3$. Images were normalized using a mean of $[0.4759, 0.4481, 0.3926]$ and a standard deviation of $[0.2763, 0.2687, 0.2813]$.

**Datasets for the Out of Distribution scenarios** For the distribution shift and OOD scenarios, we use CIFAR10-C (Hendrycks & Dietterich, 2019), which includes seven corruption types with five severity for each corruption type: 1) Gaussian Blur, 2) JPEG Compression, 3) Snow, 4) Zoom Blur, 5) Pixelate, 6) Defocus Blur, and 7) Motion Blur.

- **CIFAR10-C**: The CIFAR10-C dataset[8] consists of 10 classes, with 50,000 test images for each corruption type. It applies various corruptions to 10,000 test images from CIFAR10, with five levels of severity, each containing 10,000 images. The images are normalized using the same mean $[0.4914, 0.4822, 0.4465]$ and standard deviation $[0.2470, 0.2435, 0.2616]$ as the CIFAR10 dataset.

### E.2 MODEL ARCHITECTURE

**Model architecture utilized for the Bayesian Model Averaging and Out of Distribution tasks** Following previous works (Kim et al., 2022; 2023; Tiwary et al., 2024; Zhou et al., 2022), we used a convolutional neural network (CNN) for the **Bayesian Model Averaging comparison** experiment and the **Out of Distribution** experiment. This model is composed of several blocks, each consisting of a $3 \times 3$ convolution kernel, pre-defined normalization layer, Rectified Linear Unit (ReLU; Agarap, 2018) activation, and a $2 \times 2$ average pooling layer with a stride of 2. For datasets with resolutions of $28 \times 28 \times 1$ and $32 \times 32 \times 3$, we used 3 blocks, and for datasets with a resolution of $64 \times 64 \times 3$, we used 4 blocks. Following Zhou et al. (2022), we increase twice the number of filters when the feature dimension was halved, to prevent the feature dimensions from becoming too small. Additionally, by default, we used the Batch Normalization (Ioffe, 2015) layer for normalization unless stated otherwise. For initializing model weights, we conducted experiments using the Lecun Initialization (Cun et al., 1998) method, which is the default initialization method of the Flax library. This configuration was applied both during the model pool in the VBPC training process and in the evaluation phase.

**Model architecture utilized for the Architecture Generalization task** For the **Architecture generalization** experiments, we incorporate three additional normalization layers and three additional model architectures. The normalization layers include Instance Normalization (Ulyanov, 2016), Identity map, and Group Normalization (Wu & He, 2018). For the model architectures, we include AlexNet (Krizhevsky et al., 2012), VGG (Simonyan & Zisserman, 2014), and ResNet (He et al., 2016). Initially, we evaluate all baselines by replacing Batch Normalization in the convolution layers with the three alternative normalization methods, referring to these as CNN-IN, CNN-NN, and CNN-GN, respectively. Next, we use the three additional model architectures for evaluation. Since AlexNet does not have normalization layers in its original design, we retain this structure and refer to it as AlexNet-NN. For VGG and ResNet, we use VGG11 with Group Normalization and ResNet18 with Batch Normalization. These models are denoted as VGG11-GN and ResNet18-BN.

### E.3 PSEUDO-CORESET INITIALIZATION, PREPROCESSING, AND AUGMENTATION

**Initialization** Building on prior works (Kim et al., 2022; 2023; Tiwary et al., 2024; Zhou et al., 2022), we initialize the pseudo-coreset by randomly sampling images and labels from the original training dataset using a fixed sampling seed. For the labels, following Zhou et al. (2022), we initialize them with scaled, mean-centered one-hot vectors corresponding to each image, where the scaling factor is determined by the number of classes $k$, specifically $\frac{1}{\sqrt{k/10}}$. Here, we train both images and labels during training.

---

[7]https://tiny-imagenet.herokuapp.com/
[8]https://github.com/hendrycks/robustness?tab=readme-ov-file

**Data preprocessing and Augmentation**   Following previous works (Kim et al., 2022; Tiwary et al., 2024; Zhou et al., 2022), we perform standard preprocessing on each dataset, with the addition of ZCA (Kessy et al., 2018) transformations for all datasets with 3 channels. Consistent with Zhou et al. (2022), we apply a regularization strength of $\lambda = 0.1$ across all datasets. Similar to previous works (Kim et al., 2022; 2023; Tiwary et al., 2024; Zhou et al., 2022), we apply the following augmentations to the MNIST and Fashion-MNIST datasets: 'Gaussian noise', 'brightness', 'crop', 'rotate', 'translate', and 'cutout'. For all other datasets, we use 'flip', 'Gaussian noise', 'color', 'crop', 'rotate', 'translate', and 'cutout' augmentations. These augmentations are applied both during the training of the Bayesian pseudo-coreset and during evaluation with them.

### E.4   Hyperparamters

**Hyperparameters during training VBPC**   Following previous works (Kim et al., 2022; 2023; Tiwary et al., 2024), we select 1, 10, or 50 images per class for all datasets when training VBPC for evaluation. For $\beta_{\mathcal{S}}$, we use $\hat{n}$, which corresponds to the number of pseudo-coresets in each experiment. This setup is designed to control the gradient magnitude by averaging, rather than summing, the expected likelihood, while maintaining the influence of the KL divergence for stable training. For $\beta_{\mathcal{D}}$, we used 1e-8 as the default value, and when adjusted, it was selected from the range [1e-6, 1e-7, 1e-8] across all experiments. For $\rho$ and $\gamma$, we set the default values to $\rho = 1.0$ and $\gamma = 100.0$ for the ipc 1 and ipc 10 settings, and $\rho = 10.0$ and $\gamma = 100.0$ for the ipc 50 settings. Except for the CIFAR100 ipc 10 setting where we utilize $\rho = 10.0$ and $\gamma = 100.0$ for the default setting. When tuning these parameters, we adjusted them on a log scale in steps of 10 within the range of [-5, 5]. Following the default settings in Zhou et al. (2022), we set the number of models stored in the model pool, $P$, to 10. Additionally, as per Zhou et al. (2022), we set the number of training steps, $T$, for each model in the model pool to 100. For the model pool optimizer, we used the Adam (Kingma, 2014) optimizer with a fixed learning rate of 0.0003 across all experiments. For the pseudo-coreset optimizer, we also used the Adam optimizer by default, with a cosine learning rate schedule starting at 0.003 for both images and labels. Lastly, we used a batch size of 1024 and trained for 0.5 million steps to ensure sufficient convergence.

**Hyperparameters during variational inference and Bayesian Model Averaging**   For all experiments, the hyperparameters $\gamma$, $\rho$, and $\beta_{\mathcal{S}}$ used during evaluation were the same as those used for pseudo-coreset training in the corresponding experiment. The optimizer used for training the models during evaluation was the Adam optimizer with a constant learning rate of 0.0003. The number of training steps for each model was as follows: for MNIST and Fashion-MNIST, 100 steps for 1 ipc, 500 steps for 10 ipc, and 1000 steps for 50 ipc. For CIFAR10, 200 steps for 1 ipc, 2000 steps for 10 ipc, and 5000 steps for 50 ipc. For CIFAR100, 2000 steps for both 1 ipc and 10 ipc, and 5000 steps for 50 ipc. Lastly, for Tiny-ImageNet, 1000 steps were used for 1 ipc and 2000 steps for 10 ipc.

## F   Additional Experiment

### F.1   Full experimental results on Bayesian Model Averaging comparison

Here, we report the full experimental results for Section 5.1. We report results for FRePo and RCIG across the entire benchmark dataset and varying IPC settings additional to Table 1. Table 6 clearly demonstrates that VBPC surpasses other BPC baselines across all benchmark datasets and IPC settings in terms of ACC and NLL. Notably, VBPC achieves significantly better NLL, with large margins, while requiring only a single forward pass to conduct BMA. Although VBPC is designed to learn pseudo-coresets that approximate the variational distribution derived from the training dataset, it outperforms dataset distillation baselines, which primarily focus on achieving high ACC, in nearly all tasks, except for CIFAR100 with 1 IPC and Tiny-ImageNet. These results empirically validate that the variational distribution trained by VBPC effectively captures epistemic uncertainty with a small amount of synthetic data, while maintaining high performance.

**Comparison with dataset distillation baselines**   In Section 5.1, the performance was evaluated based on the training and evaluation methods proposed by each baseline's original papers. However, one might question whether the significant performance of VBPC is due to the trained pseudo-coreset itself or the VI method. To address this, and to validate that the significant performance of

**Table 6:** Comparison of the VBPC with BPC and additional dataset distillation baselines for the benchmark datasets. We report ACC and NLL for the BPC baselines, and ACC for the dataset distillation baselines. **Boldfaced blue color** indicates when the performance of the dataset distillation baseline surpasses that of VBPC.

| Dataset | ipc | FRePo ACC(↑) | RCIG ACC(↑) | BPC-rKL ACC(↑) | NLL(↓) | BPC-fKL ACC(↑) | NLL(↓) | FBPC ACC(↑) | NLL(↓) | BPC-CD ACC(↑) | NLL(↓) | VBPC (Ours) ACC(↑) | NLL(↓) |
|---|---|---|---|---|---|---|---|---|---|---|---|---|---|
| MNIST | 1 | 93.0±0.4 | 94.7±0.5 | 74.8±1.2 | 1.90±0.01 | 83.0±2.2 | 1.87±0.03 | 92.5±0.1 | 1.68±0.01 | 93.4±0.1 | 1.53±0.01 | **96.7**±0.4 | **0.11**±0.02 |
| | 10 | 98.6±0.1 | 98.9±0.0 | 95.3±0.2 | 1.53±0.01 | 92.1±0.4 | 1.51±0.02 | 97.1±0.2 | 1.31±0.01 | 97.7±0.2 | 1.57±0.02 | **99.1**±0.1 | **0.03**±0.01 |
| | 50 | 99.2±0.1 | 99.2±0.0 | 94.2±0.3 | 1.36±0.02 | 93.6±1.8 | 1.36±0.02 | 98.6±0.1 | 1.39±0.02 | 98.9±0.2 | 1.36±0.02 | **99.4**±0.1 | **0.02**±0.01 |
| FMNIST | 1 | 75.6±0.3 | 79.8±1.1 | 70.5±1.1 | 2.47±0.02 | 72.5±2.5 | 2.30±0.02 | 74.7±1.4 | 1.81±0.03 | 77.3±0.5 | 1.90±0.03 | **82.9**±0.6 | **0.47**±0.03 |
| | 10 | 86.2±0.2 | 88.5±0.2 | 78.8±0.2 | 1.64±0.01 | 83.3±0.6 | 1.54±0.03 | 85.2±0.1 | 1.61±0.02 | 88.4±0.2 | 1.56±0.01 | **89.4**±0.2 | **0.30**±0.01 |
| | 50 | 89.6±0.1 | 90.2±0.2 | 77.0±0.6 | 1.48±0.02 | 74.8±0.5 | 1.47±0.02 | 76.7±0.4 | 1.46±0.02 | 89.5±0.1 | 1.30±0.02 | **91.0**±0.2 | **0.25**±0.01 |
| CIFAR10 | 1 | 46.8±0.7 | 53.9±1.0 | 21.6±0.8 | 2.57±0.01 | 29.3±1.1 | 2.10±0.03 | 35.5±0.3 | 3.79±0.04 | 46.9±0.2 | 1.87±0.02 | **55.1**±0.3 | **1.34**±0.08 |
| | 10 | 65.5±0.4 | 69.1±0.4 | 37.9±1.5 | 2.13±0.02 | 49.9±1.4 | 1.73±0.01 | 62.3±0.1 | 1.31±0.02 | 56.4±0.7 | 1.72±0.03 | **69.8**±0.7 | **0.89**±0.02 |
| | 50 | 71.7±0.2 | 73.5±0.3 | 37.5±1.3 | 1.93±0.03 | 42.3±2.9 | 1.54±0.01 | 71.2±0.2 | 1.03±0.05 | 71.9±0.2 | 1.57±0.00 | **76.7**±0.5 | **0.71**±0.03 |
| CIFAR100 | 1 | 28.7±0.1 | **39.3**±0.4 | 3.6±0.1 | 4.69±0.02 | 14.7±0.2 | 4.20±0.10 | 21.0±0.8 | 3.76±0.11 | 24.0±0.1 | 4.01±0.02 | **38.4**±0.2 | **2.47**±0.04 |
| | 10 | 42.5±0.2 | 44.1±0.4 | 23.6±0.7 | 3.99±0.03 | 28.1±0.6 | 3.53±0.05 | 39.7±0.3 | 2.67±0.02 | 28.4±0.2 | 3.14±0.02 | **49.4**±0.1 | **2.07**±0.02 |
| | 50 | 44.3±0.2 | 46.7±0.3 | 30.8±0.5 | 3.57±0.17 | 37.1±0.3 | 3.28±0.24 | 44.5±0.4 | 2.63±0.01 | 39.6±0.2 | 3.02±0.01 | **52.4**±0.4 | **2.02**±0.02 |
| Tiny-ImageNet | 1 | 15.4±0.3 | **25.6**±0.3 | 3.2±0.1 | 5.91±0.07 | 4.0±0.1 | 5.63±0.03 | 10.1±0.7 | 4.69±0.05 | 8.4±0.1 | 4.72±0.01 | **23.1**±0.2 | **3.65**±0.01 |
| | 10 | 25.4±0.2 | **29.4**±0.2 | 9.8±0.6 | 5.26±0.05 | 11.4±0.5 | 5.08±0.05 | 19.4±0.5 | 4.14±0.02 | 17.8±0.4 | 3.64±0.05 | **25.8**±0.3 | **3.45**±0.02 |

**Table 7:** Ablation experiment on BMA method. Here, we conduct our variational inference method utilizing datasets trained with other baselines.

| Dataset | ipc | FRePo VI ACC(↑) | NLL(↓) | RCIG VI ACC(↑) | NLL(↓) | BPC-rKL VI ACC(↑) | NLL(↓) | BPC-fKL VI ACC(↑) | NLL(↓) | FBPC VI ACC(↑) | NLL(↓) | BPC-CD VI ACC(↑) | NLL(↓) |
|---|---|---|---|---|---|---|---|---|---|---|---|---|---|
| CIFAR10 | 1 | 28.2±0.9 | 2.22±0.02 | 27.8±0.7 | 2.20±0.01 | 10.1±0.1 | 2.30±0.01 | 10.1±0.1 | 2.32±0.01 | 10.0±0.1 | 2.37±0.02 | 10.4±0.8 | 2.35±0.02 |
| | 10 | 55.7±0.5 | 2.07±0.02 | 55.6±1.5 | 2.05±0.02 | 12.0±0.5 | 2.25±0.02 | 20.1±1.9 | 2.21±0.01 | 10.0±0.0 | 2.37±0.02 | 10.5±0.7 | 2.32±0.01 |
| CIFAR100 | 1 | 19.9±0.4 | 4.55±0.02 | 2.1±0.1 | 5.02±0.05 | 1.2±0.1 | 4.60±0.01 | 1.4±0.3 | 4.60±0.01 | 1.2±0.2 | 4.60±0.01 | 1.2±0.2 | 4.60±0.01 |
| | 10 | 34.8±0.4 | 4.50±0.01 | 2.5±0.4 | 5.45±0.12 | 2.6±0.2 | 4.59±0.02 | 4.0±0.2 | 4.59±0.02 | 1.6±0.3 | 4.59±0.02 | 11.6±0.4 | 4.54±0.02 |

VBPC is not solely attributable to the VI method, we collected the pseudo-coresets trained on all baselines used in Section 5.1 for the CIFAR10 and CIFAR100 datasets in the 1ipc and 10ipc settings. We then applied our proposed VI method to these baseline pseudo-coresets to measure their BMA performance and compared the results with those reported in Table 6. Results in Table 7 and Table 6 clearly show that the performance significantly drops for all baselines compared to their original performance. This validates that the performance is not solely attributable to the VI method, and demonstrates that VBPC successfully learns to approximate the variational distribution effectively.

## F.2 Additional Experiment Results on Large Dataset and Continual Learning

To further highlight the ability of VBPC to handle tasks that pose challenges for other BPC baselines, we conduct additional experiments on more large datasets and the continual learning setting.

**Large Datasets** First, to show that our method is uniquely scalable to large datasets compared to other BPC methods, we conducted additional experiments on the ImageNetWoof (128x128x3) dataset (Howard, 2020) and the ImageNet1k (64x64x3) dataset(Russakovsky et al., 2015). Additionally, we included an experiment in a continual learning scenario to validate that our method performs better in practical scenarios.

We conducted experiments on the ImageWoof (128x128x3) dataset with ipc 1 and ipc 10 settings, as well as the resized ImageNet1k (64x64x3) dataset with ipc 1 and ipc 2 settings, to demonstrate the scalability of our method to high-resolution images and larger datasets. Unlike existing BPC baselines, which encountered memory issues and failed to train due to out-of-memory errors on an RTX 3090 GPU as the image resolution and number of classes increased, our method successfully completed training. Table 8 clearly shows that VBPC significantly outperforms other baselines with a large margin for both the ImageWoof and resized ImageNet1k datasets.

**Continual Learning** Next, we validated the practical effectiveness of our method through continual learning experiments using pseudo-coreset images learned by each method. We followed the

**Table 8:** Experiments on the scalability utilizing ImageWoof and resized ImageNet datasets. Here '-' indicates the training fails due to the out-of-memory problems.

| Method | ImageWoof ipc 1 | | ImageWoof ipc 10 | | ImageNet ipc 1 | | ImageNet ipc 2 | |
|---|---|---|---|---|---|---|---|---|
| | ACC(↑) | NLL(↓) | ACC(↑) | NLL(↓) | ACC(↑) | NLL(↓) | ACC(↑) | NLL(↓) |
| **Random** | 14.2±0.9 | 3.84±0.25 | 27.0±1.9 | 2.83±0.33 | 1.1±0.1 | 8.32±0.05 | 1.4±0.1 | 8.10±0.05 |
| **BPC-CD** | 18.5±0.1 | 2.76±0.05 | - | - | - | - | - | - |
| **FBPC** | 14.8±0.1 | 3.73±0.02 | 28.1±0.3 | 2.69±0.09 | - | - | - | - |
| **BPC-fKL** | 14.9±0.9 | 3.74±0.23 | 25.0±0.8 | 2.90±0.27 | - | - | - | - |
| **BPC-rKL** | 12.0±0.5 | 6.07±0.31 | - | - | - | - | - | - |
| **VBPC** | **31.2**±0.1 | **2.13**±0.04 | **39.0**±0.1 | **1.84**±0.1 | **10.1**±0.1 | **5.33**±0.04 | **11.5**±0.2 | **5.25**±0.05 |

**Table 9:** Experiments on the continual learning setting. Here, we utilize the CIFAR100 dataset with ipc 20 setting. We assume 5 steps during training and each step contains data from new 20 classes in the CIFAR100 dataset. Here we only report accuracy due to the variant of the number of classes during the steps.

| Number of Classes | 20 | 40 | 60 | 80 | 100 |
|---|---|---|---|---|---|
| **BPC-CD** | 52.5±2.4 | 40.4±1.3 | 35.2±0.8 | 33.4±0.5 | 29.4±0.2 |
| **FBPC** | 61.4±1.8 | 53.2±1.5 | 48.8±0.7 | 43.9±0.4 | 41.2±0.3 |
| **BPC-fKL** | 51.8±2.2 | 39.8±1.1 | 35.5±0.7 | 33.1±0.5 | 29.5±0.3 |
| **BPC-rKL** | 48.2±2.7 | 35.5±1.8 | 32.0±1.0 | 29.8±0.6 | 25.5±0.3 |
| **VBPC** | **75.3**±2.0 | **65.8**±1.5 | **57.1**±0.9 | **53.3**±0.5 | **50.3**±0.2 |

continual learning setup described in Zhou et al. (2022); Zhao & Bilen (2021), where class-balanced training examples are greedily stored in memory, and the model is trained from scratch using only the latest memory. Specifically, we performed a 5-step class incremental learning experiment on CIFAR100 with an ipc 20 setting, following the class splits proposed in Zhou et al. (2022); Zhao & Bilen (2021). Table 9 demonstrates that VBPC consistently outperforms other baselines across all steps, confirming its superior practicality and effectiveness in real-world continual learning scenarios.

## F.3 Additional experiments on Out-of-Distribution Data

To further validate the effectiveness of VBPC, We have conducted additional Out-of-Distribution (OOD) detection experiments and reported the results. The metrics we evaluate include AUROC, AUPR-In, and AUPR-Out, where higher values indicate better performance. We used models trained with the CIFAR10 IPC 10 setting and evaluated them on CIFAR100, TinyImageNet, and SVHN (Netzer et al., 2011) datasets as OOD datasets.

The results, presented in Table 10, demonstrate that the pseudo-coreset learned by VBPC performs robustly in OOD detection scenarios. These findings, combined with the corruption experiments in the main paper, validate the effectiveness and robustness of VBPC under diverse and challenging evaluation conditions.

## F.4 Analysis on Computational Costs and Training Time

In this section, we performed analyses focusing on two aspects of computational cost.

**Cost of training the pseudo-coreset**   As mentioned in the Section 1, conventional BPC methods relying on SGMCMC require the creation of expert trajectories, which are training trajectories derived from the full dataset. Each dataset typically involves training with 10 different random seeds for these trajectories, making this step computationally expensive. Since all BPC baselines share and utilize these precomputed trajectories, their associated computational cost can be considered a shared overhead.

To isolate the computational cost of training the pseudo-coreset itself, we measured the wall-clock time required for pseudo-coreset optimization by each method. The results of this comparison are

**Table 10:** AUROC, AUPR-In, and AUPR-Out results for the OOD detection task with a model trained with the learned pseudo-coresets. Note that we used the same model structure which is utilized when training pseudo-coresets.

| Dataset | Model | AUROC(↑) | AUPR-In(↑) | AUPR-Out(↑) |
|---|---|---|---|---|
| TinyImageNet | BPC-CD | 49.09 | 52.79 | 45.88 |
| | BPC-fKL | 48.95 | 51.72 | 47.00 |
| | BPC-rKL | 48.34 | 52.71 | 44.49 |
| | FBPC | 45.39 | 49.70 | 43.14 |
| | VBPC | **52.85** | **56.22** | **49.64** |
| SVHN | BPC-CD | 55.09 | 35.64 | 73.88 |
| | BPC-fKL | 54.26 | 34.78 | 75.47 |
| | BPC-rKL | 42.61 | 28.29 | 67.15 |
| | FBPC | 41.34 | 30.12 | 62.18 |
| | VBPC | **68.50** | **48.49** | **82.91** |

**Table 11:** Wall clock time results for training pseudo-coresets with each BPC method using CIFAR10 ipc 10 settings. We used RTX3090 GPU to measure the exact training time. Here, all methods except for VBPC share the training time for expert trajectories.

| Method | BPC-CD | BPC-rKL | FBPC | BPC-fKL | VBPC |
|---|---|---|---|---|---|
| **Times** (hr) | 5+8.5 | 5+9 | 5+10.5 | 5+12 | 5.5 |

summarized in Table 11, providing insights into how VBPC reduces training costs compared to other baselines.

**Cost of inference** When performing inference, VBPC requires training only a single model, whereas other BPC baselines rely on multiple SGMCMC samples. Each sample incurs significant training and inference costs, which grow linearly with the number of samples.

To quantify this difference, we measured the wall-clock time for inference across methods, with results presented in Table 12. These results highlight how VBPC achieves superior efficiency during inference by avoiding the high computational costs associated with sampling-based approaches.

These analyses demonstrate VBPC's ability to perform Bayesian inference efficiently, both in terms of pseudo-coreset training and inference, and further reinforce the computational advantages of our method.

### F.5   ABLATION ON RANDOM INITIALIZATION

Since our method initializes the pseudo-coreset by randomly sampling images and labels from the original training dataset, following previous works (Kim et al., 2022; 2023; Tiwary et al., 2024; Zhou et al., 2022), we conducted an ablation experiment using random initialization for the pseudo-coreset. In this experiment, we first randomly initialized the pseudo-coreset by sampling pixel values from a uniform distribution $\text{Unif}[0, 1]$. We then trained the images after normalizing them with the predefined mean and variance for each dataset reported in Appendix E.1. We conducted this ablation experiment on the CIFAR10/100 1 ipc and 10 ipc settings. Fig. 2 and Fig. 3 clearly illustrate that VBPC can effectively learn semantic information even when initialized randomly. Specifically, in the CIFAR10 ipc 1 case shown in the top figures of both Fig. 2 and Fig. 3, the images after training appear similar, whether they were initialized randomly or sampled from the training dataset. Also Table 13 shows that randomly initialized VBPC shows comparable performance compared to the VBPC.

### F.6   ABLATION ON PSEUDO-CORESET OPTIMIZER

Since we use the Adam optimizer for training VBPC, which differs from the default choice in previous work (Zhou et al., 2022), we conducted an ablation experiment on the optimizer. Following

**Table 12:** Wall clock time results for inference using learned pseudo-coresets. We measure the inference time for evaluating all the test data from the CIFAR10 test dataset. After finishing training the pseudo-coresets, the inference cost for the all baselines are same because they only need SGM-CMC and BMA with same number of datasets and weight samples.

| Method | BPC-CD | BPC-rKL | FBPC | BPC-fKL | VBPC |
|---|---|---|---|---|---|
| **Times** (s) | 165 | 165 | 165 | 165 | 20 |

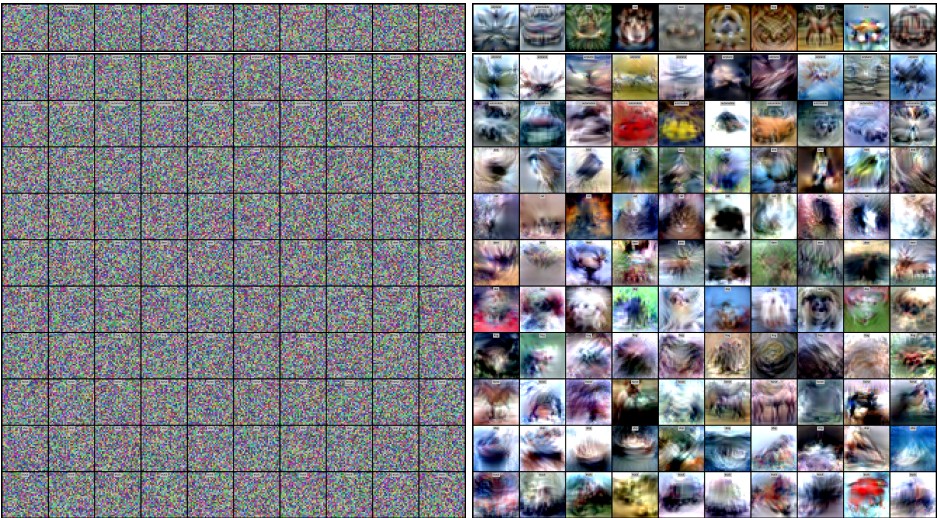

**Figure 2:** Learned VBPC images from the random initialization for the CIFAR10 ipc 1 (above) and ipc 10 (below) cases. The left figure shows the random images sampled from the uniform distribution and the right figure shows the trained VBPC images starting from the left images. Training from random initialization successfully learns semantic information from the full dataset.

Zhou et al. (2022), we used the LAMB (You et al., 2019) optimizer with a cosine learning rate schedule for this ablation. We conduct this ablation experiment on the CIFAR10 1 ipc and 10 ipc settings. As seen in Fig. 4, although there are minor differences, the images trained with the LAMB and Adam optimizers are largely similar when starting from the same pseudo-coreset initial images. Additionally, Table 14 demonstrates that our method effectively learns pseudo-coreset with varying optimizers, closely approximating the variational distribution of the full training dataset.

### F.7 ABLATION ON MODEL POOL MAXIMUM UPDATE STEPS

As mentioned in Appendix E.4, we set $T = 100$ as the maximum update step for the weights in the model pool $\mathcal{M}$ across all experiments. The model pool was introduced to address VBPC's overfitting issue, as the weights in the model pool are trained for $T$ steps, leading to a variety of feature maps. This prevents VBPC from learning based on a single feature map. To investigate the effect of $T$, we plan to conduct an ablation study to examine how changes in $T$ impact image quality and performance. We conducted an ablation experiment on the CIFAR100 ipc 10 task with

**Table 13:** Comparison between the random initialization and initialization with randomly sampled images. Random Initialization denotes the VBPC learned starting from the uniform random initialization. Here, we report ACC and NLL for both initializations.

| Dataset | ipc | Random Initialization ACC(↑) | NLL(↓) | VBPC ACC(↑) | NLL(↓) | Dataset | ipc | Random Initialization ACC(↑) | NLL(↓) | VBPC ACC(↑) | NLL(↓) |
|---|---|---|---|---|---|---|---|---|---|---|---|
| **CIFAR10** | 1 | 54.2±0.5 | 1.37±0.02 | 55.1±0.3 | 1.34±0.08 | **CIFAR100** | 1 | 37.5±0.4 | 2.51±0.06 | 38.4±0.2 | 2.47±0.04 |
| | 10 | 68.9±0.4 | 0.98±0.01 | 69.8±0.7 | 0.89±0.02 | | 10 | 48.4±0.4 | 2.20±0.03 | 49.4±0.1 | 2.07±0.02 |

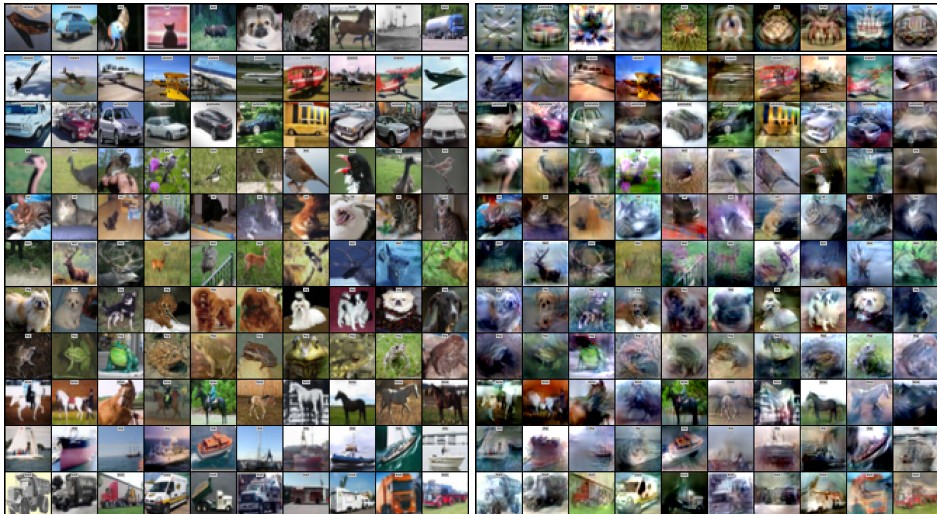

**Figure 3:** Learned VBPC images from the randomly sampled image from the original training dataset for the CIFAR10 ipc 1 (above) and ipc 10 (below) cases. The left figure shows the initial images sampled from the original dataset and the right figure show the final learned VBPC starting from the left images.

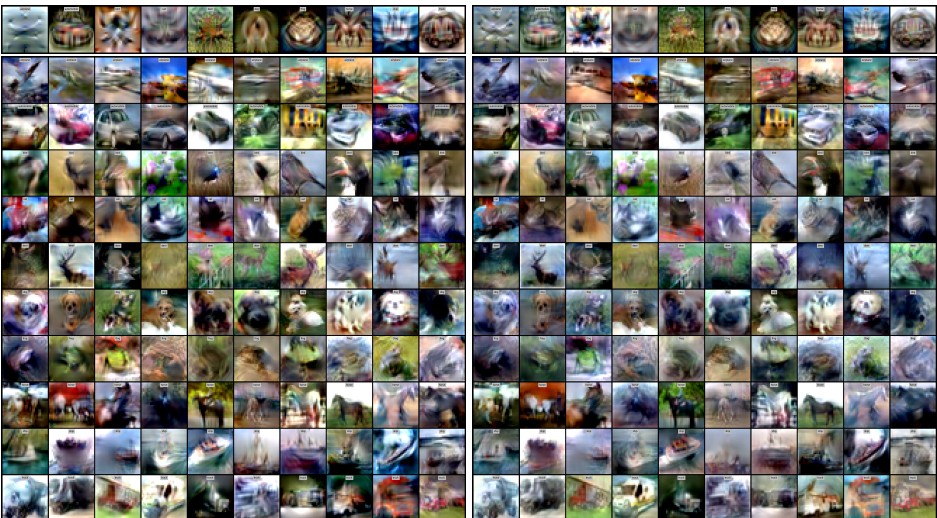

**Figure 4:** Visualization of learned VBPC images utilizing different optimizers for the CIFAR10 ipc 1 (above) and ipc 10 (below). The left figure shows the learned VBPC images with LAMB optimizer and the right figure shows the learned VBPC images with Adam optimizer.

$T = 200$ and $T = 400$. As shown in Fig. 5, the images learned with different maximum update steps appear visually similar. However, Table 15 quantitatively shows that excessive updates to the model pool weights can reduce feature diversity, potentially leading to a decline in performance for unseen feature maps.

### F.8 ABLATION ON LABEL LEARNING

Following the previous works (Kim et al., 2022; 2023; Tiwary et al., 2024; Zhou et al., 2022), we learned the labels when training the pseudo-coreset. This can be crucial and effective for learning a more informative BPC, as the mean of the pseudo-coreset variational distribution depends on the label. This dependency also impacts the loss function used in the outer optimization process. Fig. 6 shows that without label learning, trained VBPC images significantly lost the semantic informa-

**Table 14:** Comparison between the VBPC learned with LAMB optimizer and Adam optimizer. LAMB denotes the VBPC trained with LAMB optimizer. Here, we report ACC and NLL for both optimizers.

| | | LAMB | | VBPC | | | | LAMB | | VBPC | |
|---|---|---|---|---|---|---|---|---|---|---|---|
| Dataset | ipc | ACC($\uparrow$) | NLL($\downarrow$) | ACC($\uparrow$) | NLL($\downarrow$) | Dataset | ipc | ACC($\uparrow$) | NLL($\downarrow$) | ACC($\uparrow$) | NLL($\downarrow$) |
| **CIFAR10** | 1 | 54.4$_{\pm0.8}$ | 1.36$_{\pm0.03}$ | 55.1$_{\pm0.3}$ | 1.34$_{\pm0.08}$ | **CIFAR100** | 1 | 38.5$_{\pm0.5}$ | 2.46$_{\pm0.03}$ | 38.4$_{\pm0.2}$ | 2.47$_{\pm0.04}$ |
| | 10 | 69.4$_{\pm0.5}$ | 0.90$_{\pm0.02}$ | 69.8$_{\pm0.7}$ | 0.89$_{\pm0.02}$ | | 10 | 49.4$_{\pm0.2}$ | 2.25$_{\pm0.10}$ | 49.4$_{\pm0.1}$ | 2.07$_{\pm0.02}$ |

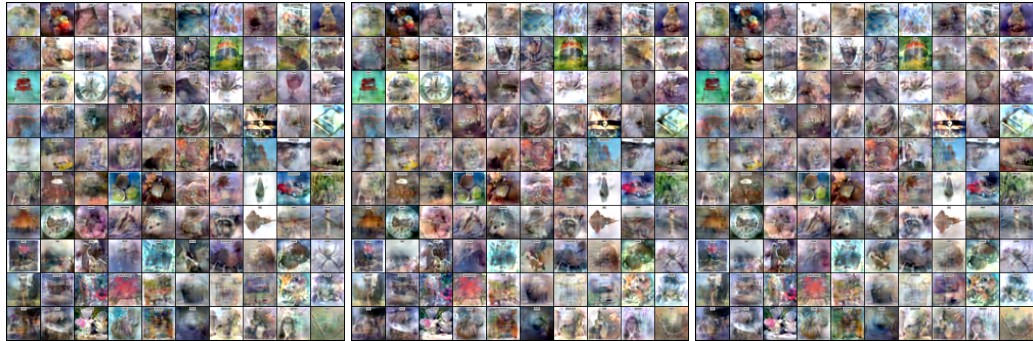

**Figure 5:** Learned VBPC images utilizing different maximum updates steps for the model pool elements in the CIFAR100 ipc 10 experiment. The left figure shows the $T = 100$ case which is the default setting for the all experiments. The middle and the right figures show the $T = 200$ and $T = 400$ cases. The learned images show minor difference in visual.

tion for each image. Also, results presented in Table 16 clearly shows that the BMA performance with VBPC variational distribution largely drops without label learning. These results validate that learning the label is important for the successful VBPC training.

### F.9 ABLATION ON GAUSSIAN NOISE AUGMENTATION

During VBPC training, we apply Gaussian noise augmentation. Based on previous findings that adding Gaussian noise to images during neural network training improves robustness to various image corruptions (Rusak et al., 2020), we incorporate Gaussian noise during VBPC training. This helps the learned pseudo-coreset dataset produce a variational posterior that is robust to unseen model structures and corrupted test datasets. Specifically, we add Gaussian noise sampled from $\mathcal{N}(0, 0.01)$ after normalizing the images using the predefined mean and standard deviation for all tasks. We conduct the ablation experiment on the existence of this Gaussian Noise during training VBPC utilizing CIFAR10/100 1 ipc and 10 ipc settings. As clearly seen in the CIFAR10 1 ipc case in Fig. 7, training with Gaussian noise results in much clearer and brighter images. In contrast, without Gaussian noise, the model tends to learn visually similar features in the background, unlike the cases where noise is applied. Table 17 confirms that, as expected, the overall performance decreases when Gaussian noise augmentation is not applied, compared to VBPC with Gaussian noise augmentation.

**Table 15:** Ablation results on the model pool maximum update steps. Here, we used CIFAR100 ipc 10 setting for the ablation. $T = 200$ and $T = 400$ indicate the maximum updates for the model pool is 200 and 400, respectively. Here, we report ACC and NLL for all the update steps.

| | | VBPC | | $T = 200$ | | $T = 400$ | |
|---|---|---|---|---|---|---|---|
| Dataset | ipc | ACC($\uparrow$) | NLL($\downarrow$) | ACC($\uparrow$) | NLL($\downarrow$) | ACC($\uparrow$) | NLL($\downarrow$) |
| **CIFAR100** | 10 | 49.4$_{\pm0.1}$ | 2.07$_{\pm0.02}$ | 48.7$_{\pm0.2}$ | 2.16$_{\pm0.03}$ | 48.0$_{\pm0.2}$ | 2.22$_{\pm0.04}$ |

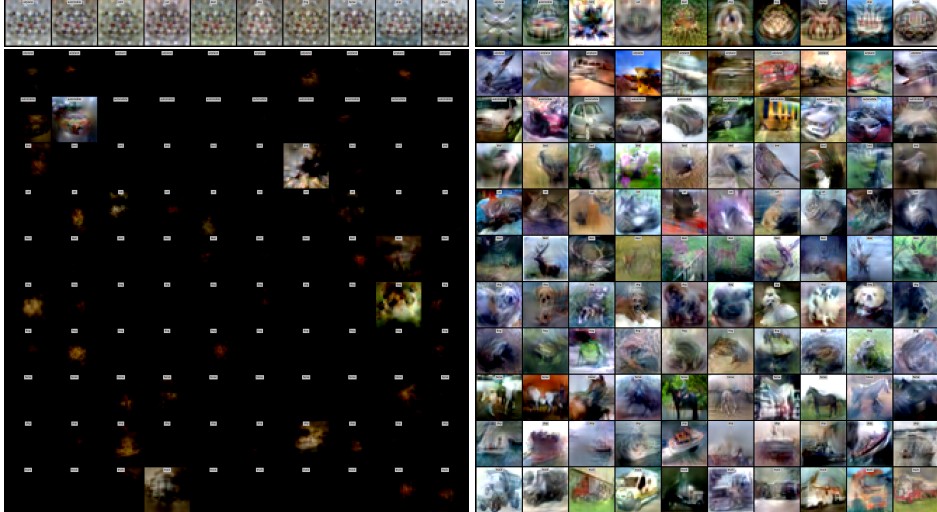

**Figure 6:** Visualization of learned VBPC images with and without label learning for the CIFAR10 ipc 1 (above) and ipc 10 (below). The left figure shows the learned VBPC images without label learning and the right figure shows the learned VBPC images with label learning.

**Table 16:** Comparison between the VBPC learned with and without label learning. No Label denotes the VBPC trained without label learing. Here, we report ACC and NLL for both results.

| Dataset | ipc | No Label ACC(↑) | NLL(↓) | VBPC ACC(↑) | NLL(↓) | Dataset | ipc | No Label ACC(↑) | NLL(↓) | VBPC ACC(↑) | NLL(↓) |
|---|---|---|---|---|---|---|---|---|---|---|---|
| **CIFAR10** | 1 | $40.6_{\pm1.3}$ | $1.83_{\pm0.02}$ | $55.1_{\pm0.3}$ | $1.34_{\pm0.08}$ | **CIFAR100** | 1 | $10.1_{\pm0.0}$ | $5.12_{\pm0.05}$ | $38.4_{\pm0.2}$ | $2.47_{\pm0.04}$ |
| | 10 | $56.6_{\pm0.3}$ | $1.54_{\pm0.03}$ | $69.8_{\pm0.7}$ | $0.89_{\pm0.02}$ | | 10 | $23.5_{\pm1.2}$ | $4.92_{\pm0.06}$ | $49.4_{\pm0.1}$ | $2.07_{\pm0.02}$ |

### F.10 ABLATION ON HYPERPARAMETER

In this section, we conduct ablation experiments on $\rho$ and $\gamma$, which are the hyperparameters newly proposed in our work. We set the default values to $\rho = 10$ and $\gamma = 100.0$ for the CIFAR100 ipc 10 settings. And, we conduct ablation experiment with the CIFAR100 ipc 10 setting. And the results presented in Table 18 and Table 19 show that even when we varied our hyperparameters by orders of magnitude (in log scale, with changes up to 10-fold), the performance remains consistently similar. And this concludes that our method works robustly with respect to hyperparameter changes.

### F.11 ABLATION ON TRAINING STEPS DURING INFERENCE

In this section, we conduct ablation experiments on the number of training steps $T'$ during inference. When learning the pseudo-coreset using the VBPC method, we leverage a model pool to allow the data to observe various feature maps, ensuring diverse learning. Therefore, even during inference, although the model may not perfectly fit the pseudo-coreset that was trained with the feature map, it can still approximate the best variational distribution for the current last-layer weights based on

**Table 17:** Comparison between the VBPC learned with and without Gaussian Noise augmentation. No Noise denotes the VBPC trained without Gaussian Noise augmentation. Here, we report ACC and NLL for both results.

| Dataset | ipc | No Noise ACC(↑) | NLL(↓) | VBPC ACC(↑) | NLL(↓) | Dataset | ipc | No Noise ACC(↑) | NLL(↓) | VBPC ACC(↑) | NLL(↓) |
|---|---|---|---|---|---|---|---|---|---|---|---|
| **CIFAR10** | 1 | $53.9_{\pm0.8}$ | $1.41_{\pm0.04}$ | $55.1_{\pm0.3}$ | $1.34_{\pm0.08}$ | **CIFAR100** | 1 | $35.4_{\pm0.3}$ | $2.62_{\pm0.05}$ | $38.4_{\pm0.2}$ | $2.47_{\pm0.04}$ |
| | 10 | $68.8_{\pm0.7}$ | $0.92_{\pm0.04}$ | $69.8_{\pm0.7}$ | $0.89_{\pm0.02}$ | | 10 | $48.5_{\pm0.4}$ | $2.22_{\pm0.06}$ | $49.4_{\pm0.1}$ | $2.07_{\pm0.02}$ |

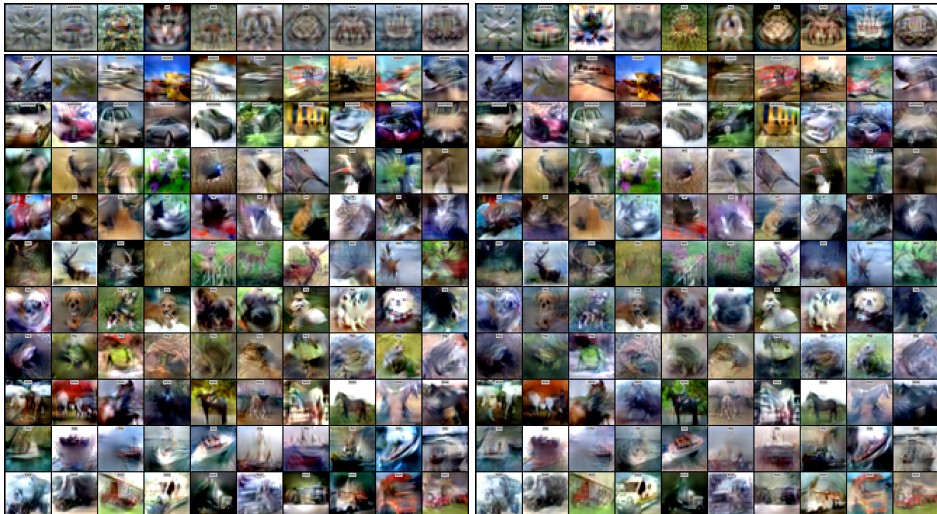

**Figure 7:** Visualization of learned VBPC images utilizing Gaussian noise during training for the CIFAR10 ipc 1 (above) and ipc 10 (below). The left figure shows the learned VBPC images without the Gaussian noise and the right figure shows the learned VBPC images with the Gaussian noise.

**Table 18:** Ablation results on the hyperparamer $\gamma$. Here, we used CIFAR100 ipc 10 setting for the ablation. $\gamma = 10$, $\gamma = 1000$, and $\gamma = 10000$ indicate that we set $\gamma$ as 10, 1000, and 10000, respectively. Our default setting is $\gamma = 1$. Here, we report ACC and NLL for all the update steps.

| | | $\gamma = 10$ | | **VBPC** | | $\gamma = 1000$ | | $\gamma = 10000$ | |
| Dataset | ipc | ACC($\uparrow$) | NLL($\downarrow$) | ACC($\uparrow$) | NLL($\downarrow$) | ACC($\uparrow$) | NLL($\downarrow$) | ACC($\uparrow$) | NLL($\downarrow$) |
|---|---|---|---|---|---|---|---|---|---|
| **CIFAR100** | 10 | $48.5_{\pm 0.2}$ | $2.30_{\pm 0.03}$ | $49.4_{\pm 0.1}$ | $2.07_{\pm 0.02}$ | $49.4_{\pm 0.4}$ | $2.14_{\pm 0.04}$ | $49.0_{\pm 0.5}$ | $2.10_{\pm 0.02}$ |

the available feature map. This enables the model to achieve sufficient BMA performance even before the pseudo-coreset learning is fully completed. As shown in Table 20, the model exhibits slightly lower performance during initial steps, such as at 400 or 800 steps, compared to the best performance. However, after these early stages, the performance becomes nearly identical to the final convergence step at 2000 steps. These results further demonstrate that our VBPC approach allows for fast and efficient posterior approximation.

# G  TRAINED VBPC IMAGES

In this section, we present the images learned under the ipc 1, 10, and 50 settings for MNIST, Fashion-MNIST, CIFAR10, CIFAR100, and Tiny-ImageNet. To avoid overwhelming the report with too many images, we have limited the number of reported images to a maximum of 100 per task.

## G.1  TAKE-HOME MESSAGE FROM LEARNED IMAGES

Regarding the learned pseudo-coreset images for CIFAR10, the results can be found in Fig. 12 and left figure of Fig. 13, showing the outcomes for ipc values of 1 and 10. These images reveal several interesting aspects of how VBPC captures information.

First, both ipc 1 and ipc 10 images show that VBPC effectively learns features associated with specific classes, such as "horse" or "automobile," as can be visually confirmed. This indicates that the pseudo-coreset images retain class-relevant information necessary for approximating the original dataset's posterior distribution. When comparing ipc 1 and ipc 10, there are notable differences. In the case of ipc 1, where only a single image per class is available, VBPC attempts to encode as many class-specific features as possible into a single image. As a result, the learned image appears to incorporate multiple discriminative features from the class symmetrically. In contrast, with ipc

**Table 19:** Ablation results on the hyperparamer $\rho$. Here, we used CIFAR100 ipc 10 setting for the ablation. $\rho = 1$, $\rho = 100$, and $\rho = 1000$ indicate that we set $\rho$ as 1, 100, and 1000, respectively. Our default setting is $\rho = 10$. Here, we report ACC and NLL for all the update steps.

| | | $\rho = 1$ | | **VBPC** | | $\rho = 100$ | | $\rho = 1000$ | |
|---|---|---|---|---|---|---|---|---|---|
| Dataset | ipc | ACC($\uparrow$) | NLL($\downarrow$) | ACC($\uparrow$) | NLL($\downarrow$) | ACC($\uparrow$) | NLL($\downarrow$) | ACC($\uparrow$) | NLL($\downarrow$) |
| **CIFAR100** | 10 | $49.0_{\pm0.3}$ | $2.10_{\pm0.02}$ | $49.4_{\pm0.1}$ | $2.07_{\pm0.02}$ | $49.0_{\pm0.2}$ | $2.20_{\pm0.03}$ | $47.5_{\pm0.4}$ | $2.35_{\pm0.04}$ |

**Table 20:** Ablation results on the training step $T'$ during inference. Here, we used CIFAR100 ipc 10 setting for the ablation. $T' = 400$, $T' = 800$, $T' = 1200$, and $T' = 1600$ indicate that the intermediate performance at step 400, 800, 1200, and 1600, respectively. Our default setting is $T' = 2000$. Here, we report ACC and NLL for all the update steps.

| | | $T' = 400$ | | $T' = 800$ | | $T' = 1200$ | | $T' = 1600$ | | **VBPC** | |
|---|---|---|---|---|---|---|---|---|---|---|---|
| Dataset | ipc | ACC($\uparrow$) | NLL($\downarrow$) | ACC($\uparrow$) | NLL($\downarrow$) | ACC($\uparrow$) | NLL($\downarrow$) | ACC($\uparrow$) | NLL($\downarrow$) | ACC($\uparrow$) | NLL($\downarrow$) |
| **CIFAR100** | 10 | $48.9_{\pm0.3}$ | $2.16_{\pm0.01}$ | $49.2_{\pm0.3}$ | $2.09_{\pm0.01}$ | $49.4_{\pm0.3}$ | $2.07_{\pm0.01}$ | $49.5_{\pm0.2}$ | $2.07_{\pm0.01}$ | $49.4_{\pm0.1}$ | $2.07_{\pm0.02}$ |

10, where more images per class are available, VBPC distributes the class-relevant features across multiple images. This leads to a greater diversity of features being captured across the pseudo-coreset, enabling a more comprehensive representation of the class.

Additionally, both ipc 1 and ipc 10 images often include low-level features beyond the main class-relevant ones. These features likely help capture the dataset's variability and ensure the learned pseudo-coreset maintains a close approximation of the original data distribution.

These observations suggest that VBPC is effective in compressing the dataset while retaining essential information. The learned images illustrate how VBPC balances feature extraction and information retention to ensure that the variational posterior distribution learned using the pseudo-coreset closely approximates the one learned using the full dataset. This further validates the interpretability and utility of VBPC in various tasks.

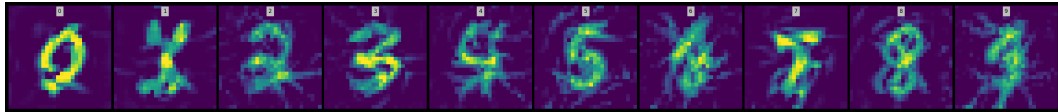

Figure 8: Visualization of learned VBPC images for the MNIST ipc 1.

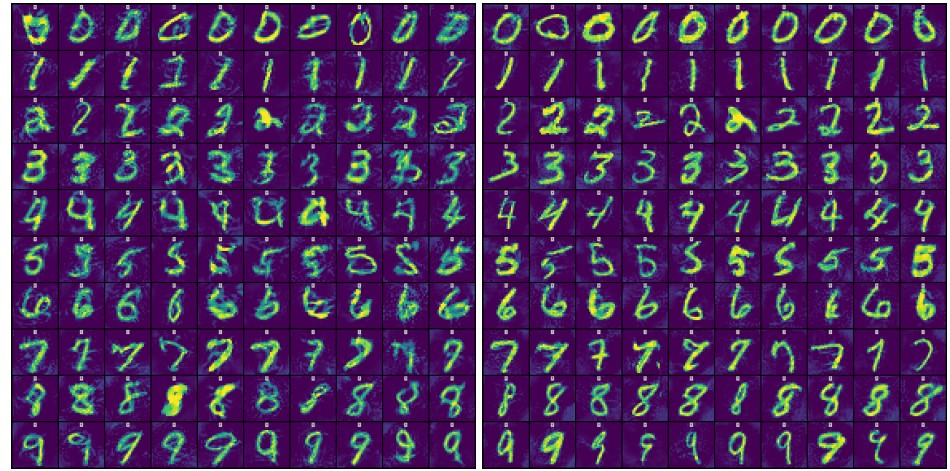

Figure 9: Visualization of learned VBPC images for the MNIST ipc10 (left) and ipc50 (right).

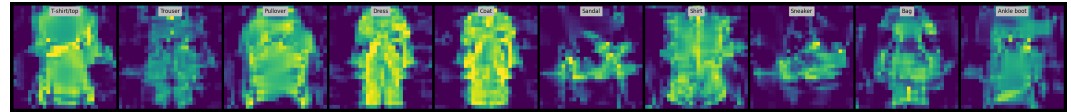

Figure 10: Visualization of learned VBPC images for the Fashion-MNIST ipc1.

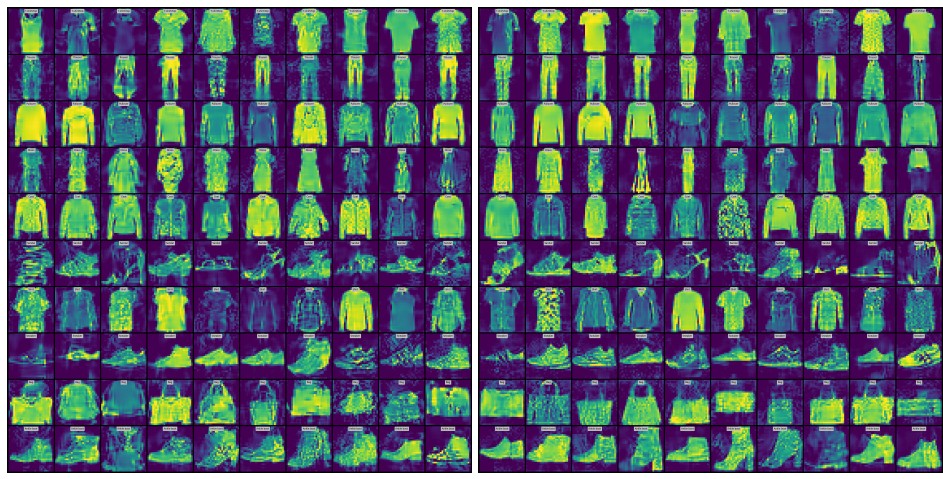

Figure 11: Visualization of learned VBPC images for the Fashion-MNIST ipc10 (left) and ipc50 (right).

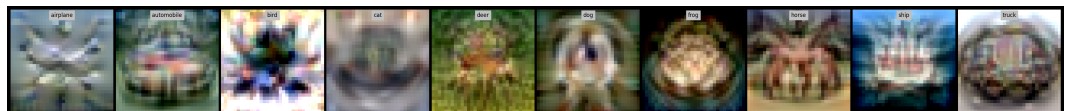

Figure 12: Learned VBPC images for the CIFAR10 ipc 1 case.

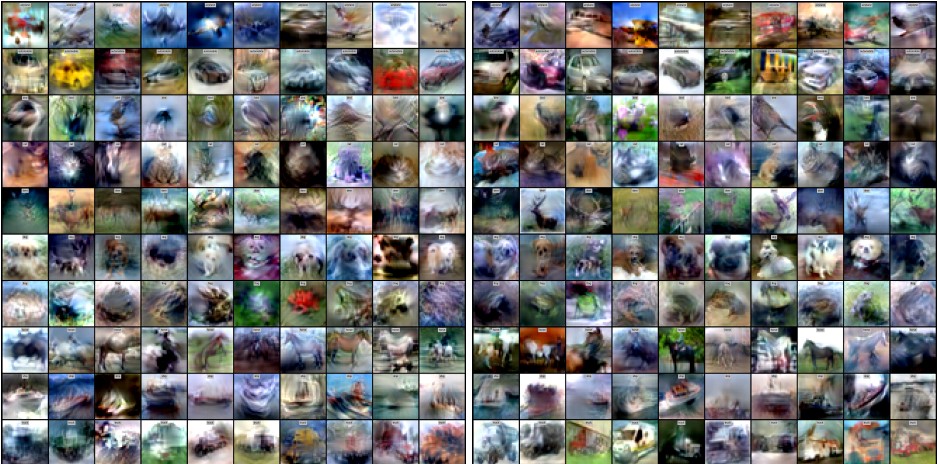

**Figure 13:** Visualization of learned VBPC images for the CIFAR10 ipc10 (left) and ipc50 (right).

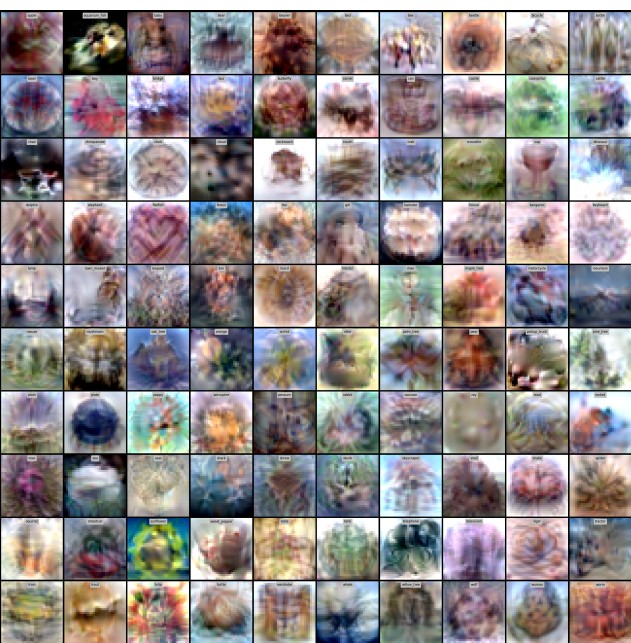

**Figure 14:** Visualization of learned VBPC images for the CIFAR100 ipc1.

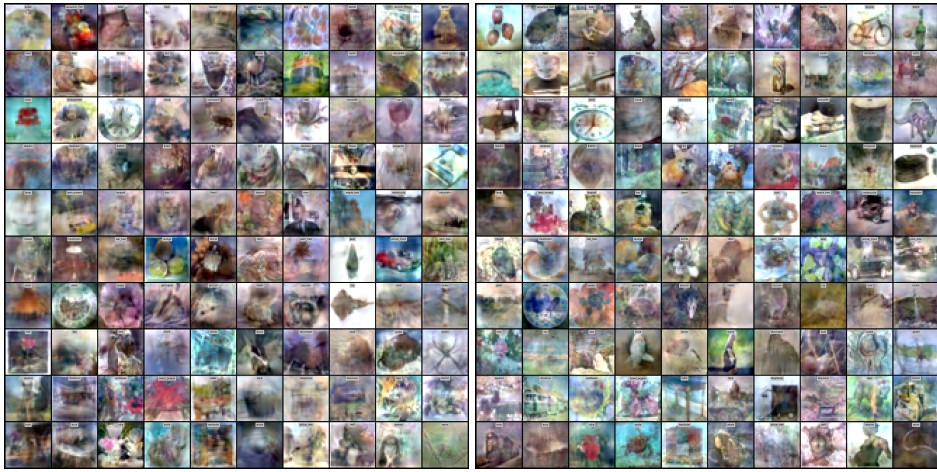

**Figure 15:** Visualization of learned VBPC images for the CIFAR100 ipc10 (left) and ipc50 (right).

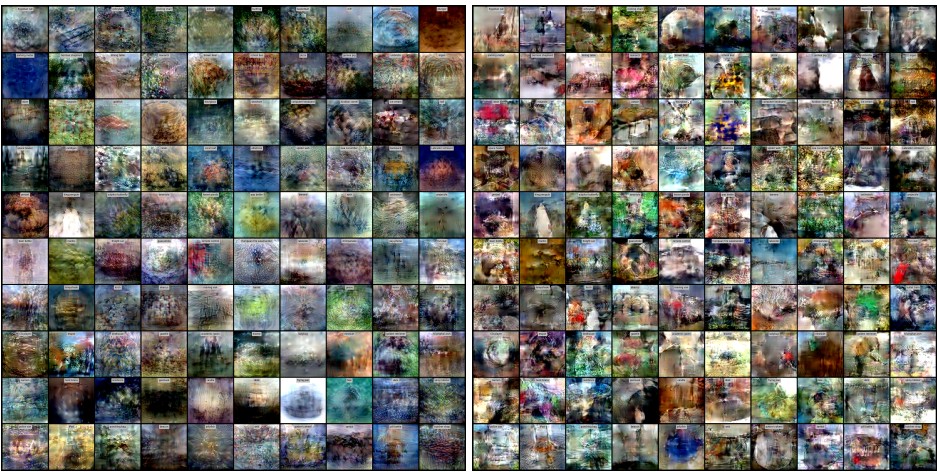

**Figure 16:** Visualization of learned VBPC images for the Tiny-ImageNet ipc1 (left) and ipc10 (right).

