# OpenReview forum: "Variational Bayesian Pseudo-Coreset"
_ICLR.cc/2025/Conference — ICLR 2025 Poster_

### Official Review · Reviewer_oE39 · 2024-10-20

**Soundness:** 3
**Presentation:** 2
**Contribution:** 2
**Rating:** 5
**Confidence:** 4

**Summary:**

The paper titled “Variational Bayesian Pseudo-Coreset” introduces a novel method aimed at reducing the computational and memory challenges associated with large datasets in deep learning, particularly within the context of Bayesian Neural Networks (BNNs). The authors address limitations in prior methods of Bayesian Pseudo-Coresets (BPCs), which often face inefficiencies in memory usage and suboptimal results during training. They propose a new approach called Variational Bayesian Pseudo-Coreset (VBPC), which leverages variational inference to approximate the posterior distribution of model weights. The key innovation of VBPC is the use of a closed-form solution to compute the posterior for the last layer of BNNs, eliminating the need for complex gradient-stopping techniques used in previous BPC methods. This significantly reduces memory usage and computational load. Additionally, VBPC allows for more efficient training and inference by using a single forward pass for predictive distribution computation. Empirical evaluations demonstrate that VBPC outperforms existing BPC methods on benchmark datasets, showing improvements in both accuracy and negative log-likelihood metrics across various datasets, such as MNIST, CIFAR10, and CIFAR100. The paper contributes to the field by enhancing the efficiency and scalability of BNNs, particularly in environments that require handling large-scale data while maintaining the benefits of Bayesian inference.

**Strengths:**

The paper introduces a novel approach to improving the efficiency of Bayesian Neural Networks (BNNs) by combining variational inference with pseudo-coresets. This innovation is notable because it addresses the longstanding challenges in the field related to computational and memory inefficiencies when scaling BNNs to large datasets. The proposal of a closed-form solution for last-layer variational inference is a significant departure from prior methods that relied on more memory-intensive sampling-based approaches. By focusing on variational inference and pseudo-coresets, the authors provide an original contribution that builds on existing methods but removes key limitations such as memory usage and the reliance on gradient stopping.
The technical rigor of the paper is high, with well-founded theoretical development and comprehensive empirical evaluations. The authors derive closed-form solutions for coreset variational inference, addressing a critical computational bottleneck in Bayesian model averaging. Their empirical results, demonstrated across multiple datasets (e.g., MNIST, CIFAR10, CIFAR100), show significant improvements over existing BPC methods in terms of both accuracy and negative log-likelihood, which strengthens the quality of their proposed method. The paper also includes a variety of comparisons with competitive baselines, reinforcing the robustness and effectiveness of their approach.
The paper is clearly structured and provides sufficient background for readers to understand both the motivation and the details of the proposed method. The explanation of the problem, the limitations of prior work, and the step-by-step presentation of the VBPC approach are clear and easy to follow. The use of mathematical derivations is well-supported by intuitive explanations, making the complex variational inference approach more accessible. The inclusion of visual results and performance tables also contributes to clarity, helping readers visualize the practical benefits of VBPC.
The significance of the work lies in its potential to influence how BNNs are applied to large-scale data problems. By significantly reducing the memory and computational burdens of Bayesian model averaging, the proposed VBPC method makes BNNs more feasible for real-world applications, such as those in healthcare and climate analysis, where uncertainty estimation is critical. The method could have broad implications for other fields requiring scalable, probabilistic neural networks. Additionally, the ability to perform Bayesian inference with less computational overhead enhances the practicality of deploying BNNs in production environments.

**Weaknesses:**

The paper focuses heavily on benchmark datasets like MNIST, CIFAR10, and CIFAR100, which are common in academic research but may not fully represent the complexity of real-world problems. While these datasets help establish baseline performance, the paper would benefit from exploring more challenging, domain-specific datasets, particularly those that are more representative of practical applications in fields such as healthcare or finance. Expanding the evaluation to datasets that feature more variability and noise could demonstrate the method’s robustness in real-world settings, which is especially important given the paper’s goal of making Bayesian Neural Networks more feasible for large-scale applications.
Αlthough the paper demonstrates memory efficiency improvements, there is no extensive discussion of the scalability of the method when applied to very large datasets beyond those tested (e.g., ImageNet or even larger datasets in natural language processing). The paper could benefit from a more detailed analysis of the method’s behavior as the dataset size grows significantly. Additionally, the paper does not provide enough insight into the sensitivity of the method to hyperparameter choices such as the coreset size or the initialization of the model pool. It would be helpful to include an ablation study or sensitivity analysis that investigates how performance degrades with suboptimal hyperparameter choices and whether the method requires careful tuning to achieve competitive results.
While the paper emphasizes memory savings, it does not provide a thorough comparison of training times between VBPC and existing methods, particularly in scenarios with high-dimensional datasets. A more detailed analysis of wall-clock time or computational complexity across different hardware configurations (e.g., GPUs versus CPUs) would be useful. This would help practitioners better understand the trade-offs between memory savings and potential increases in computational time, especially when scaling to larger architectures and datasets.
The paper relies on the last-layer variational approximation to simplify the posterior calculation, but the limitations of this approach are not thoroughly discussed. While the paper suggests that this approximation performs comparably to more complex methods, it would be valuable to include a deeper investigation of when this approximation might fail, especially in models with deep architectures or tasks requiring fine-grained uncertainty estimation. A discussion on whether the approximation is sufficient in all use cases or only certain tasks (e.g., classification versus regression) would make the paper more transparent.
The paper demonstrates that VBPC can learn pseudo-coresets that effectively approximate the full dataset’s posterior, but it doesn’t provide much insight into the interpretability of these pseudo-coresets. For example, what are the learned coresets capturing in terms of dataset distribution or feature representation? A qualitative analysis, such as visualizing the pseudo-coresets or interpreting what aspects of the data they retain, would help reinforce the method’s effectiveness. Additionally, further explanation of how these pseudo-coresets evolve during training and contribute to Bayesian uncertainty could strengthen the narrative.
While the paper briefly touches on robustness to distributional shifts using CIFAR10-C, the evaluation of predictive uncertainty in real-world settings is somewhat lacking. It would be useful to see how VBPC handles more complex out-of-distribution (OOD) detection tasks or how well it captures uncertainty under adversarial conditions, which are critical aspects of Bayesian inference in high-stakes applications like healthcare. A more thorough evaluation in these contexts could elevate the practical relevance of the method.

**Questions:**

1. Scalability to Larger Datasets: How does VBPC perform when applied to much larger datasets, such as ImageNet or larger text datasets? Does the method scale well in terms of both computational efficiency and accuracy, or does it encounter bottlenecks?

2. Hyperparameter Sensitivity: How sensitive is VBPC to hyperparameters such as the number of pseudo-coresets, coreset size, and model initialization? Do suboptimal hyperparameter settings lead to significant performance degradation?

3. Computational Costs and Training Time: Can you provide a detailed comparison of training times and wall-clock time between VBPC and other methods, particularly Bayesian Pseudo-Coreset methods using SGMCMC? How does the computational time scale with increasing dataset size?

4. Limitations of the Last-Layer Approximation: Does the last-layer variational approximation hold up in deeper networks or more complex tasks such as regression? Have you observed any failure cases where this approximation does not capture enough uncertainty?

5. Interpretability of Pseudo-Coresets: What do the learned pseudo-coresets represent? Are they capturing key features of the original dataset, and if so, how do they evolve during training? Is there a way to interpret or visualize the coreset to provide better intuition about what is being distilled?

6. Generalization to Different Tasks: Can the VBPC method be applied effectively to tasks beyond classification, such as regression or other types of Bayesian inference? If so, how does the method adapt to these different problem types?

7. Robustness to Out-of-Distribution (OOD) Data and Adversarial Attacks: Does VBPC provide any robustness to adversarial attacks or strong distribution shifts beyond what is demonstrated with CIFAR10-C? How does the method perform in more severe OOD scenarios?

8. Memory-Efficient Loss Computation: How significant is the impact of memory-efficient loss computation during training in terms of accuracy or stability? Does it introduce any trade-offs in performance, particularly in very high-dimensional settings?

---

> ### Author Response · Authors · 2024-11-18
> **Rebuttal by Authors**
>
> **[Q1] Scalability to Larger Datasets**
>
> Thanks for the constructive comment. Following your suggestion, we aim to further demonstrate the effectiveness of our VBPC method by showing that it not only achieves good performance on larger-scale datasets, which other BPC methods even struggle to train for the large ipc but also outperforms other BPC baselines in a continual learning setting. By doing so, we hope to highlight VBPC's ability to handle tasks that pose challenges for other BPC baselines, proving its versatility and effectiveness.
>
> First, to show that our method is uniquely scalable to large datasets compared to other BPC methods, we conducted additional experiments on the ImageNetWoof (128x128x3) dataset [1] and the resized ImageNet1k (64x64x3) dataset [2]. Additionally, we included an experiment in a continual learning scenario to validate that our method performs better in practical scenarios.
>
> We conducted experiments on the ImageWoof (128x128x3) dataset with ipc 1 and ipc 10 settings, as well as the resized ImageNet1k (64x64x3) dataset with ipc 1 and ipc 2 settings, to demonstrate the scalability of our method to high-resolution images and larger datasets. Unlike existing BPC baselines, which encountered memory issues and failed to train due to out-of-memory errors on an RTX 3090 GPU as the image resolution and number of classes increased, our method successfully completed training. Table R.7 clearly shows that VBPC significantly outperforms other baselines with a large margin for both the ImageWoof and resized ImageNet1k datasets.
>
> Next, we validated the practical effectiveness of our method through continual learning experiments using pseudo-coreset images learned by each method. We followed the continual learning setup described in [3,4], where class-balanced training examples are greedily stored in memory, and the model is trained from scratch using only the latest memory. Specifically, we performed a 5-step class incremental learning experiment on CIFAR100 with an ipc 20 setting, following the class splits proposed in [3,4]. Table R.8 demonstrates that VBPC consistently outperforms other baselines across all steps, confirming its superior practicality and effectiveness in real-world continual learning scenarios.
>
> __Table R.7.__ Experiments on the scalability utilizing ImageWoof and resized ImageNet datasets. Here ‘-’ indicates the training fails due to the out-of-memory problems.
> | Metric  | ImageWoof |ipc 1          | ImageWoof| ipc 10          | ImageNet| ipc 1             | ImageNet |ipc 2         |
> |:-           |:-       |:-                     |:-          |:-                   |:-                |:-           |:-                |:-          |
> |             | ACC       | NLL       | ACC       | NLL       | ACC       | NLL       | ACC       | NLL       |
> |:-           |:-       |:-                     |:-          |:-                   |:-                |:-           |:-                |:-          |
> | Random  | 14.2 ± 0.9 | 3.84 ± 0.25 | 27.0 ± 1.9 | 2.83 ± 0.33 | 1.1 ± 0.1 | 8.32 ± 0.05 | 1.4 ± 0.1 | 8.10 ± 0.05 |
> | BPC-CD  | 18.5 ± 0.1 | 2.76 ± 0.05 | -          | -          | -         | -          | -         | -          |
> | FBPC    | 14.8 ± 0.1 | 3.73 ± 0.02 | 28.1 ± 0.3 | 2.69 ± 0.09 | -         | -          | -         | -          |
> | BPC-fKL | 14.9 ± 0.9 | 3.74 ± 0.23 | 25.0 ± 0.8 | 2.90 ± 0.27 | -         | -          | -         | -          |
> | BPC-rkL | 12.0 ± 0.5 | 6.07 ± 0.31 | -          | -          | -         | -          | -         | -          |
> | VBPC    | **31.2 ± 0.1** | **2.13 ± 0.04** | **39.0 ± 0.1** | **1.84 ± 0.1** | **10.0 ± 0.1** | **5.33 ± 0.04** | **11.5 ± 0.2** | **5.25 ± 0.05** |
>
>
> __Table R.8.__ Experiments on the continual learning setting. Here, we utilize the CIFAR100 dataset with ipc 20 setting. We assume 5 steps during training and each step contains data from new 20 classes in the CIFAR100 dataset. Here we only report accuracy due to the variant of the number of classes during the steps.
> | Number of Classes |      20        |           40          |          60         |           80        |          100  |
> |---------------------------|---------------|---------------------|-------------------|-------------------|----------------|
> |BPC-CD                  | 52.5 ± 2.4   |             40.4 ± 1.3  |            35.2 ± 0.8 |           33.4 ± 0.5 |      29.4 ± 0.2 |
> |FBPC                      | 61.4 ±  1.8   |           53.2 ± 1.5  |             48.8 ± 0.7 |          43.9 ± 0.4 |      41.2 ± 0.3  |
> |BPC-fKL                  | 51.8 ± 2.2 |             39.8 ± 1.1  |            35.5 ±  0. 7|         33.1 ± 0.5  |      29.5 ± 0.3 |
> |BPC-rKL                  | 48.2 ± 2.7 |           35.5 ±  1.8    |            32.0 ± 1.0 |          29.8 ± 0.6 |       25.5 ± 0.3 |
> |VBPC                      | 75.3 ± 2.0 |           65.8± 1.5      |           57.1 ±  0.9  |        53.3 ± 0.5  |      50.3 ± 0.2 |

---

> ### Author Response · Authors · 2024-11-18
> **Rebuttal by Authors**
>
> **[Q2] Hyperparameter Sensitivity**
>
> Thank you for the constructive comment. We appreciate your suggestion that analyzing the impact of hyperparameter selection on performance could provide deeper insights into our method. We fully agree with this point and have provided an extensive ablation study in Appendix E, where we detail the impact of various hyperparameters on model performance and the learned pseudo-coreset.
>
> Regarding hyperparameter sensitivity, our experiments show that while parameters like random initialization of the pseudo-coreset, $\gamma$, and $\rho$ exhibit minimal impact on performance, the most significant factor influencing the results is whether the pseudo-coreset labels are learned. As shown in Figure 6 and Table 11, not learning the labels results in substantially degraded performance compared to the full VBPC method. As analyzed in Appendix E.5, this can be attributed to the pseudo-coreset variational distribution’s mean being dependent on the labels, which directly affects the computation of the dataset VI problem. This indicates that learning the labels plays a critical role in our method's success.
>
> Furthermore, we observed a clear trend in the number of pseudo-coreset points on performance. As shown in our experiments for ipc = 1, 10, 50, increasing the number of pseudo-coreset points enhances performance. And we can think that the number approaches that of the full dataset, and the performance converges to that of using the entire dataset. These findings validate the trade-offs between memory/computation efficiency and model performace in our method.
>
> **[Q3] Computational Costs and Training Time**
>
> Thank you for highlighting the importance of our contribution regarding efficient Bayesian inference with respect to computational cost. To address your suggestion, we performed analyses focusing on two aspects of computational cost:
>
> 1. Cost of Training the pseudo-coreset:
>
> As mentioned in the paper, conventional BPC methods relying on SGMCMC require the creation of expert trajectories, which are training trajectories derived from the full dataset. Each dataset typically involves training with 10 different random seeds for these trajectories, making this step computationally expensive. Since all BPC baselines share and utilize these precomputed trajectories, their associated computational cost can be considered a shared overhead.
>
> To isolate the computational cost of training the pseudo-coreset itself, we measured the wall-clock time required for pseudo-coreset optimization by each method. The results of this comparison are summarized in Table R.9, providing insights into how VBPC reduces training costs compared to other baselines.
>
> 2. Cost of Inference:
>
> When performing inference, VBPC requires training only a single model, whereas other BPC baselines rely on multiple SGMCMC samples. Each sample incurs significant training and inference costs, which grow linearly with the number of samples.
>
> To quantify this difference, we measured the wall-clock time for inference across methods, with results presented in Table R.10. These results highlight how VBPC achieves superior efficiency during inference by avoiding the high computational costs associated with sampling-based approaches.
>
> These analyses demonstrate VBPC’s ability to perform Bayesian inference efficiently, both in terms of pseudo-coreset training and inference, and further reinforce the computational advantages of our method.
>
> __Table.R.9__ Wall clock time results for training pseudo-coresets with each BPC method using CIFAR10 ipc 10 settings. We used RTX3090 GPU to measure the exact training time. Here, all methods except for VBPC share the training time for expert trajectories.
>
> |Method|BPC-CD|BPC-rKL|FBPC|BPC-fKL|VBPC|
> |----------|-----------|-------------|--------|-----------|--------|
> |Times (hr)| 5 + 8.5 | 5 + 9 | 5 + 10.5 | 5 + 12 | 5.5|
>
> __Table.R.10__ Wall clock time results for inference using learned pseudo-coresets. We measure the inference time for evaluating all the test data from the CIFAR10 test dataset. After finishing training the pseudo-coresets, the inference cost for the all baselines are same because they only need SGMCMC and BMA with same number of datasets and weight samples.
>
> |Method|BPC-CD|BPC-rKL|FBPC|BPC-fKL|VBPC|
> |----------|------------|-----------|--------|------------|--------|
> |Time (s)| 165      |   165    |  165   |   165    | 20     |

---

> ### Author Response · Authors · 2024-11-18
> **Rebuttal by Authors**
>
> **[Q4] Limitations of the Last-Layer Approximation**
>
> Thank you for the insightful comment. As you pointed out, there might be concerns that considering the posterior distribution of only the last layer weights, rather than the entire parameter set, could limit the model's ability to capture uncertainty effectively, especially as the model size increases and tasks become more complex. We fully agree that this is a valid concern and would like to provide a discussion based on related findings.
>
> Specifically, [5] provides extensive empirical evidence on the effectiveness of last-layer variational inference. Their experiments span diverse tasks, including regression with UCI datasets, image classification using a Wide ResNet model, and sentiment classification leveraging LLM features from the OPT-175B model. They compared their method with other Bayesian inference approaches such as Dropout, Ensemble methods, and Laplace approximation for the full model. Their results demonstrate that even though last-layer variational inference focuses solely on the final layer weights, it achieves performance comparable to other comprehensive Bayesian inference techniques across various tasks.
>
> These findings indicate that while conducting Bayesian inference on the full set of weights in a neural network could potentially lead to more precise uncertainty estimation, employing last-layer variational inference is still effective in capturing meaningful uncertainty.
>
> We believe that extending VBPC to incorporate full-weight variational inference could be a promising direction for future work, offering the potential to further enhance the method's uncertainty estimation capabilities. We will include this discussion in the final manuscript to provide a balanced perspective and acknowledge possible avenues for improvement.
>
> **[Q5] Interpretability of Pseudo-Coresets**
>
> Thank you for asking about the interpretability of the VBPC images. Regarding the learned pseudo-coreset images for CIFAR10, the results can be found in Figure 1 of the main paper and Figure 12 in the appendix, showing the outcomes for ipc values of 1 and 10. These images reveal several interesting aspects of how VBPC captures information.
>
> First, both ipc 1 and ipc 10 images show that VBPC effectively learns features associated with specific classes, such as "horse" or "automobile," as can be visually confirmed. This indicates that the pseudo-coreset images retain class-relevant information necessary for approximating the original dataset’s posterior distribution. When comparing ipc 1 and ipc 10, there are notable differences. In the case of ipc 1, where only a single image per class is available, VBPC attempts to encode as many class-specific features as possible into a single image. As a result, the learned image appears to incorporate multiple discriminative features from the class symmetrically. In contrast, with ipc 10, where more images per class are available, VBPC distributes the class-relevant features across multiple images. This leads to a greater diversity of features being captured across the pseudo-coreset, enabling a more comprehensive representation of the class.
>
> Additionally, both ipc 1 and ipc 10 images often include low-level features beyond the main class-relevant ones. These features likely help capture the dataset's variability and ensure the learned pseudo-coreset maintains a close approximation of the original data distribution.
>
> These observations suggest that VBPC is effective in compressing the dataset while retaining essential information. The learned images illustrate how VBPC balances feature extraction and information retention to ensure that the variational posterior distribution learned using the pseudo-coreset closely approximates the one learned using the full dataset. This further validates the interpretability and utility of VBPC in various tasks.

---

> ### Author Response · Authors · 2024-11-18
> **Rebuttal by Authors**
>
> **[Q6] Generalization to Different Tasks like regression task**
>
> Thank you for the constructive comment regarding extending our method to other tasks such as regression tasks. Indeed, our VBPC framework is versatile, as it can be adapted to various tasks by adjusting the dataset VI’s likelihood function.
>
> In regression tasks, different likelihoods can be used; however, Gaussian likelihoods are especially prevalent, as they can enhance compatibility between pseudo-coreset VI and dataset VI, facilitating more effective learning of the pseudo-coreset. Even when using other likelihood functions, Gaussian likelihood can still be applied as an approximation during training. Additionally, if the likelihood has a closed-form solution, it would be feasible to align both the pseudo-coreset VI and dataset VI likelihoods to improve coherence and potentially achieve better performance.
>
> This flexibility proves the potential of our method to generalize to a broader range of tasks, including regression. We will incorporate this discussion into the final manuscript to clarify how our approach can be extended to handle different types of tasks through appropriate likelihood selection.
>
> **[Q7] Robustness to Out-of-Distribution Data**
>
>  Following your constructive suggestion, we have conducted additional Out-of-Distribution (OOD) detection experiments and reported the results. The metrics we evaluate include AUROC, AUPR-In, and AUPR-Out, where higher values indicate better performance. We used models trained with the CIFAR10 IPC 10 setting and evaluated them on CIFAR100, TinyImageNet, and SVHN datasets as OOD datasets.
>
> The results, presented in Table R.11, demonstrate that the pseudo-coreset learned by VBPC performs robustly in OOD detection scenarios. These findings, combined with the corruption experiments in the main paper, validate the effectiveness and robustness of VBPC under diverse and challenging evaluation conditions.
>
> __Table R.11__ AUROC, AUPR-In, and AUPR-Out results for the OOD detection task with a model trained with the learned pseudo-coresets. Note that we used the same model structure which is utilized when training pseudo-coresets.
>
> |                 |                 |AUROC|AUPR-In|AUPR-Out|
> |--------------|--------------|----------|------------|---------------|
> |Dataset    |Model       |            |               |                  |
> |--------------|--------------|----------|------------|---------------|
> |CIFAR100|BPC-CD   |49.84   | 49.74    | 50.13        |
> |                |BPC-fKL   |51.21   | 49.61    | 51.72        |
> |                |BPC-rKL  | 48.53   | 48.64    | 48.63        |
> |                |FBPC       | 49.69  |  49.28    | 49.70       |
> |                |VBPC       |  **54.61** | **54.59**   |  **54.25**       |
> |-------------|--------------|----------|------------|---------------|
> |TinyImage|BPC-CD   |49.09   | 52.79    | 45.88        |
> |                |BPC-fKL   |48.95   | 51.72    | 47.00        |
> |                |BPC-rKL  | 48.34   | 52.71    | 44.49        |
> |                |FBPC       | 45.39  |  49.70    | 43.14       |
> |                |VBPC       |  **52.85** |  **56.22**   |  **49.64**       |
> |-------------|--------------|----------|------------|---------------|
> |SVHN      |BPC-CD   |55.09   | 35.64    | 73.88        |
> |                |BPC-fKL   |54.26   | 34.78    | 75.47        |
> |                |BPC-rKL  | 42.61   | 28.29    | 67.15        |
> |                |FBPC       | 41.34  |  30.12    | 62.18       |
> |                |VBPC       | **68.50** |  **48.49**   |  **82.91**       |

---

> ### Author Response · Authors · 2024-11-18
> **Rebuttal by Authors**
>
> **[Q8] Effect of Memory Efficient Loss Computation**
>
> Thank you for raising the question about how our memory-efficient computation impacts learning. In fact, our memory-efficient loss computation and variational inference are mathematically equivalent to their non-memory-efficient counterparts. Through mathematical reformulations, we reduce operations like matrix inversion and multiplication into computations involving smaller matrices, resulting in the same theoretical outcomes.
>
> Consequently, these memory-efficient approaches yield identical learning results and performance in theory. Practically, while numerical errors can arise during computations, our method mitigates this by operating on smaller-scale matrices, which are less prone to significant numerical errors. This ensures that our approach not only reduces memory usage but also maintains robust and accurate computations, leading to reliable results in practice.
>
> **References**
>
> [1] Jeremy Howard. A smaller subset of 10 easily classified classes from imagenet, and a little more french, 2020. URL https://github.com/fastai/imagenette/
>
> [2] Olga Russakovsky, Jia Deng, Hao Su, Jonathan Krause, Sanjeev Satheesh, Sean Ma, Zhiheng Huang, Andrej Karpathy, Aditya Khosla, Michael Bernstein, Alexander C. Berg, and Li Fei-Fei. ImageNet Large Scale Visual Recognition Challenge. International Journal of Computer Vision (IJCV), 115(3):211–252, 2015.
>
> [3] Yongchao Zhou, Ehsan Nezhadarya, and Jimmy Ba. Dataset distillation using neural feature regression. In Advances in Neural Information Processing Systems 35 (NeurIPS 2022), 2022.
>
> [4] Bo Zhao and Hakan Bilen. Dataset condensation with distribution matching. CoRR, abs/2110.04181, 2021. URL https://arxiv.org/abs/2110.04181.
>
> [5] J. Harrison, J. Willes, and J. Snoek. Variational Bayesian last layers. In International Conference on Learning Representations (ICLR), 2024a.

---

> ### Author Response · Authors · 2024-11-22
>
> Thank you for your dedication and interest in our paper. As the author and reviewer discussion period approaches its end, we are curious to know your thoughts on our rebuttal and whether you have any additional questions.

---

> ### Author Response · Authors · 2024-11-24
>
> We sincerely appreciate the effort and dedication you have put into reviewing our paper. As the deadline for authors to respond or engage in further discussions approaches, we are curious if you have any remaining concerns. We kindly request your feedback on our responses to address any additional questions you may have.

---

### Official Review · Reviewer_sqcH · 2024-10-31

**Soundness:** 3
**Presentation:** 3
**Contribution:** 2
**Rating:** 8
**Confidence:** 3

**Summary:**

The paper studies the problem of core set extraction using Bayesian inference. The proposed solution builds on a two-stage variational inference scheme where the first stage is responsible for inferring the optimal core set while the second is to fit this core set to the full-scale data set at hand. The developed solution overarches the whole family of the distributions that belong to the exponential family.

**Strengths:**

- The paper is particularly well-written with a clearly defined problem scope and a solid solution methodology that follows a well-justified sequence of development steps,
- The proposed bilevel variational inference formulation is neat and sensible.
- The computational complexity analysis is indeed helpful to see the merit of the devised solution.
- The reported results are strong on the chosen group of data sets.

**Weaknesses:**

- The paper motivates the core set extraction problem with use cases such as processing big data and addressing continual learning setups. However, the presented results are on data sets that can be considered in the present technological landscape as toy problems. I do symphathize the idea of prototyping. But given the strong applied component of the present work, I am still unconvinced about the generalizability of the observed scores to a case where coreset extraction is an actual necessity. The issue may be addressed during the rebuttal by showing results on a large enough data set used as a standard coreset extraction benchmark or a continual learning application.
- The need to use the Gaussian likelihood to avoid the need for an approximation stage is only partially convincing. It is an artiffact of choosing variational Bayes as the inference scheme, which is exogeneous to the problem anyway. Maybe this issue, linked to my first question below, will be clarified during the rebuttal.

**Questions:**

- Would Laplace approximation on the softmax likelihood be a better option than first choosing a variational inference scheme and then using a Gaussian likelihood? Laplace proves to be a powerful approach in Gaussian process classification.
- Is the claim of Manousakas et al. 2020 contrary to the main message of the paper? If correct, would this not undermine the significance of the proposed solution? If incorrect, why?
- Do we really lack an analytical solution or at least an EM-like algorithm where the E-step has an analytical solution when only the last layer of a neural net is probabilistic?
- What is the take-home message of Figure 1? I am missing to see a particular pattern there that helps motivate the proposed solution.

My initial score is a borderline as the paper has both certain merits and clear question marks. I am happy to consider significant score updates based on a convincing rebuttal discussion.

---
POST-REBUTTAL: The authors gave convincing answers to the above questions and they published additional results that demonstrate the advantages of the proposed method more clearly than the experiments reported in the original submission. I update my score to an accept.

---

> ### Author Response · Authors · 2024-11-18
> **Rebuttal by Authors**
>
> **[W1] Large Dataset and Continual Learning**
>
> Thanks for the constructive comment. Following your suggestion, we aim to further demonstrate the effectiveness of our VBPC method by showing that it not only achieves good performance on larger-scale datasets, which other BPC methods even struggle to train for the large ipc but also outperforms other BPC baselines in a continual learning setting. By doing so, we hope to highlight VBPC's ability to handle tasks that pose challenges for other BPC baselines, proving its versatility and effectiveness.
>
> First, to show that our method is uniquely scalable to large datasets compared to other BPC methods, we conducted additional experiments on the ImageNetWoof (128x128x3) dataset [1] and the ImageNet1k (64x64x3) dataset [2]. Additionally, we included an experiment in a continual learning scenario to validate that our method performs better in practical scenarios.
>
> We conducted experiments on the ImageWoof (128x128x3) dataset with ipc 1 and ipc 10 settings, as well as the resized ImageNet1k (64x64x3) dataset with ipc 1 and ipc 2 settings, to demonstrate the scalability of our method to high-resolution images and larger datasets. Unlike existing BPC baselines, which encountered memory issues and failed to train due to out-of-memory errors on an RTX 3090 GPU as the image resolution and number of classes increased, our method successfully completed training. Table R.5 clearly shows that VBPC significantly outperforms other baselines with a large margin for both the ImageWoof and resized ImageNet1k datasets.
>
> Next, we validated the practical effectiveness of our method through continual learning experiments using pseudo-coreset images learned by each method. We followed the continual learning setup described in [3,4], where class-balanced training examples are greedily stored in memory, and the model is trained from scratch using only the latest memory. Specifically, we performed a 5-step class incremental learning experiment on CIFAR100 with an ipc 20 setting, following the class splits proposed in [3,4]. Table R.6 demonstrates that VBPC consistently outperforms other baselines across all steps, confirming its superior practicality and effectiveness in real-world continual learning scenarios.
>
> __Table R.5.__ Experiments on the scalability utilizing ImageWoof and resized ImageNet datasets. Here ‘-’ indicates the training fails due to the out-of-memory problems.
> | Metric  | ImageWoof |ipc 1          | ImageWoof| ipc 10          | ImageNet| ipc 1             | ImageNet |ipc 2         |
> |:-           |:-       |:-                     |:-          |:-                   |:-                |:-           |:-                |:-          |
> |             | ACC       | NLL       | ACC       | NLL       | ACC       | NLL       | ACC       | NLL       |
> |:-           |:-       |:-                     |:-          |:-                   |:-                |:-           |:-                |:-          |
> | Random  | 14.2 ± 0.9 | 3.84 ± 0.25 | 27.0 ± 1.9 | 2.83 ± 0.33 | 1.1 ± 0.1 | 8.32 ± 0.05 | 1.4 ± 0.1 | 8.10 ± 0.05 |
> | BPC-CD  | 18.5 ± 0.1 | 2.76 ± 0.05 | -          | -          | -         | -          | -         | -          |
> | FBPC    | 14.8 ± 0.1 | 3.73 ± 0.02 | 28.1 ± 0.3 | 2.69 ± 0.09 | -         | -          | -         | -          |
> | BPC-fKL | 14.9 ± 0.9 | 3.74 ± 0.23 | 25.0 ± 0.8 | 2.90 ± 0.27 | -         | -          | -         | -          |
> | BPC-rkL | 12.0 ± 0.5 | 6.07 ± 0.31 | -          | -          | -         | -          | -         | -          |
> | VBPC    | **31.2 ± 0.1** | **2.13 ± 0.04** | **39.0 ± 0.1** | **1.84 ± 0.1** | **10.0 ± 0.1** | **5.33 ± 0.04** | **11.5 ± 0.2** | **5.25 ± 0.05** |
>
>
> __Table R.6.__ Experiments on the continual learning setting. Here, we utilize the CIFAR100 dataset with ipc 20 setting. We assume 5 steps during training and each step contains data from new 20 classes in the CIFAR100 dataset. Here we only report accuracy due to the variant of the number of classes during the steps.
> | Number of Classes |      20        |           40          |          60         |           80        |          100  |
> |---------------------------|---------------|---------------------|-------------------|-------------------|----------------|
> |BPC-CD                  | 52.5 ± 2.4   |             40.4 ± 1.3  |            35.2 ± 0.8 |           33.4 ± 0.5 |      29.4 ± 0.2 |
> |FBPC                      | 61.4 ±  1.8   |           53.2 ± 1.5  |             48.8 ± 0.7 |          43.9 ± 0.4 |      41.2 ± 0.3  |
> |BPC-fKL                  | 51.8 ± 2.2 |             39.8 ± 1.1  |            35.5 ±  0. 7|         33.1 ± 0.5  |      29.5 ± 0.3 |
> |BPC-rKL                  | 48.2 ± 2.7 |           35.5 ±  1.8    |            32.0 ± 1.0 |          29.8 ± 0.6 |       25.5 ± 0.3 |
> |VBPC                      | 75.3 ± 2.0 |           65.8± 1.5      |           57.1 ±  0.9  |        53.3 ± 0.5  |      50.3 ± 0.2 |

---

> ### Author Response · Authors · 2024-11-18
> **Rebuttal by Authors**
>
> **[W2, Q1] Would Laplace approximation on the softmax likelihood be a better option than first choosing a variational inference scheme and then using a Gaussian likelihood?**
>
> Thank you for asking such an insightful and constructive question. First, we place great importance on considering potential future directions to improve our approach. Here, we will discuss some concerns and challenges we foresee in adopting the reviewer’s suggestion.
>
> Specifically, if we switch from using a Gaussian likelihood to employing a softmax likelihood with Laplace approximation for variational inference, there are two cases to consider: (1) using Laplace approximation on the last-layer weights without any updates, and (2) updating the last-layer weights with some gradient descent steps before applying Laplace approximation.
>
> In the first case—applying Laplace approximation to weights without updating the last layer—two main issues may arise. First, the Laplace approximation assumes that the weights are near a minimum, allowing for the approximation of the first-order term in Taylor expansion as zero. However, this assumption may not hold for untrained weights, leading to significant approximation error. Additionally, the computational burden of calculating the Hessian for Laplace approximation is substantial, and the need to compute gradients through this Hessian during pseudo-coreset updates increases the computational load further.
>
> In the second case—updating the last layer weights with gradient steps before applying Laplace approximation—there’s the advantage of reducing Taylor expansion error. However, this approach involves a large computational graph, which can be problematic due to the computational expense typical in bilevel optimization settings. Additionally, the need to compute gradients through the Hessian remains a challenge.
>
> Overall, we believe that solving these issues could lead to new meaningful future work for VBPC.
>
> **[Q2] Is the claim of Manousakas et al. 2020 contrary to the main message of the paper?**
>
> Thank you for your comment. If we understand correctly, you’re referring to the paper *Bayesian Pseudocoresets* by Manousakas et al. (2020). We’re not entirely certain about which aspect of this work might be contrary to our approach and goals. Could you clarify which specific details or results from this paper you find relevant? Any additional insights on this point would help us provide a more precise and thorough response.
>
> **[Q3] Do we really lack an analytical solution or at least an EM-like algorithm where the E-step has an analytical solution when only the last layer of a neural net is probabilistic?**
>
> Thank you for the suggestion to further improve our VBPC method. While leveraging the EM algorithm is an intriguing idea, there are still practical challenges associated with its application.
>
> Primarily, the E-step and M-step of the EM algorithm lack closed-form solutions in this context. Even if we were to derive an approximation for one step to compute it in closed form, the other step would still require iterative computation. This sequential nature of the EM algorithm would lead to the accumulation of the computational graph during iterations, resulting in a memory-inefficient process when updating the pseudo-coreset.
>
> Despite these challenges, we agree that exploring efficient approximations to address these issues could significantly enhance the utility of VBPC. Investigating such improvements represents a **highly valuable future research direction**. We will include this discussion in the revised manuscript to acknowledge the potential of the EM algorithm as a future direction.

---

> ### Author Response · Authors · 2024-11-18
> **Rebuttal by Authors**
>
> **[Q4] What is the take-home message of Figure1?**
>
> Thank you for asking about the interpretability of the VBPC images. Regarding the learned pseudo-coreset images for CIFAR10, the results can be found in Figure 1 of the main paper and Figure 12 in the appendix, showing the outcomes for ipc values of 1 and 10. These images reveal several interesting aspects of how VBPC captures information.
>
> First, both ipc 1 and ipc 10 images show that VBPC effectively learns features associated with specific classes, such as "horse" or "automobile," as can be visually confirmed. This indicates that the pseudo-coreset images retain class-relevant information necessary for approximating the original dataset’s posterior distribution. When comparing ipc 1 and ipc 10, there are notable differences. In the case of ipc 1, where only a single image per class is available, VBPC attempts to encode as many class-specific features as possible into a single image. As a result, the learned image appears to incorporate multiple discriminative features from the class symmetrically. In contrast, with ipc 10, where more images per class are available, VBPC distributes the class-relevant features across multiple images. This leads to a greater diversity of features being captured across the pseudo-coreset, enabling a more comprehensive representation of the class.
>
> Additionally, both ipc 1 and ipc 10 images often include low-level features beyond the main class-relevant ones. These features likely help capture the dataset's variability and ensure the learned pseudo-coreset maintains a close approximation of the original data distribution.
>
> These observations suggest that VBPC is effective in compressing the dataset while retaining essential information. The learned images illustrate how VBPC balances feature extraction and information retention to ensure that the variational posterior distribution learned using the pseudo-coreset closely approximates the one learned using the full dataset. This further validates the interpretability and utility of VBPC in various tasks.
>
> **References**
>
> [1] Jeremy Howard. A smaller subset of 10 easily classified classes from imagenet, and a little more french, 2020. URL https://github.com/fastai/imagenette/
>
> [2] Olga Russakovsky, Jia Deng, Hao Su, Jonathan Krause, Sanjeev Satheesh, Sean Ma, Zhiheng Huang, Andrej Karpathy, Aditya Khosla, Michael Bernstein, Alexander C. Berg, and Li Fei-Fei. ImageNet Large Scale Visual Recognition Challenge. International Journal of Computer Vision (IJCV), 115(3):211–252, 2015.
>
> [3] Yongchao Zhou, Ehsan Nezhadarya, and Jimmy Ba. Dataset distillation using neural feature regression. In Advances in Neural Information Processing Systems 35 (NeurIPS 2022), 2022.
>
> [4] Bo Zhao and Hakan Bilen. Dataset condensation with distribution matching. CoRR, abs/2110.04181, 2021. URL https://arxiv.org/abs/2110.04181.

---

> ### Author Response · Authors · 2024-11-22
>
> Thank you for your dedication and interest in our paper. As the author and reviewer discussion period approaches its end, we are curious to know your thoughts on our rebuttal and whether you have any additional questions.

---

> ### Comment · Reviewer_sqcH · 2024-11-22
> **Thanks**
>
> Thanks for your detailed answer. This satisfies all my major concerns, in particular the scalability of the method to nontrivial tasks, which you successfully demonstrate as well as the clarification of the approximation procedure. I raise my score to an accept.

---

> ### Author Response · Authors · 2024-11-22
>
> Thank you for your positive response! We will incorporate the discussions and experimental results into the final manuscript.

---

### Official Review · Reviewer_8mt5 · 2024-11-02

**Soundness:** 3
**Presentation:** 3
**Contribution:** 2
**Rating:** 6
**Confidence:** 3

**Summary:**

This paper proposes Variational Bayesian Pseudo-Coreset (VBPC), a novel approach to efficiently approximate posterior distribution in Bayesian Neural Networks (BNNs). Bayesian Neural Networks often face issues with largescale datasets due to their high-dimensional parameter space. To reduce the computational load, many Bayesian Pseudo-Coreset (BPC) methods have been proposed, but they suffer from memory inefficiencies. VBPC addresses these limitations by using variational inference (VI) to approximate the posterior distribution. Moreover, this paper provides a memory-efficient method to approximate the predictive distribution with only a single forward pass instead of multiple forwards, making the approach computationally and memory-efficient.

**Strengths:**

- This paper leverages the variational formulation to obtain the closed-form posterior distribution of the last layer weights, which resolves the issue of suboptimal performance seen in previous approaches.
- And, the method approximates the predictive distribution with only a single forward pass instead of multiple forwards, making the approach computationally and memory-efficient.

**Weaknesses:**

- The experiment is not enough to illustrate the function of the algorithm.

**Questions:**

- The accuracy of these algorithms on cifar10, cifar100 and Tiny-Imagenet is too low. VBPC is effective relative to several existing BPC baselines, but the performance is significantly lower compared to existing state-of-the-art classifiers.
- In the field of image classification, at least the scenes you choose for classification, these experiments do not seem to show the advantages of your method convincingly.
- Please try to provide some new experiments, in more convincing scenarios, to illustrate the practical application value of your method.

---

> ### Author Response · Authors · 2024-11-18
> **Rebuttal by Authors**
>
> **[Q1] Compared to exisiting state-of-the-art classifier**
>
> Thank you for the considerable effort and assistance you've put into reviewing our paper. However, it seems there may be a misunderstanding regarding this particular weakness you pointed out. The goal of our method is not to develop a state-of-the-art model for a specific dataset, nor is it to create an inference method that achieves higher performance by leveraging existing state-of-the-art models. Rather, our research focuses on effectively summarizing a large volume of training data into a minimal yet well-representative set of data points, thereby reducing the computational and memory burdens needed for learning.
>
> For example, in the case of CIFAR10, we demonstrated that our method could achieve a strong accuracy of 55% using only 10 images (which is just 0.2% of the training data) instead of the original 60,000 images. Furthermore, the model and dataset setups we use are consistent with the benchmark configurations employed in various dataset distillation and Bayesian pseudo-coreset studies, such as [1,2,3,4,5,6]. These studies, for the sake of fair comparison, fix model architecture to certain layer sizes and kernel sizes, which, of course, results in models that may not match the performance of SOTA models. However, our VBPC method could indeed be practically utilized in conjunction with SOTA models to learn a pseudo-coreset.
>
> Notably, our method requires only the last layer for variational inference, making it significantly easier to apply to large models such as ViTs compared to existing Bayesian pseudo-coreset methods. A major drawback of previous BPC methods is that they require a pre-trained target model (e.g., ViT) along with a large number of expert trajectories obtained by training the ViT model multiple times with different random seeds. In contrast, our method does not require pre-training multiple ViT models, making it a much more efficient approach to pseudo-coreset learning.

---

> ### Author Response · Authors · 2024-11-18
> **Rebuttal by Authors**
>
> **[Q2] Convincing experiments**
>
> Although we conducted extensive additional ablation experiments alongside various tests commonly performed in BPC studies to demonstrate the effectiveness and efficiency of our VBPC approach (e.g., in terms of BMA performance, out-of-distribution performance, and model generalization), to further show that VBPC works for various scenarios which other BPC baselines do not work well, we have also performed two more experiments based on the reviewer's request.
>
> First, to show that our method is uniquely scalable to large datasets compared to other BPC methods, we conducted additional experiments on the ImageNetWoof (128x128x3) dataset and the ImageNet1k (64x64x3) dataset. Additionally, we included an experiment in a continual learning scenario to validate that our method performs better in practical scenarios.
>
> We conducted experiments on the ImageWoof (128x128x3) dataset with ipc 1 and ipc 10 settings, as well as the resized ImageNet1k (64x64x3) dataset with ipc 1 and ipc 2 settings, to demonstrate the scalability of our method to high-resolution images and larger datasets. Unlike existing BPC baselines, which encountered memory issues and failed to train due to out-of-memory errors on an RTX 3090 GPU as the image resolution and number of classes increased, our method successfully completed training. Table R.3 clearly shows that VBPC significantly outperforms other baselines with a large margin for both the ImageWoof and resized ImageNet1k datasets.
>
> Next, we validated the practical effectiveness of our method through continual learning experiments using pseudo-coreset images learned by each method. We followed the continual learning setup described in [4,7], where class-balanced training examples are greedily stored in memory, and the model is trained from scratch using only the latest memory. Specifically, we performed a 5-step class incremental learning experiment on CIFAR100 with an ipc 20 setting, following the class splits proposed in [4,7]. Table R.4 demonstrates that VBPC consistently outperforms other baselines across all steps, confirming its superior practicality and effectiveness in real-world continual learning scenarios.

---

> ### Author Response · Authors · 2024-11-18
> **Rebuttal by Authors**
>
> __Table R.3.__ Experiments on the scalability utilizing ImageWoof and resized ImageNet datasets. Here ‘-’ indicates the training fails due to the out-of-memory problems.
> | Metric  | ImageWoof |ipc 1          | ImageWoof| ipc 10          | ImageNet| ipc 1             | ImageNet |ipc 2         |
> |:-           |:-       |:-                     |:-          |:-                   |:-                |:-           |:-                |:-          |
> |             | ACC       | NLL       | ACC       | NLL       | ACC       | NLL       | ACC       | NLL       |
> |:-           |:-       |:-                     |:-          |:-                   |:-                |:-           |:-                |:-          |
> | Random  | 14.2 ± 0.9 | 3.84 ± 0.25 | 27.0 ± 1.9 | 2.83 ± 0.33 | 1.1 ± 0.1 | 8.32 ± 0.05 | 1.4 ± 0.1 | 8.10 ± 0.05 |
> | BPC-CD  | 18.5 ± 0.1 | 2.76 ± 0.05 | -          | -          | -         | -          | -         | -          |
> | FBPC    | 14.8 ± 0.1 | 3.73 ± 0.02 | 28.1 ± 0.3 | 2.69 ± 0.09 | -         | -          | -         | -          |
> | BPC-fKL | 14.9 ± 0.9 | 3.74 ± 0.23 | 25.0 ± 0.8 | 2.90 ± 0.27 | -         | -          | -         | -          |
> | BPC-rkL | 12.0 ± 0.5 | 6.07 ± 0.31 | -          | -          | -         | -          | -         | -          |
> | VBPC    | **31.2 ± 0.1** | **2.13 ± 0.04** | **39.0 ± 0.1** | **1.84 ± 0.1** | **10.0 ± 0.1** | **5.33 ± 0.04** | **11.5 ± 0.2** | **5.25 ± 0.05** |
>
> __Table R.4__ Experiments on the continual learning setting. Here, we utilize the CIFAR100 dataset with ipc 20 setting. We assume 5 steps during training and each step contains data from new 20 classes in the CIFAR100 dataset. Here we only report accuracy due to the variant of the number of classes during the steps.
> | Number of Classes |      20        |           40          |          60         |           80        |          100  |
> |---------------------------|---------------|---------------------|-------------------|-------------------|----------------|
> |BPC-CD                  | 52.5 ± 2.4   |             40.4 ± 1.3  |            35.2 ± 0.8 |           33.4 ± 0.5 |      29.4 ± 0.2 |
> |FBPC                      | 61.4 ±  1.8   |           53.2 ± 1.5  |             48.8 ± 0.7 |          43.9 ± 0.4 |      41.2 ± 0.3  |
> |BPC-fKL                  | 51.8 ± 2.2 |             39.8 ± 1.1  |            35.5 ±  0. 7|         33.1 ± 0.5  |      29.5 ± 0.3 |
> |BPC-rKL                  | 48.2 ± 2.7 |           35.5 ±  1.8    |            32.0 ± 1.0 |          29.8 ± 0.6 |       25.5 ± 0.3 |
> |VBPC                      | 75.3 ± 2.0 |           65.8± 1.5      |           57.1 ±  0.9  |        53.3 ± 0.5  |      50.3 ± 0.2 |
>
>
> References
>
> [1] Balhae Kim, Jungwon Choi, Seanie Lee, Yoonho Lee, Jung-Woo Ha, and Juho Lee. On divergence measures for bayesian pseudocoresets. In Advances in Neural Information Processing Systems 35 (NeurIPS 2022), 2022.
>
> [2] Balhae Kim, Hyungi Lee, and Juho Lee. Function space bayesian pseudocoreset for bayesian neural networks. In Advances in Neural Information Processing Systems 36 (NeurIPS 2023), 2023.
>
> [3] Piyush Tiwary, Kumar Shubham, Vivek V Kashyap, and AP Prathosh. Bayesian pseudo-coresets via contrastive divergence. In The 40th Conference on Uncertainty in Artificial Intelligence, 2024.
>
> [4] Yongchao Zhou, Ehsan Nezhadarya, and Jimmy Ba. Dataset distillation using neural feature regression. In Advances in Neural Information Processing Systems 35 (NeurIPS 2022), 2022.
>
> [5] Noel Loo, Ramin Hasani, Mathias Lechner, and Daniela Rus. Dataset distillation with convexified implicit gradients. In Proceedings of The 39th International Conference on Machine Learning (ICML 2023), 2023.
>
> [6] George Cazenavette, Tongzhou Wang, Antonio Torralba, Alexei A Efros, and Jun-Yan Zhu. Dataset distillation by matching training trajectories. In Proceedings of the IEEE/CVF Conference on Computer Vision and Pattern Recognition, pp. 4750–4759, 2022.
>
> [7] Bo Zhao and Hakan Bilen. Dataset condensation with distribution matching. CoRR, abs/2110.04181, 2021. URL https://arxiv.org/abs/2110.04181.
>
> [8] Jeremy Howard. A smaller subset of 10 easily classified classes from imagenet, and a little more french, 2020. URL https://github.com/fastai/imagenette/
>
> [9] Olga Russakovsky, Jia Deng, Hao Su, Jonathan Krause, Sanjeev Satheesh, Sean Ma, Zhiheng Huang, Andrej Karpathy, Aditya Khosla, Michael Bernstein, Alexander C. Berg, and Li Fei-Fei. ImageNet Large Scale Visual Recognition Challenge. International Journal of Computer Vision (IJCV), 115(3):211–252, 2015.

---

> ### Author Response · Authors · 2024-11-22
>
> Thank you for your dedication and interest in our paper. As the author and reviewer discussion period approaches its end, we are curious to know your thoughts on our rebuttal and whether you have any additional questions.

---

> > ### Comment · Reviewer_8mt5 · 2024-11-22
> >
> > Thank you for your thoughtful and detailed response. Your explanation has basically resolved most of the issues I was concerned about and provided clear guidance on how to address them. I will raise your score.

---

> ### Author Response · Authors · 2024-11-22
>
> Thank you for the positive review of our paper. We will organize the experiments and discussions conducted during the discussion period and incorporate them into the final manuscript.

---

### Official Review · Reviewer_hhFq · 2024-11-02

**Soundness:** 3
**Presentation:** 3
**Contribution:** 3
**Rating:** 8
**Confidence:** 2

**Summary:**

The paper presents the Variational Bayesian Pseudo-Coreset (VBPC) method, aimed at efficiently approximating the posterior distribution in Bayesian Neural Networks (BNNs). Given that BNNs face substantial computational challenges when dealing with large datasets due to their high-dimensional parameter spaces, VBPC provides a promising method. Traditional Bayesian Pseudo-Coreset (BPC) techniques have been proposed to alleviate these issues, yet they often struggle with memory inefficiencies. VBPC addresses this by leveraging variational inference (VI) to approximate the posterior distribution. This method achieves a memory-efficient approximation of the predictive distribution using only a single forward pass, which makes it appealing for computationally intensive applications.

**Strengths:**

The paper effectively utilizes variational inference to derive a closed-form posterior distribution for the weights of the last layer, thereby addressing some of the performance limitations observed in prior BPC approaches.
VBPC’s capability to approximate the predictive distribution in a single forward pass enhances both computational and memory efficiency, positioning it as a potentially valuable method for large-scale applications.

**Weaknesses:**

The experimental validation on practical application is limited.

**Questions:**

- While VBPC demonstrates improvement over some BPC baselines, its classification accuracy on benchmark datasets like CIFAR-10, CIFAR-100, and Tiny-ImageNet remains notably lower than that of state-of-the-art classifiers. This raises concerns about the practical competitiveness of VBPC in real-world applications, particularly in image classification tasks where accuracy is crucial. Could additional optimizations or refinements to the VBPC approach improve performance?
- The current experiments may not adequately showcase the strengths of VBPC in relevant scenarios. To enhance the paper’s impact and applicability, I suggest conducting additional experiments in settings where VBPC’s efficiency gains could be more convincingly demonstrated.

---

> ### Author Response · Authors · 2024-11-18
> **Rebuttal by authors**
>
> **[Q1] Performance is lower than that of state-of-the-art classifiers. Could additional optimizations or refinements to the VBPC approach improve performance?**
>
> Thank you for the considerable effort and assistance you've put into reviewing our paper. However, it seems there may be a misunderstanding regarding this particular weakness you pointed out. The goal of our method is not to develop a state-of-the-art model for a specific dataset, nor is it to create an inference method that achieves higher performance by leveraging existing state-of-the-art models. Rather, our research focuses on effectively summarizing a large volume of training data into a minimal yet well-representative set of data points, thereby reducing the computational and memory burdens needed for learning.
>
> For example, in the case of CIFAR10, we demonstrated that our method could achieve a strong accuracy of 55% using only 10 images (which is just 0.2% of the training data) instead of the original 60,000 images. Furthermore, the model and dataset setups we use are consistent with the benchmark configurations employed in various dataset distillation and Bayesian pseudo-coreset studies, such as [1,2,3,4,5,6]. These studies, for the sake of fair comparison, fix model architecture to certain layer sizes and kernel sizes, which, of course, results in models that may not match the performance of SOTA models. However, our VBPC method could indeed be practically utilized in conjunction with SOTA models to learn a pseudo-coreset.
>
> Notably, our method requires only the last layer for variational inference, making it significantly easier to apply to large models such as ViTs compared to existing Bayesian pseudo-coreset methods. A major drawback of previous BPC methods is that they require a pre-trained target model (e.g., ViT) along with a large number of expert trajectories obtained by training the ViT model multiple times with different random seeds. In contrast, our method does not require pre-training multiple ViT models, making it a much more efficient approach to pseudo-coreset learning.

---

> ### Author Response · Authors · 2024-11-18
> **Rebuttal by Authors**
>
> **[Q2] The current experiments may not adequately showcase the strengths of VBPC in relevant scenarios. Conducting additional experiments in setting where VBPC’s efficiency gains could be more convincingly demonstrated.**
>
> Although we conducted extensive additional ablation experiments alongside various tests commonly performed in BPC studies to demonstrate the effectiveness and efficiency of our VBPC approach (e.g., in terms of BMA performance, out-of-distribution performance, and model generalization), to further show that VBPC works for various scenarios which other BPC baselines do not work well, we have also performed two more experiments based on the reviewer's request.
>
> First, to show that our method is uniquely scalable to large datasets compared to other BPC methods, we conducted additional experiments on the ImageNetWoof (128x128x3) dataset [8] and the resized ImageNet1k (64x64x3) dataset [9]. Additionally, we included an experiment in a continual learning scenario to validate that our method performs better in practical scenarios.
>
> We conducted experiments on the ImageWoof (128x128x3) dataset with ipc 1 and ipc 10 settings, as well as the resized ImageNet1k (64x64x3) dataset with ipc 1 and ipc 2 settings, to demonstrate the scalability of our method to high-resolution images and larger datasets. Unlike existing BPC baselines, which encountered memory issues and failed to train due to out-of-memory errors on an RTX 3090 GPU as the image resolution and number of classes increased, our method successfully completed training. Table R.1 clearly shows that VBPC significantly outperforms other baselines with a large margin for both the ImageWoof and resized ImageNet1k datasets.
>
> Next, we validated the practical effectiveness of our method through continual learning experiments using pseudo-coreset images learned by each method. We followed the continual learning setup described in [4,7], where class-balanced training examples are greedily stored in memory, and the model is trained from scratch using only the latest memory. Specifically, we performed a 5-step class incremental learning experiment on CIFAR100 with an ipc 20 setting, following the class splits proposed in [4,7]. Table R.2 demonstrates that VBPC consistently outperforms other baselines across all steps, confirming its superior practicality and effectiveness in real-world continual learning scenarios.

---

> ### Author Response · Authors · 2024-11-18
> **Rebuttal by Authors**
>
> __Table R.1.__ Experiments on the scalability utilizing ImageWoof and resized ImageNet datasets. Here ‘-’ indicates the training fails due to the out-of-memory problems.
>
> | Metric  | ImageWoof |ipc 1          | ImageWoof| ipc 10          | ImageNet| ipc 1             | ImageNet |ipc 2         |
> |:-           |:-       |:-                     |:-          |:-                   |:-                |:-           |:-                |:-          |
> |             | ACC       | NLL       | ACC       | NLL       | ACC       | NLL       | ACC       | NLL       |
> |:-           |:-       |:-                     |:-          |:-                   |:-                |:-           |:-                |:-          |
> | Random  | 14.2 ± 0.9 | 3.84 ± 0.25 | 27.0 ± 1.9 | 2.83 ± 0.33 | 1.1 ± 0.1 | 8.32 ± 0.05 | 1.4 ± 0.1 | 8.10 ± 0.05 |
> | BPC-CD  | 18.5 ± 0.1 | 2.76 ± 0.05 | -          | -          | -         | -          | -         | -          |
> | FBPC    | 14.8 ± 0.1 | 3.73 ± 0.02 | 28.1 ± 0.3 | 2.69 ± 0.09 | -         | -          | -         | -          |
> | BPC-fKL | 14.9 ± 0.9 | 3.74 ± 0.23 | 25.0 ± 0.8 | 2.90 ± 0.27 | -         | -          | -         | -          |
> | BPC-rkL | 12.0 ± 0.5 | 6.07 ± 0.31 | -          | -          | -         | -          | -         | -          |
> | VBPC    | **31.2 ± 0.1** | **2.13 ± 0.04** | **39.0 ± 0.1** | **1.84 ± 0.1** | **10.0 ± 0.1** | **5.33 ± 0.04** | **11.5 ± 0.2** | **5.25 ± 0.05** |
>
> __Table R.2__ Experiments on the continual learning setting. Here, we utilize the CIFAR100 dataset with ipc 20 setting. We assume 5 steps during training and each step contains data from new 20 classes in the CIFAR100 dataset. Here we only report accuracy due to the variant of the number of classes during the steps.
> | Number of Classes |      20        |           40          |          60         |           80        |          100  |
> |---------------------------|---------------|---------------------|-------------------|-------------------|----------------|
> |BPC-CD                  | 52.5 ± 2.4   |             40.4 ± 1.3  |            35.2 ± 0.8 |           33.4 ± 0.5 |      29.4 ± 0.2 |
> |FBPC                      | 61.4 ±  1.8   |           53.2 ± 1.5  |             48.8 ± 0.7 |          43.9 ± 0.4 |      41.2 ± 0.3  |
> |BPC-fKL                  | 51.8 ± 2.2 |             39.8 ± 1.1  |            35.5 ±  0. 7|         33.1 ± 0.5  |      29.5 ± 0.3 |
> |BPC-rKL                  | 48.2 ± 2.7 |           35.5 ±  1.8    |            32.0 ± 1.0 |          29.8 ± 0.6 |       25.5 ± 0.3 |
> |VBPC                      | 75.3 ± 2.0 |           65.8± 1.5      |           57.1 ±  0.9  |        53.3 ± 0.5  |      50.3 ± 0.2 |
>
> References
>
> [1] Balhae Kim, Jungwon Choi, Seanie Lee, Yoonho Lee, Jung-Woo Ha, and Juho Lee. On divergence measures for bayesian pseudocoresets. In Advances in Neural Information Processing Systems 35 (NeurIPS 2022), 2022.
>
> [2] Balhae Kim, Hyungi Lee, and Juho Lee. Function space bayesian pseudocoreset for bayesian neural networks. In Advances in Neural Information Processing Systems 36 (NeurIPS 2023), 2023.
>
> [3] Piyush Tiwary, Kumar Shubham, Vivek V Kashyap, and AP Prathosh. Bayesian pseudo-coresets via contrastive divergence. In The 40th Conference on Uncertainty in Artificial Intelligence, 2024.
>
> [4] Yongchao Zhou, Ehsan Nezhadarya, and Jimmy Ba. Dataset distillation using neural feature regression. In Advances in Neural Information Processing Systems 35 (NeurIPS 2022), 2022.
>
> [5] Noel Loo, Ramin Hasani, Mathias Lechner, and Daniela Rus. Dataset distillation with convexified implicit gradients. In Proceedings of The 39th International Conference on Machine Learning (ICML 2023), 2023.
>
> [6] George Cazenavette, Tongzhou Wang, Antonio Torralba, Alexei A Efros, and Jun-Yan Zhu. Dataset distillation by matching training trajectories. In Proceedings of the IEEE/CVF Conference on Computer Vision and Pattern Recognition, pp. 4750–4759, 2022.
>
> [7] Bo Zhao and Hakan Bilen. Dataset condensation with distribution matching. CoRR, abs/2110.04181, 2021. URL https://arxiv.org/abs/2110.04181.
>
> [8] Jeremy Howard. A smaller subset of 10 easily classified classes from imagenet, and a little more french, 2020. URL https://github.com/fastai/imagenette/
>
> [9] Olga Russakovsky, Jia Deng, Hao Su, Jonathan Krause, Sanjeev Satheesh, Sean Ma, Zhiheng Huang, Andrej Karpathy, Aditya Khosla, Michael Bernstein, Alexander C. Berg, and Li Fei-Fei. ImageNet Large Scale Visual Recognition Challenge. International Journal of Computer Vision (IJCV), 115(3):211–252, 2015.

---

> ### Author Response · Authors · 2024-11-22
>
> Thank you for your dedication and interest in our paper. As the author and reviewer discussion period approaches its end, we are curious to know your thoughts on our rebuttal and whether you have any additional questions.

---

> > ### Comment · Reviewer_hhFq · 2024-11-22
> >
> > Thank you for your detailed response. You have addressed my concerns thoroughly. I will raise your score.

---

> ### Author Response · Authors · 2024-11-22
>
> Thank you for the positive review of our paper. We will organize the experiments and discussions conducted during the discussion period and incorporate them into the final manuscript.

---

### Author Response · Authors · 2024-11-18
**General response**

We sincerely thank all reviewers for their thoughtful and detailed feedback on our work. We are pleased that the reviewers acknowledged the strengths of our approach, including its computational and memory efficiency enabled by approximating the predictive distribution in a single forward pass (hhFq, 8mt5). We are also grateful that the well-structured problem formulation and methodological rigor, particularly the bilevel variational inference formulation and computational complexity analysis, were recognized as key merits of our paper (sqcH). Furthermore, we appreciate the recognition of the practical contributions of VBPC, notably its potential to enhance the scalability of Bayesian Neural Networks through variational inference and pseudo-coresets while addressing challenges of prior BPC methods (oE39).
And also, we thank the reviewers for their encouraging remarks regarding the clarity, robustness, and potential impact of our proposed method. Their insights provide valuable validation of our contributions and motivate us to continue improving the manuscript.

---

### Author Response · Authors · 2024-11-28
**Revision for Paper**

We sincerely thank all the reviewers for their valuable time and effort in helping to improve our paper and enhance its completeness through additional analyses, ablation studies, and new tasks showcasing the effectiveness of VBPC. As the discussion period concludes, we would like to share that we have incorporated the feedback and additional experiments into our revised manuscript. We promise that we will more carefully review and revise for the final manuscript.

The key updates in our revision include:

- Discussion on the Laplace approximation with softmax likelihood: Sec D
- Discussion on the Last-Layer Approximation: Sec D
- Additional experiments on large datasets and continual learning: Sec F.2
- Additional experiments for OOD detection: Sec F.3
- Computational cost analysis: Sec F.4
- Key insights from learned images: Sec G.1

---

### Meta-Review · Area_Chair_2R4d · 2024-12-21

**Metareview:**

The authors propose a solution to the coreset problem that uses a two-stage variational inference approach. Four authors reviewed the paper and three recommended accept after the rebuttal period, with one borderline reject. They felt the paper was well-written and motivated, technically well developed and original, and contained a compelling set of results on good data sets. The authors provided a convincing additional set of results in the rebuttal. Space permitting, these may be incorporated in the final draft, but the paper is strong enough as is and should be accepted to the conference.

**Additional Comments On Reviewer Discussion:**

The authors raised valid concerns about review authenticity from two reviewers who submitted extremely similar reviews. While AI was suggested, it is notable to me that these two reviewers are both PhD students at the same university and so may have simply been working together on their reviews, which of course is still inappropriate. However, that then raises the question why they thought it wouldn't be noticed, and so AI may have been used. I've discounted their opinions, but recognize that the authors provided a thorough response to all reviewers and were able to address the concerns of all reviewers in their (and my) opinions. I have also read the paper and think it is technically interesting and well developed, with good and detailed empirical evaluation. I think it would make a good contribution to the conference and recommend acceptance.

---

### Decision · Program_Chairs · 2025-01-22

Accept (Poster)